# sciCSR infers B cell state transition and predicts class-switch recombination dynamics using single-cell transcriptomic data

Joseph C. F. Ng [1,4] ✉, Guillem Montamat Garcia [2,4], Alexander T. Stewart[3,4], Paul Blair[2], Claudia Mauri[2], Deborah K. Dunn-Walters[3] & Franca Fraternali[1] ✉

Class-switch recombination (CSR) is an integral part of B cell maturation. Here we present sciCSR (pronounced 'scissor', single-cell inference of class-switch recombination), a computational pipeline that analyzes CSR events and dynamics of B cells from single-cell RNA sequencing (scRNA-seq) experiments. Validated on both simulated and real data, sciCSR re-analyzes scRNA-seq alignments to differentiate productive heavy-chain immunoglobulin transcripts from germline 'sterile' transcripts. From a snapshot of B cell scRNA-seq data, a Markov state model is built to infer the dynamics and direction of CSR. Applying sciCSR on severe acute respiratory syndrome coronavirus 2 vaccination time-course scRNA-seq data, we observe that sciCSR predicts, using data from an earlier time point in the collected time-course, the isotype distribution of B cell receptor repertoires of subsequent time points with high accuracy (cosine similarity ~0.9). Using processes specific to B cells, sciCSR identifies transitions that are often missed by conventional RNA velocity analyses and can reveal insights into the dynamics of B cell CSR during immune response.

B cells are the main drivers of the humoral response in developing protection against infectious diseases. An understanding of how this process is regulated over time is crucial to evaluate the quality of the antibodies produced and, in turn, the effectiveness of vaccination and therapeutic strategies[1]. B cells mature from a naive state to acquire memory against the antigen and differentiate into antibody-producing plasma cells[1]. Traditionally, the dynamics of B cell maturation is investigated by isolating different B cell subsets based on their surface protein expression, and comparing the proportion of these subsets before and after antigen exposure, or through a more detailed time course[2,3]. Such subsets are typically distinct cell states, and one

would require additional experimental evidence to probe the molecular details of how cells transition from one state into another. Single-cell profiling offers rich descriptions of the transcriptomic features of B cell states and how they are altered during B cell maturation[4,5]. In combination with B cell receptor (BCR) repertoires, these data have yielded great insights into the heterogeneity and functional relevance of different B cell subsets in health and disease[4–7].

In the past few years a plethora of computational methods have been created to infer the dynamics of cell state transitions from single-cell RNA sequencing (scRNA-seq) data. Typically they fall into two categories: the first belongs to a large family of methods which

[1]Department of Structural and Molecular Biology, Division of Biosciences and Institute of Structural and Molecular Biology, University College London, London, UK. [2]Division of Infection and Immunity and Institute of Immunity and Transplantation, Royal Free Hospital, University College London, London, UK. [3]School of Biosciences and Medicine, University of Surrey, Guildford, UK. [4]These authors contributed equally: Joseph C. F. Ng, Guillem Montamat Garcia, Alexander T. Stewart. ✉e-mail: joseph.ng@ucl.ac.uk; f.fraternali@ucl.ac.uk

infer 'pseudotime' ordering of cells[8], and the second corresponds to exploiting the balance between unspliced and spliced reads to infer what is called 'RNA velocity' (refs. [9,10]). Pseudotemporal ordering is derived from 'trajectories' of cell differentiation fitted to the data by imposing directionality onto a transcriptional similarity network of cells. The directionality can either be indicated by using prior knowledge (requiring users to specify start and end states, for example, slingshot[11]), or be learned from the data (for example, Monocle[12]). As such, cells can be ordered along a 'pseudotime' axis often understood to encode the differentiation potential of cells[13]. The second category of tools which describe cell transition dynamics estimate 'RNA velocity' by considering the splicing kinetics for genes[9,10]. The observation of nascent, unspliced messenger RNA and its mature, spliced counterparts across single cells allows one to extrapolate the future state of the system given its current state[9]. Recent years have seen methodological development in the estimation of RNA velocity, accounting for variations of splicing rates and expression levels across genes to improve the reconstruction of cell differentiation trajectories[10,14]. This accompanies experimental approaches to capture the time component of cellular development, for example, lineage tracing[15] and metabolite labeling[16] approaches coupled to scRNA-seq data generation. Computationally, methods such as CellRank[17] build on both RNA velocity and pseudotime methods, to fit statistical models which describe the overall dynamics of cell state transitions observed in the data. These methods, although successful in experimental validation, are designed and tested on use cases in developmental biology, where typically both the progenitor(s) and mature cell state(s) are transcriptionally distinct and well defined. Hematopoiesis has been studied with scRNA-seq, including the application of RNA velocity and pseudotime ordering tools, to some degree of success[8]. On the other hand, it is posited that samples such as peripheral blood mononuclear cells (PBMCs) are cell types with already equilibrated RNA metabolism, lacking the dynamic information required for RNA velocity analysis. This limitation results in noisy velocity profiles and erroneous inference of transitions[18]. While computational approaches begin to offer diagnostic approaches to determine the suitability of RNA velocity analysis on these datasets[14], a viable alternative to study transitions in scRNA-seq datasets of immune cell types is lacking.

This raises methodological questions on how to improve the approaches to study transition dynamics in immunological systems, especially mature cell types such as B cells in circulation and in secondary lymphoid tissues. Notably, B cells continue to mature after exiting the bone marrow by somatic hypermutation (SHM) and class-switch recombination (CSR)[19,20] to optimize the BCR for function. SHM targets the variable (V), diversity (D) and joining (J) gene segments and introduces mutations catalyzed by activation-induced cytidine deaminase (AID)[21,22] (Fig. 1a, left) to optimize antigen binding. On the other hand, CSR alters the constant (C) region to adapt the BCR to function in different immune challenges and tissue contexts. CSR also depends on AID, which recognizes 'switch' genomic regions 5' to each C gene enriched in motifs such as 5'-AGCT-3' (refs. [20,22,23]) and catalyzes deamination[20,22] (Fig. 1a, right). The repair of these mutational events entails DNA recombination that brings the immunoglobulin variable region proximal to a downstream C gene that, when transcribed and translated, would encode an immunoglobulin molecule switched to this isotype[24]. AID is targeted to the desired downstream C gene in part by transcriptional activity caused by specific combinations of molecular signals; such transcription initiates at positions immediately 5' to the target C gene and, as such, lacks the VDJ gene segments to encode a fully functional immunoglobulin heavy chain (IgH)[20,25]. The production of these germline or 'sterile' transcripts therefore signifies predisposition to CSR events[20,26,27] (Fig. 1a, right). Theoretically, both CSR and SHM would leave transcriptomic footprints (BCR sequence substitutions for SHM; sterile transcripts for CSR) that can be captured in scRNA-seq and scBCR-seq (that is, profiling the BCR

repertoire barcoded with originating cell and molecule identifiers) data (Fig. 1b,c). Neither standard RNA velocity or pseudotime methods have access to CSR and SHM, but it is these processes that are of interest in describing B cell maturation and the function of the resultant antibodies secreted during an immune response[20,28]. Extracting signals of CSR and SHM from the data, and utilizing them to infer B cell transitions, would complement existing velocity/pseudotime methods and yield a more faithful reconstruction of B cell maturation dynamics from single-cell transcriptomic data.

In this Article, we introduce sciCSR (pronounced 'scissor', acronym for 'single-cell inference of class switch recombination'), a computational tool designed to infer state transitions between mature B cell states and predict the direction of CSR in B cells assayed using scRNA-seq. Motivated by the need to faithfully capture B cell state transitions, sciCSR implements routines which extract B-cell-native information, such as CSR and SHM, from scRNA-seq data and, if available, BCR repertoire of the same population of B cells. This information is used as input to CellRank[17] to infer transition probabilities between cells. We further applied transition path theory (TPT) onto this transition matrix to analyze the dynamics of transitions between cell clusters. This method, popularized in analyzing conformational state transitions in molecular dynamics (MD) simulation data[29,30], describes transitions by considering an ensemble of possible paths governed by the given transition matrix and, as such, allows for more flexible analysis of transitions beyond visualization of arrows depicting velocity streams that are typical in RNA velocity analysis. We validated sciCSR using immunization, in vitro cell culture and gene knockout studies, and showed that sciCSR could recover BCR isotype distributions at the steady state and capture CSR dynamics, which could be subsequently verified over a time course.

## Results

### Extracting CSR signal from scRNA-seq data

We hypothesize that the use of information native to B cells, namely CSR and SHM, would improve the inference of cell state transitions in mature B cells as observed in scRNA-seq data. SHM level can be easily retrieved from BCR sequencing data, which are routinely obtained in parallel to transcriptome-wide single-cell profiling: SHM is inversely related to the sequence identity between the observed BCR sequence and the corresponding germline immunoglobulin gene[31,32] (Fig. 1b). The detection of CSR, on the other hand, is less trivial at the single-cell level: CSR is more easily characterized by either comparing the expression level of different IgH isotypes at the transcript or protein level for a cell population[33], or by studying BCR clonotypes that comprise sequences of different isotypes[31]. In sciCSR a series of routines has been implemented to distinguish productive and sterile IgH transcripts from scRNA-seq data, by enumerating mRNA molecules (identifiable by observing combinations of cell barcode and unique molecular identifier (UMI)) with reads mapping to the VDJ, 5' C or C regions in the IgH genomic locus (Fig. 1c). Molecules with insufficient evidence to be classified as productive or sterile (for example, if only one read mapped to the C exonic regions is found) are labeled as 'uninformative'. This quantification complements conventional data processing workflow: in standard genome annotations, individual immunoglobulin V, D, J and C genes are treated as separate gene entities; reads that are 5' of C genes would therefore be considered as intergenic reads typically discarded in transcriptomic data analysis workflows.

We first asked whether these 5' C reads could be detected in B cell scRNA-seq data. We previously sorted mature, circulating human B cells into different phenotypically defined subsets and generated scRNA-seq libraries[4] using the 5' 10x Genomics protocol; Fig. 2a shows the distribution of sequencing reads across the IgH genomic locus. We observed reads mapped to regions 5' to *IGHG1* and *IGHG2* in unswitched B cell subsets (CD19+IgD+CD27−CD10− naive B cells and CD19+IgD+CD27+ IgM memory cells), and notably with peaks at the transcription start

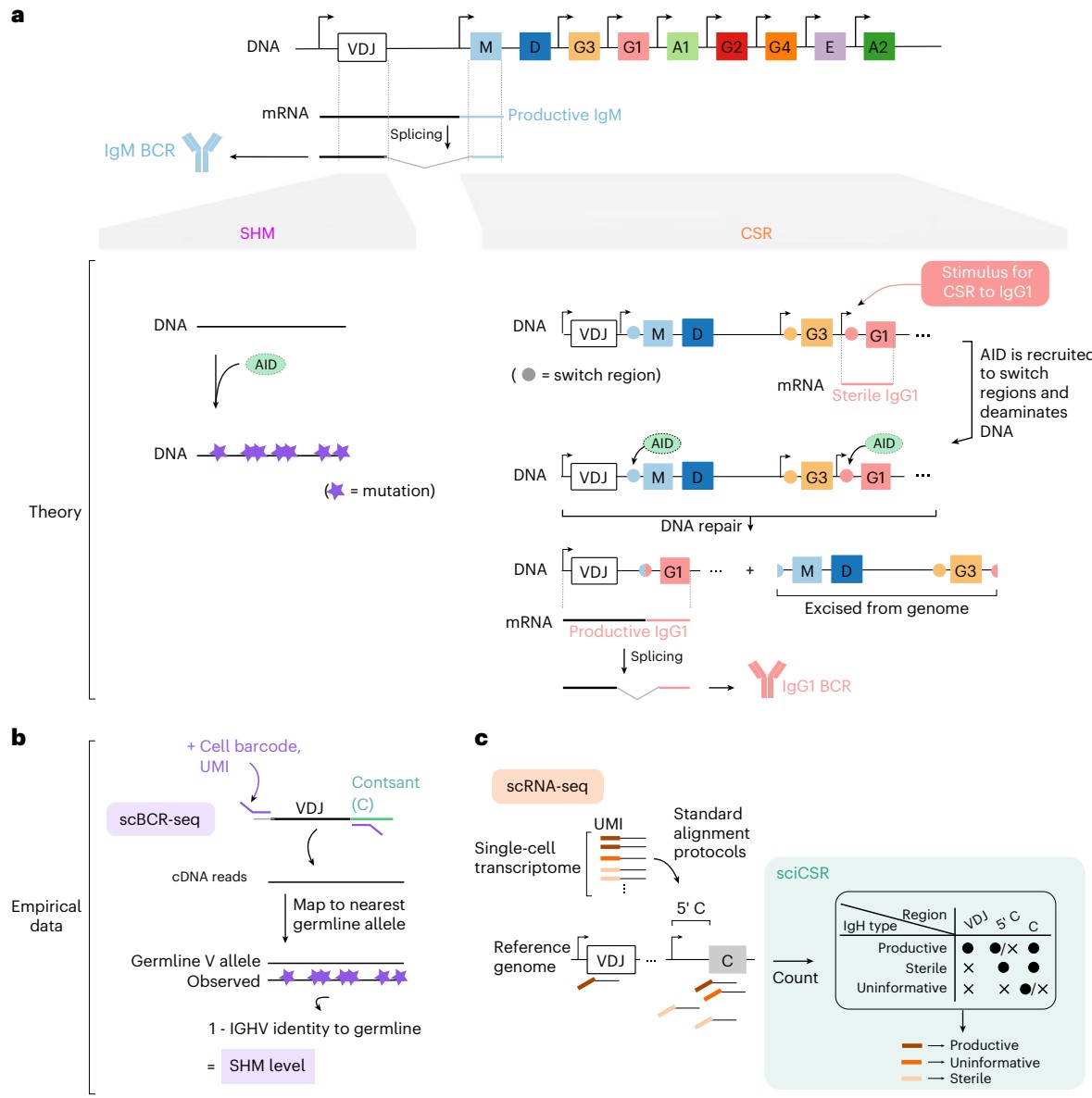

**Fig. 1 | Overview of the theories of SHM and CSR and their footprints in empirical data. a**, Schematic of the IgH genomic locus to illustrate SHM and CSR in B cells. For details, see Results. **b,c**, Both SHM and CSR can be observed in sequencing datasets: SHM can be analyzed in scBCR-seq (**b**), by comparing the observed VDJ sequences with the nearest germline allele and enumerating mutations. CSR can be analyzed in scRNA-seq (**c**); sciCSR provides functionalities to reconstitute each heavy-chain transcript as productive or sterile, based on whether reads mapping to the VDJ, C or 5′ C (that is, region 5′ to the C region coding segment) regions can be observed for each transcript. Dots and crosses in the table denote presence and absence of reads mapping to these regions, respectively. IGHV, immunoglobulin heavy-chain variable genes.

sites (TSSs) (Fig. 2a and Extended Data Fig. 1). While these reads could be consistent with either genuine sterile IgH transcripts (and therefore suggesting these cells are poised to CSR), or productive IgH transcripts where the intron between the variable and constant regions is yet to be spliced, we reason that the latter case is unlikely given the scRNA-seq library preparation protocol enriches mRNA at the 5′ end: in our data we observe that reads that were concentrated at typically around 600 base pairs downstream of TSS (when inspecting single-exon transcripts, for example, that of *JUNB* and *RHOB*, to eliminate splicing effects on the read distributions, see Extended Data Fig. 2), and the IgH read peaks in Fig. 2a would have been around 1–2 kilobase downstream of the TSS of a productive IgH transcript, thus discounting the likelihood that IgH transcripts bearing the VDJ segments were the sources of these 5′ C reads.

Applying the sciCSR routine depicted in Fig. 1 to simulated reads sampled from the human IgH genomic locus (Methods), we confirm that commonly used RNA read aligners STAR[34] (which is used in the 10x cellranger preprocessing workflow) and HISAT2 (ref. 35) can accurately identify the isotype corresponding to sterile IgH transcripts (Fig. 2b); notably, these aligners are precise in identifying the exact IgG and IgA subtypes, supporting the direct use of aligned reads generated by these data preprocessing pipelines to analyze productive/sterile IgH transcription in sciCSR. In further support of this, we find that both aligners can accurately recover the composition of mixtures of sterile IgG and IgA transcripts simulated with fixed proportion of each subtype (Fig. 2c). The major requirement for sciCSR is the adoption of a 5′ enrichment protocol during library preparation, as reads biased to the 3′ end do not capture the 5′ C region necessary to define sterile transcripts (Fig. 2b).

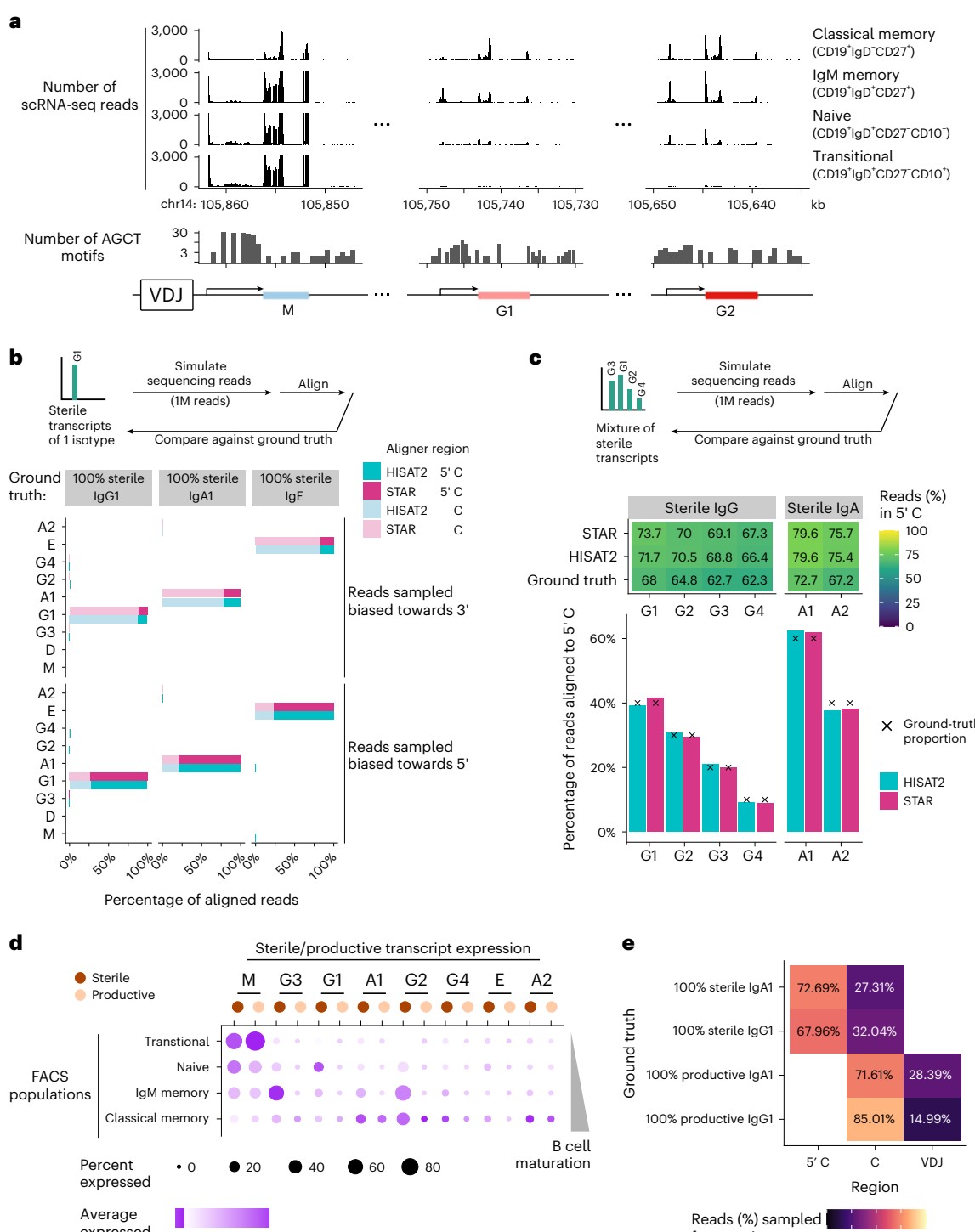

**Fig. 2 | sciCSR reconstitutes productive and sterile IgH transcription levels in different B cell subsets. a**, Histograms of scRNA-seq reads for mature human B cells from peripheral blood FACS-sorted into different B cell subsets (data from Stewart et al.). Counts of 5′-AGCT-3′ motif are displayed to indicate the locations of switch regions. Only reads mapped to IgM, IgG1 and IgG2 are shown here; for the complete IgH locus, see Extended Data Fig. 1. Notice the IgH coding sequences are on the minus strand of human chromosome 14; the horizontal axis is depicted in reverse. **b**, Alignment of simulated reads (using the polyester package), sampled from IgG1, IgA1 and IgE which are mapped to the 5′ C (light shade) or the C exons (dark shade), using either HISAT2 (teal) or STAR (magenta). Reads sampling are biased either to the 3′ (top) or the 5′ (bottom) end of transcripts to mimic typical scRNA-seq library preparation protocols. **c**, Alignment of polyester-simulated reads sampled from sterile transcripts of

a mixture of IgG (left) or IgA (right) subtypes. The heatmap at the top depicts the proportion of sampled reads mapped to the 5′ C region that is informative of indicating sterile IgH. The bar plots depict the proportions of reads aligned to each isotype using HISAT2 (teal) or STAR (magenta). The ground-truth proportions were noted by crosses (X). **d**, Dotplot depicting productive and sterile transcription level recovered using sciCSR for the Stewart et al. dataset shown in **a**. Dot size corresponds to the proportion of cells with positive expression while color intensity represents expression level. **e**, Proportion of simulated reads sampled from the 5′ C, C and VDJ regions (columns), of sterile and productive IgG1 and IgA1 (rows). The differences in the relative positions of these regions in the transcripts lead to variations in the number of reads attributable to each sterile/productive transcript.

The simulation results suggest that given this prerequisite is fulfilled, the aligned reads generated by commonly used aligners in scRNA-seq data preprocessing workflows can be used to analyze sterile IgH levels using sciCSR. The sciCSR workflow extracts information about both the current state of a B cell (the IgH isotype it currently produces) and its immediate future[26], ideal as a basis to build models which describe B cell transition dynamics.

### sciCSR characterizes B cells expressing specific sterile transcript

The routine above generates count matrices just like the standard transcriptome-wide count data typically used for downstream scRNA-seq analysis, such as differential expression and visualization of expression levels. For example, Fig. 2d shows the sterile and productive IgM transcription levels in the Stewart et al. dataset[4] visualized in Fig. 2a. This captures B cells at different stages of maturation, including IgM$^+$IgD$^+$ antigen-inexperienced B cells, IgM memory cells with increased sterile transcription (most notably IgG subtypes) and the classical memory subset with productive transcription of IgG and IgA subtypes. The deconvolution into sterile and productive transcripts is useful in highlighting the diversity of CSR tendencies between B cell subsets. For example, depending on B cell subsets, up to 25–30% of cells express sterile transcripts of more than 1 isotype in human B cell scRNA-seq atlases we have analyzed (Extended Data Fig. 3). Users should, however, exercise caution for interpreting counts corresponding to productive IgH: they are fairly sparse (Extended Data Fig. 3) given their definition requires at least two reads mapped for each molecule. Analysis of the simulated data reveals that for productive transcripts, most (>70%) sequencing reads would fall in the C region, while reads mapped to the VDJ region (that is, evidence that reads correspond to bona fide productive transcripts) are much rarer (Fig. 2e). Noting this undersampling of productive transcripts, direct comparison of sterile and productive transcript counts for specific isotypes can be confounded by technical factors. For accurate quantification of productive transcripts, the isotype assignments from scBCR-seq data could be used to reliably identify and group B cells by their IgH isotypes. We noted additionally that the presence of sterile transcripts in a B cell does not necessarily indicate the productive BCR isotype of cells belonging to the same clonotype (Extended Data Fig. 4).

The sterile transcript count data can be used for investigating the underlying biology of CSR control mechanisms. We analyzed in vitro cultures of naive B cells exposed to a cocktail of anti-IgM, CD40L (mimicking BCR engagement and T cell help, respectively) and IFNγ (Fig. 3a). IFNγ is known to stimulate sterile transcription of IgG[36]; these CSR polarizing conditions should therefore lead to increased sterile transcription and CSR towards different IgG subtypes. We generated scRNA-seq and scBCR-seq data in parallel of cells sampled from the time course (day 0 before addition of stimuli, day 3 and day 6). scRNA-seq data show increase in proliferation (expression of proliferative marker *MKI67*) at day 3 and day 6 that generated distinct cell populations from day 0 (Fig. 3b,c). sciCSR confirms the induction of sterile IgG transcripts, particularly IgG3 (Fig. 3d). The scBCR-seq data suggest that a small amount (~1%) of cells were positive for IgG productive transcripts at the end of the time course (Fig. 3e), which agrees with flow cytometry analysis on the same culture experiment detecting intracellular and surface IgG proteins (Fig. 3f). To investigate the molecular profiles of cells expressing IgG sterile transcripts, sciCSR allows users to group cells by the sterile transcripts they express. This enables the application of routine analyses, including differential expression: hypothesizing that the IgG$_{sterile}$$^+$ cells would exhibit elevated IFNγ signaling, we performed differential expression analysis comparing IgG$_{sterile}$$^+$ versus IgM$^+$IgG$_{sterile}$$^-$ cells, and observed that genes involved in IFNγ signaling and response were indeed upregulated in the IgG$_{sterile}$$^+$ cells (Fig. 3g). Specifically, *IFNGR1* (encoding a component of the IFNγ receptor), *JAK2* and *STAT1* transcription were upregulated at

day 3 and day 6 (Fig. 3h), and more IgG$_{sterile}$$^+$ cells were positive for these transcripts (Fig. 3i). Coupling the capability of sciCSR in identifying sterile transcripts with more sophisticated computational tools, such as the inference of gene regulatory networks[37], will facilitate investigation into how CSR signals are coordinated to induce sterile transcription, and identify additional endogenous factors that control CSR.

### The sciCSR pipeline

We reason that, beyond simple enumeration, productive and sterile IgH transcript counts enable modeling of B cell maturation dynamics. To derive metrics for inferring cellular transition dynamics, it is advisable to leverage signals from productive and sterile transcripts of all IgH C genes collectively for robustness; ideally, metrics that summarize CSR status should also be comparable across datasets, such that they are robust to different data integration protocols commonly used to aggregate scRNA-seq datasets. In sciCSR we address this by defining 'isotype signatures' using non-negative matrix factorization[38] (NMF) over the productive and sterile transcripts of all isotypes from reference data (Fig. 4a), and using these signatures to score the CSR status of cells. These 'isotype signatures' describe the expression level of all IgH productive and sterile transcripts for naive/memory B cell states. The other output of the NMF analysis is a weighting specific for each B cell in the data, which are scores indicating its resemblance to naive/memory state based on the observed productive/sterile IgH counts (Fig. 4a). The weighting for the naive state for each cell is taken to define a metric that we term 'CSR potential'; this metric orders cells from naive to isotype-switched state (which would typically comprise both classical memory B cells and switched plasmablast/plasma cells). Visualizing the distribution of 'CSR potential' in real data from both human peripheral blood[4] and tonsils[5], we observe that this metric orders B cells correctly by their maturation stages (Fig. 4b), which can be cross-referenced with changes in productive and sterile transcription levels.

To generate CSR potentials which can be compared across different conditions, we apply NMF on reference atlases of mouse and human B cells[4–6,39] covering cell states present in germinal center (GC) and circulating B cells, to learn isotype signatures separately for each species (Extended Data Fig. 5). These pretrained signatures can be used directly to obtain the CSR potential from new, user-supplied datasets (Fig. 4c). If parallel libraries of BCR sequences (hereafter 'scBCR-seq') are available, sciCSR can also calculate the SHM level of each cell, by computing the quantity (1 − percentage identity to germline IgH V gene). Both CSR potential and SHM frequency can be interpreted effectively as pseudotime ordering of B cells defined on the basis of these processes that are biologically relevant to B cell maturation. These metrics are used as input to the CellRank algorithm[17] to impose directionality onto the cell–cell $k$-nearest neighbor (kNN) network, to derive a transition matrix that describes the probability of transition between cells (Fig. 3d). We can directly compare the inference results using CSR/SHM information to conventional CellRank analyses that uses RNA velocity to bias the kNN network.

CellRank automatically aggregates individual cells into 'macrostates' that share similar transitional behaviors, detects the start and end points of the transition pathway by examining the properties of the cell-cell transition matrix, and fits a Markov state model to describe the cellular dynamics of the system. In our application to B cells, we require a more flexible analytical framework to accommodate use cases where (1) cell states are defined not solely by transcriptome-wide features, but rather by using other criteria of interest (for example in CSR, where 'states' are defined on the basis of the productive IgH isotype of individual B cells, but not necessarily its overall transcriptomic profile), and (2) multiple possible traversal pathways are possible and expected (for example, in CSR, one could switch isotype in a stepwise manner along the IgH genomic locus, or jump directly between isotypes whilst skipping those located physically between the source and destination

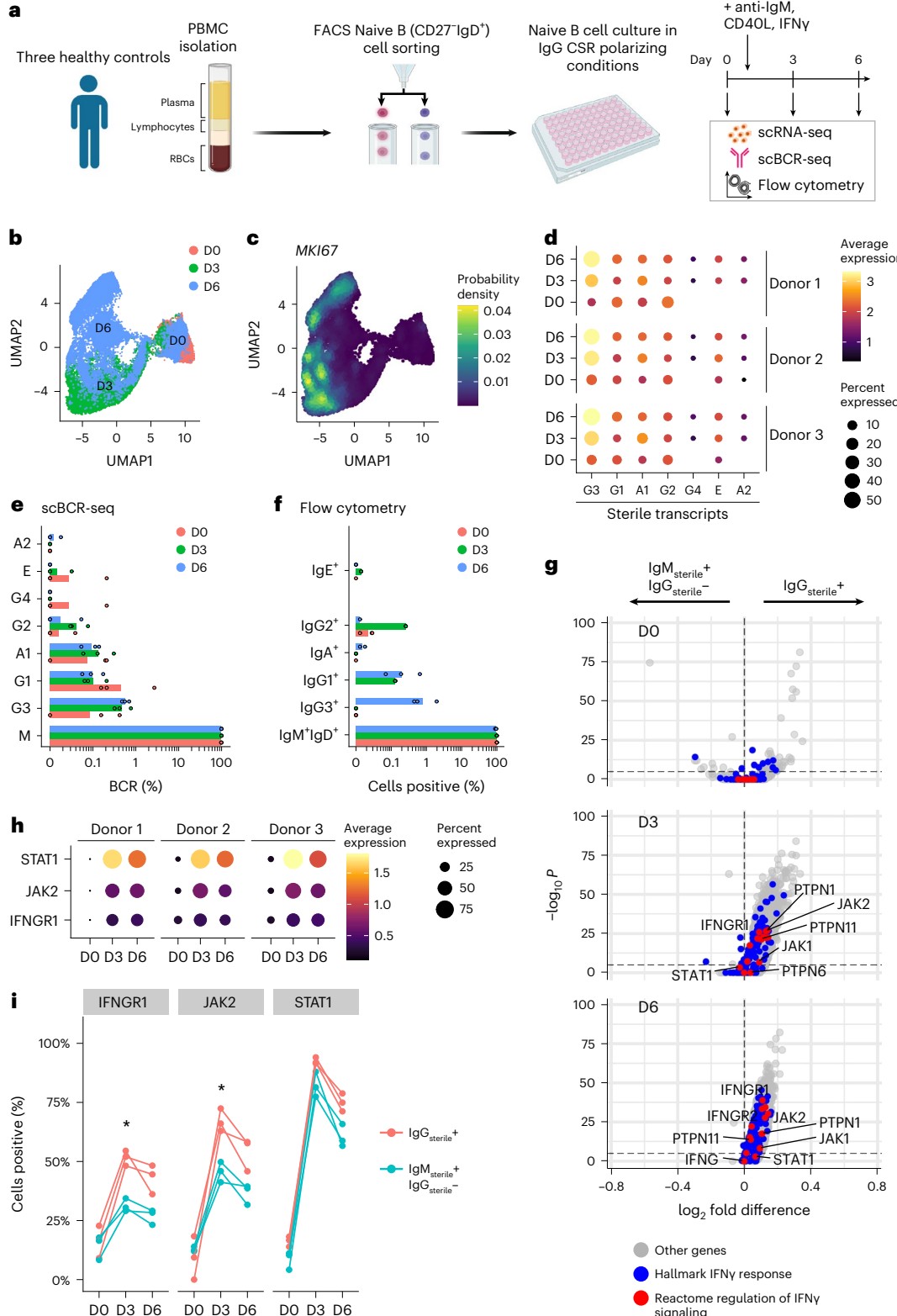

**Fig. 3 | Investigating molecular profiles of sterile transcripts expressing B cells using sciCSR. a**, Schematic of the in vitro culture experiment. RBCs, red blood cells. **b,c**, UMAP projection of scRNA-seq data of cells colored by time point (**b**) and *MKI67* expression (**c**). **d**, Level of sterile transcription of different isotypes (columns, '-S' signifies sterile transcripts) across three donors (separate panels) and time points (rows). **e**, Productive isotype distribution from scBCR-seq data. Notice different numerical scale for IgM. Individual data points correspond to measurements from cells sampled from each healthy control donor. **f**, Quantification of isotype distribution using flow cytometry. **g**, Differential expression analysis of IgG$_{sterile}^+$ versus IgM$^+$IgG$_{sterile}^-$ cells. Genes

highly expressed in the IgG$_{sterile}^+$ cells are represented with positive fold changes. Genes in the Hallmark and Reactome IFNγ pathways are highlighted in blue and red, respectively. **h**, Dotplot depicting expression of *IFNGR1*, *JAK2* and *STAT1*. **i**, Percentage IgG$_{sterile}^+$ versus IgM$^+$IgG$_{sterile}^-$ cells that are positive for *IFNGR1*, *JAK2* and *STAT1* transcripts across the three donors (separate lines) and time points (x axis). Two-sided Student's t-tests were performed comparing IgG$_{sterile}^+$ and IgM$^+$IgG$_{sterile}^-$ cells at each time point for each gene. Significant (*, Benjamini–Hochberg-corrected P value <0.05) comparisons were highlighted: *IFNGR1* at day (D) 3 (P = 0.011) and *JAK2* at D3 (P = 0.021).

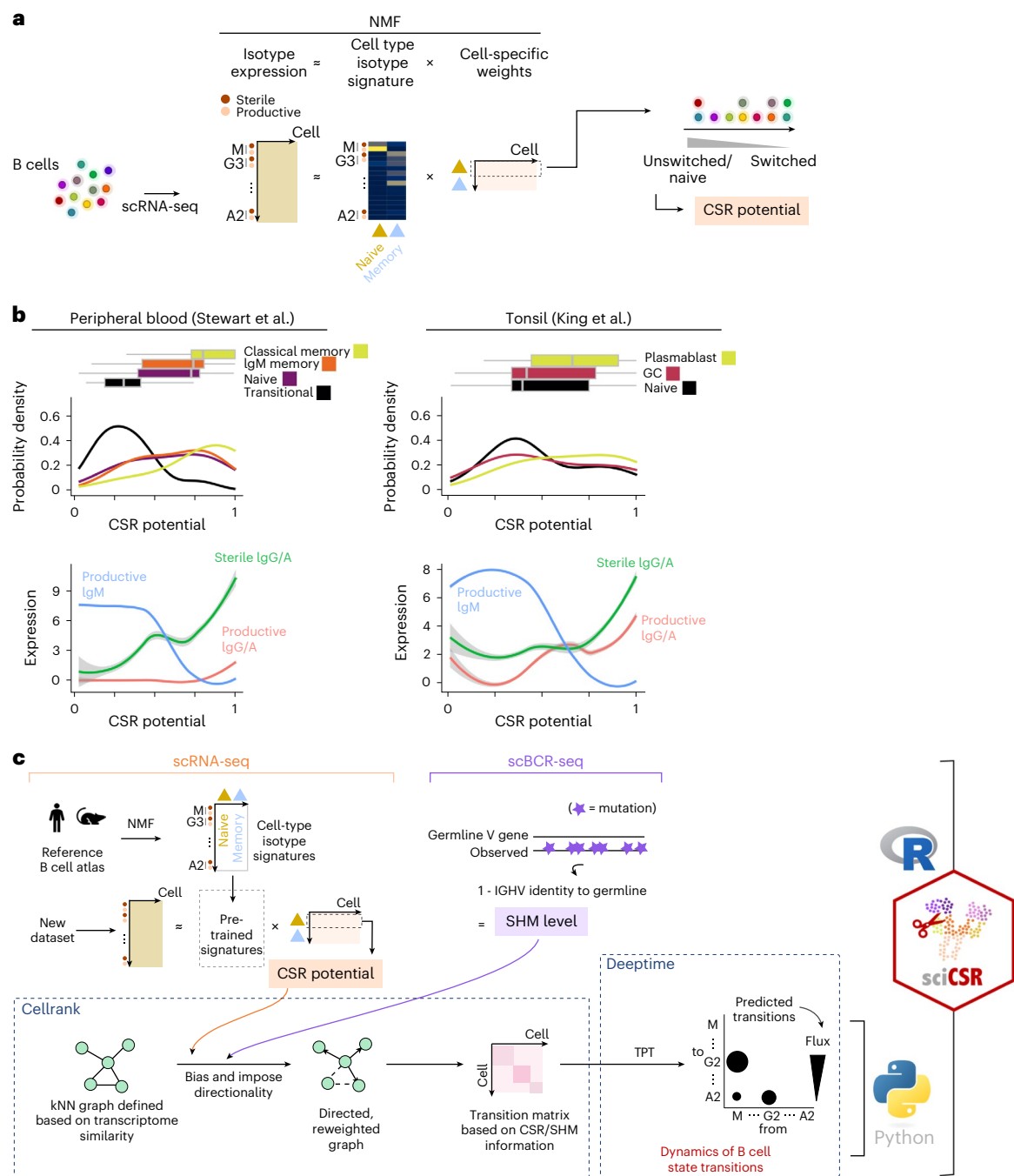

**Fig. 4 | The sciCSR pipeline to infer B cell state transitions. a,** Reference B cell atlases from mouse and human were subjected to NMF to extract isotype signatures (see Main) that describe naive/memory B cell subsets. The naive signature was used to rank the cells by their maturation status, to generate a score termed 'CSR potential'. **b,** Top: distribution of CSR potential score for B cell subsets in the Stewart et al. peripheral blood (left) and King et al. tonsil (right)

datasets. Boxplots depict distribution of the CSR potential scores, with vertical lines indicating lower quartile, median and upper quartile of the distributions. Bottom: expression level of different productive and sterile IgH transcripts along the axis of the inferred CSR potentials in the datasets. **c,** Schematic to illustrate the sciCSR workflow for inferring state transitions. For further explanation, please see Results and Methods. IGHV, immunoglobulin heavy-chain variable genes.

isotype C genes). To address these challenges, sciCSR incorporates the following innovations in analyzing cell transition dynamics based on scRNA-seq data: first, we introduce TPT to analyze the Markov model and describe the dynamics of transitions between cell groupings (Fig. 4c). TPT is used in physics and chemistry to understand how the potential energy landscape is explored in chemical reactions or in conformational changes[30], and popularized in biology to analyze biomolecular dynamics simulation data[29]. Instead of nominating a single pathway to navigate the state landscape, TPT considers the ensemble of

all possible transition paths given the transition matrix, and identifies transitions frequently implicated in navigating the states. State transitions are characterized by the amount of flux between states, and a list of possible traversal pathways are ranked by their likelihoods given the flux estimates. The inclusion of multiple traversals provides flexibility in modeling transitions more relevant to contexts such as stepwise versus direct CSR events; stepwise switching is partitioned into individual pairwise fluxes to reflect the relative abundance of each step of the modeled pathway. Second, to robustly identify unidirectional

transitions such as CSR, we reason that insignificant fluxes should have magnitudes comparable to null transition models defined by reshuffling columns of the transition matrix; sciCSR carries out random shuffling of the transition matrix and fits null TPTs, from which *P* values are calculated to evaluate the relevance of fluxes inferred from the data, to assist the interpretation of TPT fluxes and down-weigh random transitions. Finally, sciCSR groups cells either by user-defined cluster labels or by the BCR isotypes they express. This allows evaluating transitions at the cell cluster level arising from trajectories of B cell subsets in the data, or infer CSR that are directly testable by comparing the fluxes against the isotype distributions observed in BCR repertoire sequencing data (Fig. 4c). sciCSR imports functionality implemented in CellRank to fit Markov models, and allows user to use either CSR or SHM as input for estimating the transition matrix; these can be compared against CellRank models fitted using RNA velocity, to evaluate the impact of using different information to infer B cell dynamics from scRNA-seq data.

### sciCSR predicts CSR directionality in temporal scRNA-seq data

The design principle of sciCSR enables the use of a scRNA-seq 'snapshot' of B cells to infer their CSR tendencies (that is, the isotype(s) the cells are going to switch from/to) during immune response. We first tested this idea by utilizing a published dataset by Kim et al.[40] that profiled GC B cells following severe acute respiratory syndrome coronavirus 2 (SARS-CoV-2) mRNA vaccination to monitor long-term B cell maturation. Figure 5 shows data from two donors for which complete datasets with data from weeks 4, 7, 15 and 29 post first dose of vaccination are available, with the latter three time points sampling B cells from lymph nodes and profiling using scRNA-seq and scBCR-seq. We used sciCSR on week 7 scRNA-seq data of GC B cells and sought to predict the isotype distribution observed using week 15 scBCR-seq data, and similarly analyzed week 15 scRNA-seq to predict week 29 scBCR-seq. The two donors displayed differences in their BCR isotype distributions (Fig. 5b). sciCSR successfully predicted these distributions using donor-specific GC B cells with high accuracy (median cosine similarity 0.949) when comparing the TPT inward fluxes (that is, amount of CSR towards each isotype) against the isotype distribution observed in scBCR-seq data at the subsequent time point (Fig. 5c,d; Extended Data Fig. 6). The flux matrix and its associated *P* values can be visualized in a bubble plot that breaks down the inferred CSR fluxes, revealing different switching sequences (IgG1 to A1 to G2 for donor 07, and IgG1 to IgG4 accompanied by switching to IgA1 for donor 20) that can be validated with scBCR-seq data. sciCSR successfully predicts the directionality of CSR and uncovers nuances in CSR trajectories that are otherwise hidden in the scRNA-seq data space, given that these GC B cells do not appear transcriptomically distinct in the original analysis by Kim et al.[40] (Extended Data Fig. 7).

### sciCSR demonstrates CSR differences introduced by gene knockouts

We reason that sciCSR can be an ideal tool to analyze functional genomics experiments that aim to uncover gene effects on B cell maturation by introducing perturbation to the system. To investigate this in greater detail, we first collected from the literature scRNA-seq data of gene knockouts with reports of CSR effects. Figure 6 shows application of sciCSR to analyze two such datasets, on mice with two genes (*Aicda*[41] and *Il23* (ref. [42])) knocked out, either at a whole-organism level or conditionally in Cd19+ B cells (Fig. 6a and Extended Data Fig. 8). In the original reports, *Aicda* knockouts decreased both CSR and SHM according to scBCR-seq data[41], while *Il23* knockouts biased the cells away from IgG2b, measured via enzyme-linked immunosorbent assay and immunofluorescence experiments[42]. Applying sciCSR on the knockout and wild-type (WT) mice scRNA-seq data, we recapitulated the observed isotype distribution in the scBCR-seq data reported in these studies

(Fig. 6b,c). Since the TPT workflow implemented in sciCSR accepts any one of RNA velocity, CSR or SHM as inputs, we asked whether these methods capture different information in the inference of cell state transitions. We considered the Hong et al. *Il23* knockout dataset[42], which sampled murine splenic B cells covering both GC and memory subsets (Fig. 6d), and inferred the cell–cell transition matrices using RNA velocity (using the scVelo package[10] following default settings, see section '*Il23* gene knockout mice study from Hong et al.' in Methods), CSR and SHM. This allowed us to simulate transition paths given the three Markov chains, and compare the frequencies these simulated paths visited each state in the data (Fig. 6d,e). We observed that RNA velocity consistently provided different inference than to CSR and SHM: the latter two methods faithfully reproduced the cell type distributions in the data, while RNA velocity appeared to bias towards specific states (Fig. 6e). We mapped from these Markov chains the frequency of sampling each state onto the uniform manifold approximation and projection (UMAP) visualization; these indicate that, compared to RNA velocity, CSR and SHM allow for a more extensive sampling of cell states in the data (Fig. 6f). These results suggest that sciCSR captures CSR information that is consistent with evidence from SHM in inferring transitions that can be missed by RNA velocity analysis. Both CSR and SHM represent B-cell-specific evidence to infer state transitions relevant to the system. Given that RNA velocity methods are known to give noisy inference in mature immune cell types[14,18], sciCSR could serve as a viable alternative by harnessing CSR and SHM to analyze B cell state transitions.

## Discussion

A lack of tools to study immune cell state transitions in mature cell types precludes accurate computational analyses of these processes from scRNA-seq datasets of important biological phenomena such as immune response to vaccines, pathogens or malignant cells. sciCSR addresses this gap by implementing a novel approach for studying B cell state transitions. The uniqueness of our approach focuses on extracting B cell-specific information (expression of sterile/productive IgH transcripts, SHM level) from scRNA-seq/scBCR-seq datasets, while benefiting from CellRank[17] to provide underlying implementations that infer cellular trajectories. We introduce TPT[43] to analyze CellRank-inferred transition matrices. Inspired by analysis of MD simulation data, TPT and the generation of null transitions matrices (offer a principled way to assess the robustness of the inferred transitions. While the robustness of sciCSR predictions is dependent on the amount of CSR activity implicated in the dataset (Supplementary Note 1), sciCSR is robust in recovering transitions between small populations (Supplementary Note 2), making it ideal to study transitions in rare B cell populations (for example, antigen-specific B cells from peripheral blood).

We believe a standout feature of sciCSR is in utilizing cell-type-specific biological features to inform trajectory inference: here, most commonly used tools are agnostic to the biological system of study. CellRank by default uses RNA velocity to build cell-to-cell transition matrices; here we derived a custom cell ordering based on B cell biology and used this as input to CellRank to infer transitions. Tools that utilize system-specific biological knowledge to study cellular dynamics have recently received more attention, owing to their potential in overcoming the limitations of general trajectory inference tools. Some tools necessitate collection of orthogonal data types, for example, barcode-based lineage tracing of the same cell population sampled using scRNA-seq (PhyloVelo[44]). Here, while B cells represent an attractive system to apply such methodologies as phylogenies can be readily reconstructed from scBCR-seq V(D)J sequences, the limited number of B cells represented in many currently available scRNA-seq libraries implies restricted sampling of these lineages, posing challenges to the subsequent trajectory inference step. One approach (Pseudocell Tracer[45]) considered the BCR isotype expression profiles and trained deep-learning-based generative models to overcome the sparsity of

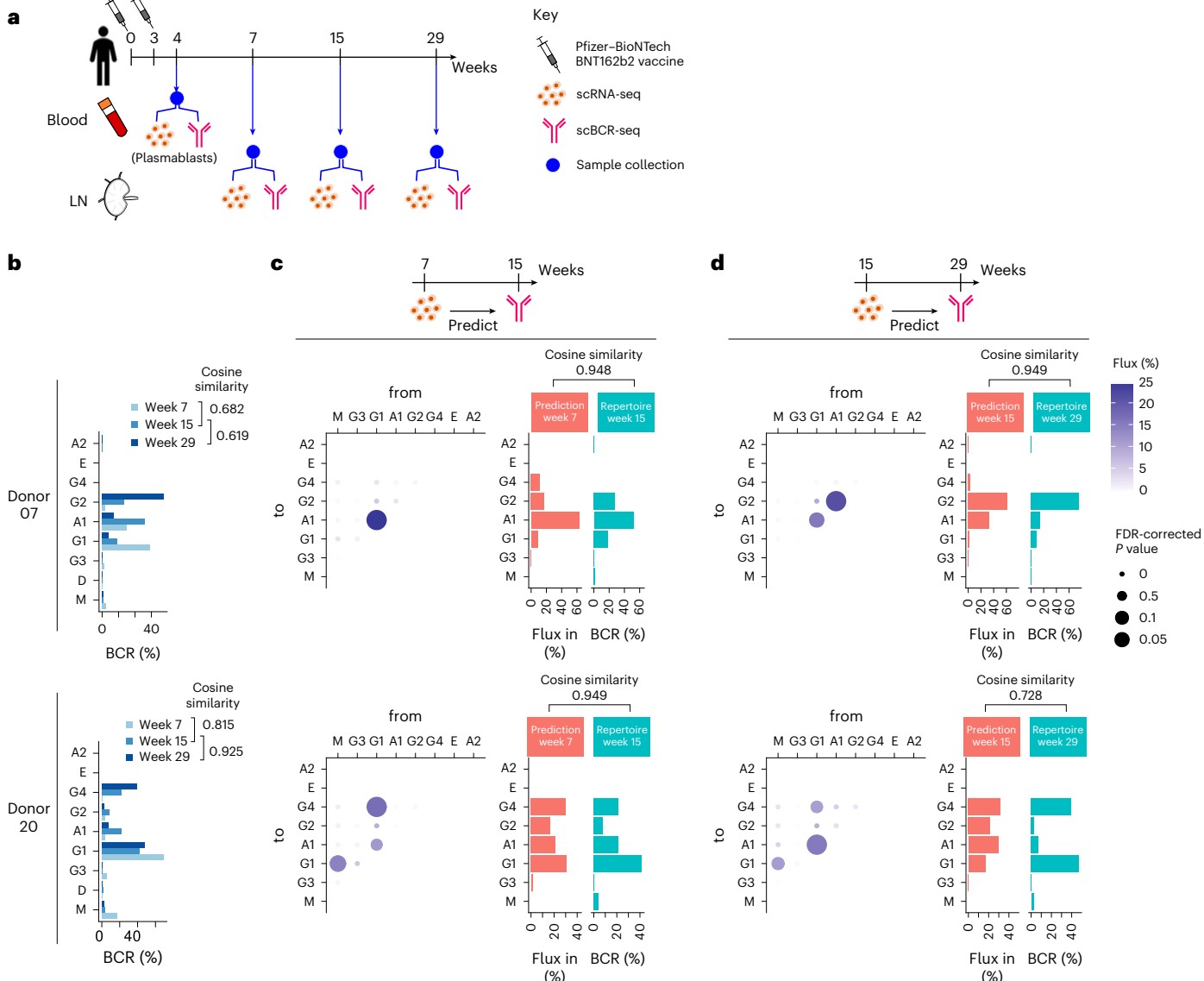

**Fig. 5 | sciCSR predicts BCR isotype distribution in time-course immunization data. a**, Schematic of immunization and follow-up data collection from the Kim et al. dataset. LN, lymph node. **b**, Isotype distribution of GC B cells for donors 7 (top) and 20 (bottom), across weeks 7, 15 and 29 scBCR-seq data. **c,d**, sciCSR-predicted CSR fluxes, based on TPT, from and to each isotype. Predictions were made on the basis of week 7 scRNA-seq data to predict scBCR-seq isotype distribution at week 15 (**c**), or using week 15 scRNA-seq data to predict week 29 scBCR-seq isotypes (**d**). Left: CSR fluxes expressed as a bubble plot. The colors of bubbles correspond to the amount of flux between the given pair of isotypes predicted by sciCSR, and their sizes correspond to statistical significance of these fluxes compared to predictions generated from randomized transitions (see Main and Methods). Right: the sciCSR-predicted switching events to each isotype, represented by total inward TPT fluxes towards each isotype, are compared against the observed BCR isotype distribution at the subsequent time point. These two distributions were compared using cosine similarity.

cell states in the data. These generative models were then used to map out the inherent CSR trajectory in the latent space. This represents a viable approach to model CSR in scRNA-seq data; however, it lacks a publicly available implementation. Here we build upon CellRank[17] and interface with well-supported scRNA-seq data analysis packages in R and Python (Seurat[46], scanpy[47] and so on) to provide a method that also incorporates CSR and SHM signals from the data. We have evaluated the performance of sciCSR in a range of scRNA-seq datasets generated under common model systems used in immunology such as vaccination and mouse knockout models. With these datasets we show that sciCSR offers accurate predictions of CSR that can be verified by measuring the BCR isotype expression subsequent to perturbations.

The major innovative feature in sciCSR is the enumeration of productive and sterile IgH transcription in single cells, which can be applied in any human or mouse B cell scRNA-seq dataset. While analysis of sterile transcription was previously attempted in some B cell scRNA-seq studies, they were performed at the stage of raw data preprocessing[5,48], without the relevant code made available in the associated publications. sciCSR allows this analysis to be performed on an ad-hoc, a posteriori basis as for other common scRNA-seq data analysis tasks. Counts for productive and sterile IgH transcripts are sparse, especially for the productive transcripts; this can be mediated with parallel scBCR-seq libraries obtained for the same cells, which are increasingly popular in B cell studies relating to their antigen-binding features[5,40].

A recent analysis by Horton et al.[26] uses lineage tracing to study the establishment of B cell fates, and found that the CSR fate of a B cell is largely independent from its predecessors; CSR fate is instead

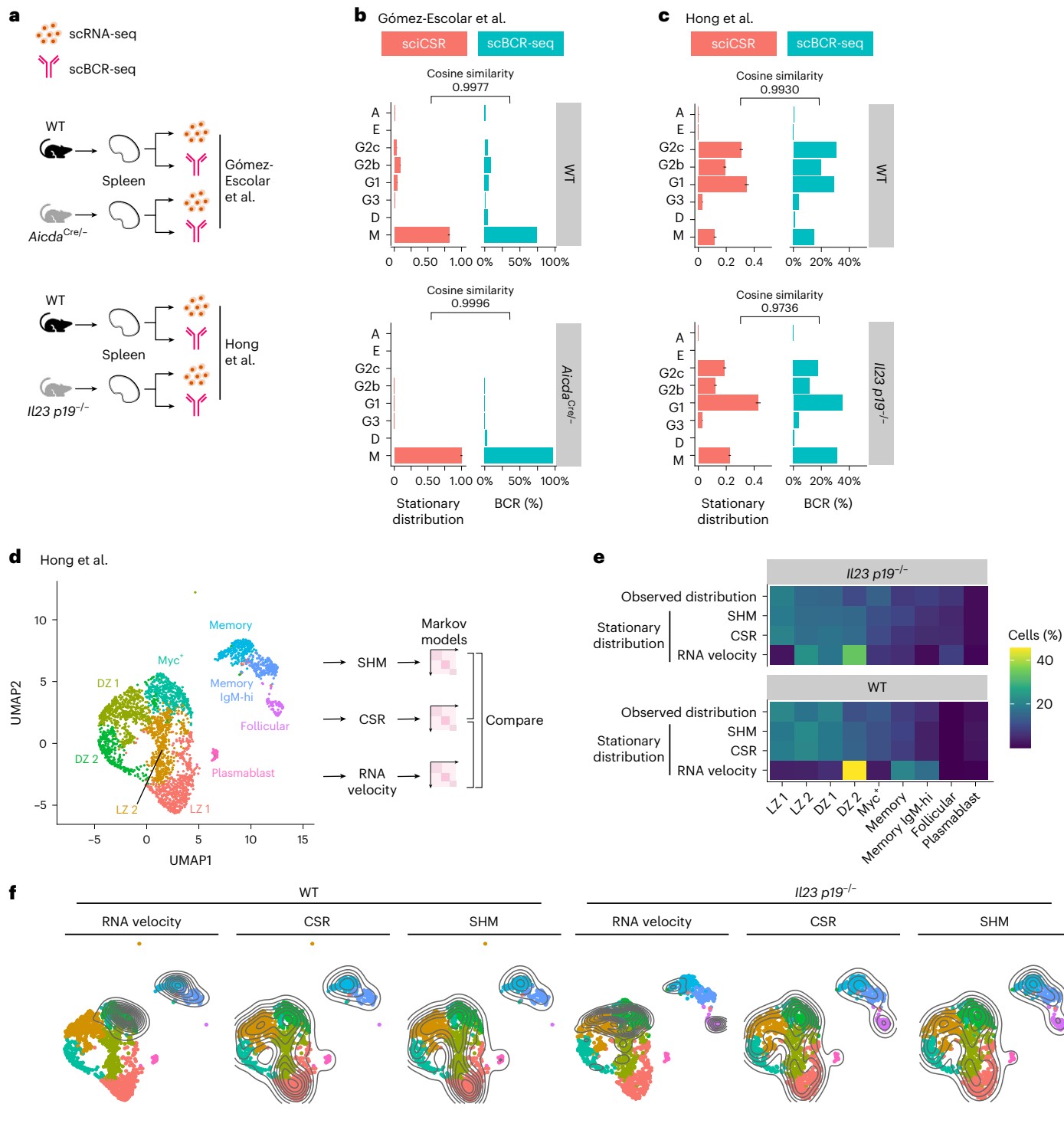

**Fig. 6 | sciCSR recapitulates CSR effects in gene knockout scRNA-seq studies.**
**a**, Schematic of scRNA-seq/scBCR-seq data from mouse gene knockout studies
considered in this paper. Notations of knockout genotypes follow the respective
publications. **b,c**, Isotype distribution in the Gómez-Escolar et al.[41] (*Aicda*
knockout, **b**) and Hong et al. (*Il23 p19* knockout, **c**) data. Stationary distributions
predicted by sciCSR compared against scBCR-seq isotype distributions. sciCSR
were applied separately for WT and knockout cells. Error bars represent 95%
confidence intervals obtained by bootstrapped sampling of cells for each
isotype. **d**, Projection of the Hong et al. data using UMAP. Transition matrices
were constructed using CSR, SHM and RNA velocity (calculated using scVelo)
information. The resultant Markov chains derived from each transition matrix
were used to sample trajectories of cell state transitions. DZ, dark zone; LZ, light
zone. **e**, Heatmap depicting the observed distribution of cell types in the Hong et
al. dataset and the stationary distributions inferred using RNA velocity, CSR and
SHM information using scRNA-seq data. **f**, Projection of the sampled trajectories
based on RNA velocity, CSR and SHM on the Hong et al. UMAP visualization.
Contour maps depict the 2D kernel density estimates of the frequency the cells
are sampled in the Markov chains based on the three types of information.

determined by a combination of AID expression and sterile transcription[26], as previously noted by others[20,22,27]. In contrast with the Horton et al. approach to predict CSR behaviors of individual B cell lineages, here in sciCSR we model the switching dynamics of a heterogeneous group of B cell lineages.

In the future, sampling a large number of B cell lineages using single-cell genomics will allow a more detailed model of CSR that permits prediction of switching behavior from scRNA-seq data. We believe that sciCSR will open the door to further explore the basic mechanisms of CSR: the enumeration protocol in sciCSR can support the use of scRNA-seq to study the variegation of CSR fates, and motivate mathematical modeling approaches to quantify CSR likelihoods beyond the analysis presented in Horton et al.[26]. Coupling with experimental approaches to study single-cell epigenetic landscape of B cells[49] and computational approaches to infer gene regulatory activities[37], sciCSR could contribute towards understanding how sterile transcription of different isotypes are differentially regulated, how this reconciles with their observed baseline expression levels, and how different molecular stimuli could modulate sterile transcription and ultimately fine-tune CSR in vitro and in vivo.

sciCSR is a freely available R package providing implementations to build inference of CSR and B cell maturation using scRNA-seq data. The enumeration of productive and sterile IgH transcripts does have its technological limitations: it necessitates usage of the 5′-biased read sampling protocol that would currently preclude datasets generated using the more common 3′ protocol and the use of spatial transcriptomics data. More detailed analyses in the future (for example, identification of latent features of CSR states that are also detectable using 3′ data) will extend the utility of sciCSR to different technological platforms. We believe sciCSR offers a starting point to model B cell maturation in scRNA-seq data; these models can be further analyzed to understand the molecular cues of CSR and different steps of the maturation process, their regulation in situ within tissues, and their dysregulation in diseases.

## Online content

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

## Methods

### The sciCSR algorithm

sciCSR is a R package to annotate productive and sterile IgH transcripts given scRNA-seq aligned reads, and process scRNA-seq (and scBCR-seq if available) data to infer state transitions using CSR and SHM information. The inputs to sciCSR are (1) BAM files of the processed and aligned scRNA-seq reads; (2) processed and normalized cell-by-gene count matrix corresponding to the BAM files, and (3) optionally, associated scBCR-seq data for the same set of cells. If (1) and (2) are available, sciCSR will extract CSR information and use this to infer cell state transition. If (3) is also provided, sciCSR will extract both CSR and SHM information and users can opt to use either in the inference. For notational convention let $N$ be the number of cells and $G$ be the number of genes transcriptome-wide recorded for a given scRNA-seq dataset. sciCSR aims to:

(i) Enumerate productive and sterile IgH transcripts, as defined in the main text and illustrated in Fig. 1c, for each cell. These results produce a separate cell-by-transcript count matrix where for each IgH isotype, two transcripts will be listed, one corresponding to the productive transcript of the given isotype, and another corresponding to sterile transcripts. For a species with $S$ isotypes, this will yield a $2S \times N$ matrix which we will refer to as 'isotype matrix' in the following text.

(ii) Use the isotype matrix or, if scBCR-seq data are available, the SHM level as input to generate a cell ordering, as a 'pseudo-time' input to CellRank to infer a cell-by-cell transition matrix. CellRank[17] is used in accordance with author-recommended protocols, that is, the final cell transition matrix combines the matrix inferred using the kNN network that describes transcriptomic similarity between cells, and the matrix inferred using CSR/SHM information. CSR/SHM information is used as bias to impose directionality on the kNN-based transition matrix.

(iii) Infer the fluxes between either cell clusters or isotypes by applying TPT on the transition matrix.

**Mapping positions of sterile transcription at the IgH genomic loci.** The problem of identifying sterile IgH transcripts given scRNA-seq reads mapped to the reference genome is nontrivial, since sterile transcripts are not annotated in standard genome references such as those provided by Ensembl or UCSC genome browser. The distinction between productive and sterile IgH transcripts lies in the TSS: productive IgH transcripts are transcribed beginning at positions 5′ to the Ig variable (V) gene in the leader region, while sterile transcripts begin at some position 5′ to the first constant region (C) exon[50–54]. We first clarified the exact genomic position of TSS for the sterile transcript for every human and mouse IgH C gene with the exception of IgD where sterile transcription is not well understood. We collected from the literature a set of DNA genomic sequences isolated using traditional molecular cloning and restriction enzyme analysis, of the promoter region of sterile transcripts observed in experimental and/or clinical models of CSR[51–53,55–61]. These sequences are mapped to the hg38 (for human)/mm10 (for mouse) reference genomes using BLAT[62] (v.36×4). Supplementary Table 1 lists the positions of sterile transcript TSS; this table is stored in the sciCSR package to guide the extraction of aligned scRNA-seq reads for enumerating productive and sterile transcripts (see below).

**Re-analyzing scRNA-seq read alignments to distinguish productive and sterile IgH transcripts.** The first step of sciCSR reads in user-supplied BAM files where aligned reads are listed with additional fields noting their associated cell (with tag 'CB') and molecular (tag 'UB') nucleotide barcode; these tags that are used in 10x cellranger-generated output BAM files are set as default. The BAM file is scanned three times, each time extracting reads mapped to a different genomic location:

(1) the exonic regions encoding any IgH V, diversity (D) or joining (J) gene, (2) the exonic regions encoding any IgH C gene, and (3) the 5′ region starting from the sterile transcript TSS to the site immediately preceding the start site of the first C exon (hereafter '5′ C'). In each scan sciCSR extracts the aligned genomic positions for reads fulfilling each criterion listed above. Spliced reads that merely span across a given genomic range without actual matches to the genomic region of interest, as well as those without cell and/or molecular barcodes, are removed in each scan. We next implemented the following scheme to summarize these three lists of aligned reads to discern, for each molecule with a unique combination of cell and molecular barcodes, whether they represent productive or sterile IgH transcripts (Fig. 1c):

(i) A 'productive' transcript must have at least two reads, one mapping to the VDJ genes and another to the C exonic regions;

(ii) A 'sterile' transcript must have at least one read mapping to the 5′ C region, and optionally to the C exonic regions, but no read mapping to the VDJ genes, and;

(iii) Transcripts where insufficient information exists to be classified as productive or sterile are labeled 'uninformative'.

The number of cell–molecule barcode combinations were enumerated for each productive/sterile/uninformative IgH transcript type. Discounting the uninformative transcripts, this gives, for each cell, a gene count vector of length $2S$ where $S$ is the total number of IgH C genes for the given species. Concatenating these vectors column-wise yield a $2S \times N$ matrix where $N$ is the total number of cells. We term this matrix the 'isotype matrix'.

**Validating workflow to identify productive and sterile IgH transcripts.** We validated the enumeration of sterile and productive transcripts by utilizing a simulated dataset where we sampled sequencing reads from the human IgH genomic locus. Comparison between the ground-truth distribution of reads we sampled and the isotype distribution reconstituted after aligning these reads to the reference genome would indicate whether the workflow can distinguish between productive and sterile IgH transcripts. To generate this simulated dataset we concatenated the DNA sequence for the VDJ region of the antibody CR3022 (GenBank accession DQ168569)[63] with the exonic regions of each human IgH C gene (GRCh38 reference genome from Ensembl); this set of sequences would be the full-length productive transcripts. For sterile transcripts, the genomic sequences (genome build hg38) of each human isotype between the sterile transcript TSS and the end of the C gene coding region was retrieved from the UCSC genome browser application programming interface. All productive/sterile transcript sequence constructs are included in Supplementary Dataset 1. We next designed scenarios where cells express either one specific sterile/productive transcript, or a combination of sterile/productive transcripts of different isotypes at given proportions; all such scenarios are shown in Fig. 2b,c. The R package polyester[64] (v1.29.1.1) was used to sample reads based on these ground-truth sampling proportions, using the function 'simulate_experiment_countmat'. To mimic experimental data where reads are biased towards either the 3′ or the 5′ ends of transcripts, we included the positional bias model from Li and Jiang[65] as an argument for polyester to generate sets of reads biased to the 3′ end, and the mirror image of the Li and Jiang model as input to generate 5′-biased reads. All other parameters to run polyester were unchanged. A total of 1 million reads were sampled for each designed scenario. The sampled reads were aligned to the chromosome 14 genomic sequence from the GRCh38 reference genome, using either HISAT2 (ref. 35) (v2.2.1) or STAR[34] (v2.5.1.b) (for further details, see Supplementary Methods, heading 'Analysis of simulated productive and sterile IgH transcripts').

**Deriving CSR potential.** For CSR, sciCSR makes use of the productive/sterile isotype matrix to derive a score that ranks cells from naive to memory/terminally differentiated. The intuition is that the isotype

matrix can be decomposed into 'signatures' of isotype expression that are representative of naïve/memory cell types; these signatures can then be used to deconvolute the isotype matrix and score the resemblance of each cell to naive or memory state, in terms of their IgH transcription profile. To guide the definition of these isotype signatures we first applied the sciCSR productive/sterile transcript quantification to scRNA-seq data of B cells at reference B cell states; these reference B cell atlases were compiled for human and mouse (see heading 'Reference atlases' under 'Datasets' section). These datasets were chosen as reference B cell atlases to cover a wide range of B cell states found in circulation and in secondary lymphoid organs, and, wherever possible, sampled from healthy subjects. We next applied NMF[38] separately on the human and mouse data, to decompose the isotype matrix $T$ (of dimension $2S$ (total number of isotypes) $\times N$ (number of cells)) into two matrices:

$$T \approx P \times C$$

The matrix $P$ is of dimension $2S \times H$, where $H$ is the number of isotype signatures. $C$ is a matrix of dimension $H \times N$. Each isotype signature consists of weights indicating the importance of the given productive/sterile transcript for a cell type of interest, while an entry of the $C$ matrix $c_{ij}$ would refer to the contribution of isotype signature $i$ in cell $j$. Note the approximation sign in the equation; applying NMF is to jointly estimate $P$ and $C$, which together approximates best the input matrix $T$. Our goal here is to obtain one signature (that is, column) $v$ from the matrix $P$, which either biases towards memory B cells and/or terminally differentiated cells in the lineage such as plasma cells (that is, $c_{vj}$ increases as cells go from naive to memory), or one that is biased to the sterile/productive transcripts of IgM (that is, the developmental trajectory would go inversely of this signature) to rank the cells. In principle we could obtain signatures specific to IgG/A/E using this method; here the IgM-biased signature was deemed preferable as it could be applied to different biological contexts where any switching events away from IgM were of interest.

Separate NMF runs were performed on the human and mouse data to derive a two-signature matrix for each, with one IgM-biased signature signifying naive B cells and another signifying memory cells (mainly IgG, see Extended Data Fig. 6 and Supplementary Methods heading 'NMF analysis to derive isotype signatures'). These $P$ matrices are stored to score cells from a new, user-supplied, dataset $T'$, using non-negative least square (NNLS) regression, that is, estimating $T' \approx P \times C'$ using $P$ trained on the reference dataset $T$. The entries $c'_{vj}$ in the $C'$ matrix corresponds to the contribution of the naive (that is, IgM-biased) signature $v$, trained on the reference atlas, in cell $j$ in the user-supplied data. $c'_{vj}$ for all $j$ were taken to derive the CSR potential by scaling into the range [0,1] and reversing the scale such that switched cells (that is, less naive-like) have a higher CSR potential:

$$c''_{vj} = \frac{c'_{vj} - \min(c'_{vj})}{\max(c'_{vj}) - \min(c'_{vj})}, \; c''_{vj} \in [0,1]$$

$$\text{CSR potential} = 1 - c''_{vj}$$

NMF was performed using the R NMF package (v0.24.0) using the 'nmf' function, applying the alternating least square approach from Kim and Park[66] (argument 'methods = 'snmf/r''). The 'nmf' function attempted ten NMF fits, and the best (that is, the smallest approximation error) fit was taken as the final signature matrix to be stored and used to score user-supplied datasets. For the human atlas, all cells were considered and the 'nmf' function was applied once; for the mouse atlas, to reduce memory usage, the 'nmf' function was called ten times, each time using a different, randomly selected set of 20,000 cells from the atlas to fit ten NMF. Signature matrices from the best NMF fit from each of the ten subsetted datasets were summarized as element-wise mean values to derive the final signature matrix for storage and decomposition of user-supplied productive/sterile transcript counts. NNLS was performed using the R package nnls (v1.4).

**Deriving SHM level.** If scBCR-seq data have been integrated with scRNA-seq results, sciCSR calculates the SHM level of each cell as follows:

$$\text{SHM} = 1 - \text{nucleotide identity to assigned germline V gene } (\%)$$

This value is subsequently scaled into range [0,1] similar to the derivation of the CSR potential.

**Applying CellRank.** The CSR and SHM ranking of cells were taken as 'pseudotime' orderings, to generate cell–cell transition matrix using the CellRank[17] (v1.5.1) 'PseudotimeKernel'. We followed the CellRank recommended protocols and generated a final transition matrix $M$ for each dataset by combining transition matrices using two different sources of information:

$$M = 0.8 \times M_{\text{kNN}} + 0.2 \times M_{\text{Pseudotime}}$$

where $M_{\text{kNN}}$ is the transition matrix ('ConnectivityKernel' in CellRank) derived by considering the kNN network calculated using scanpy (v1.9.1), and $M_{\text{Pseudotime}}$ is the transition matrix derived from using the CSR/SHM information as pseudotime ordering of cells. The respective weights of $M_{\text{kNN}}$ and $M_{\text{Pseudotime}}$ were defaults for combining kernels in CellRank[17].

**TPT.** TPT is a statistical framework to analyze the properties of transition paths governed by a discrete-state Markov chain[30,43,67]. The major premise of TPT is, since transition paths between states can adopt a variety of behaviors (for example, to reach state B from A, this can be accomplished via a direct jump, or a more convoluted path involving some other intermediate state(s), or simply fail to reach B and return back to A), it is important to describe the statistical properties of an ensemble of transition paths, from a source state to a target state (both defined by the user), to capture more truthfully the transition dynamics in the system[43,67]. This is in contrast to Markov chain analyses that seeks to extract a maximum likelihood solution to fit the observed data. TPT is a well-established technique in physics and chemistry to study dynamics of chemical reactions and conformational changes of molecules[30,43,68]; the latter has been adopted in computational biology in the analysis of MD simulation trajectories[29,69,70]. Here, in the context of CSR, TPT seeks to analyze the Markov chain, and extract specific switching events that are prominent in the model: considering the Markov chain, TPT reports which states (isotypes), and therefore which transition paths (isotype switches), are more likely to be observed if a cell consistent with the input data is to begin CSR. Supplementary Note 3 contains further discussion on the motivation of TPT.

We used the Python package deeptime[71] (v0.4.2) (which powered the backend calculations of TPT in the MD analysis package PyEMMA2 (ref. [29])) for TPT calculations. deeptime implements TPT in a Python class 'ReactiveFlux' to fit TPT on the CellRank-derived transition matrix $M$, such that it describes the transition dynamics of the cells from a source to a target state indicated by the user. These source/target states can either be cell states defined by the cell cluster labels users typically give to scRNA-seq datasets, or BCR isotypes specifically for the analysis of CSR dynamics, or any arbitrary grouping defined by the user. Since the transition matrix $M$ from CellRank has dimension $N \times N$ where $N$ is the total number of cells, a grouping ('coarse-graining' in the deeptime package) step is required to group fluxes such that they can be interpreted between clusters of cells. We fit TPT on the $N \times N$ transition matrix giving vectors of cell indices corresponding to the source

and the target state, respectively, and then use the ReactiveFlux.coarse_grain() function in deeptime to coarse-grain the fluxes at the cell cluster level. These 'clusters' are supplied by the user; they can either be cell identities defined by clustering the transcriptomic data, or BCR isotypes.

The ReactiveFlux object computes various quantities typical of Markov chain and TPT analysis, but we deem the following the most informative about cell state transitions:

(1) Gross flux $F$, a $n \times n$ matrix where $n$ is the total number of cell states. Each element $f_{ij}$ describes the amount of flux that flows from state $i$ to state $j$. Recall that TPT analyzes all possible path of traversing between all states, and reports transitions between specific pairs of states which are more likely to be sampled in these traversal paths. Fluxes break down these paths into individual jumps between states, and quantify the relative abundance of these jumps in the ensemble of paths. Importantly, TPT provides flux estimates for any pair of states, not just limited to direct fluxes between the source and target states. We adopted the recommendations in PyEMMA2 (ref. 29) to interpret these as relative fluxes, that is, instead of the absolute flux values these are normalized as $f_{ij} = \frac{f_{ij}}{\sum_{i,j} f_{ij}}$, which describe the proportion of total flux in the system that flows from state $i$ to $j$. These fluxes are the indicators of transitions and can be visualized with, for example, projection of arrows on dimensionality reduction plots, typical in scRNA-seq transition dynamics analyses such as RNA velocity[9,10].

(2) Stationary distribution $\pi$. This is a vector of length $n$, where each element is a probability describing the likelihood to reach a given state when the system reaches equilibrium over a long period of time. Intuitively, the stationary distribution is informative in the comparison of transition dynamics observed in perturbation experiments where the perturbation (for example, a gene knockout) yields a discernible effect to the likelihood to sample certain states. All elements in $\pi$ sum to 1, that is, $\sum_n \pi = 1$.

(3) Mean first passage time (MFPT) between the source and target states. This refers to the amount of time (in arbitrary unit) needed to reach the target after leaving the source state, as a measure of the efficiency of transition given the data.

(4) A list of transition pathways with intermediate states from the source to the target, with probabilities of their traversal to reach the target from the source.

These results are provided when users execute the fit_TPT() function in the sciCSR R package together with analogous null/bootstrapped estimates to assess statistical robustness (see below).

**Interpreting TPT.** TPT fluxes are descriptions of transitions between states estimated by considering the frequencies each state is sampled in the data and the transition behaviors of each cell as specified in the input transition matrix. Since the goal of TPT is to evaluate possible pathways to traverse the state landscape to reach a user-specified target state from a source state, the choice of source and target states influence the TPT inference results (Extended Data Fig. 7). Specifically, the magnitude of flux tends to be higher for fluxes involving either the chosen source or target states, since the algorithm is designed to consider pathways that necessarily pass through these states as required by user definition. It is therefore important to choose appropriate states with the aim of describing the fluxes and stationary distributions of all states in the system. Here we chose source and target states which were (1) sampled in the data and (2) as close to the ultimate start/end points of the relevant biological process as possible. For example, for analyzing CSR we would want to set the source state as IgM and the target state as IgA2 (human)/IgA (mouse) given these isotypes were

observed in the data; transition paths involving any possible pairs of isotypes in the middle of the locus would then be traversed, some of which would be of interest to the analysis. For analyzing transitions between cell clusters, we chose the cell cluster representing naive B cells (if available) as the source state, and the plasma cell/plasmablast cluster as the target state. We reasoned that in most use cases users would have an intuition of the observed states and be able to choose source and target states using prior knowledge.

**Pruning TPT results.** While quantities listed in the section 'TPT' are typically analyzed in usage of TPT in biological settings (for example, in MD analysis[29,70]), we reasoned that additional measures were necessary to prune the TPT results and give uncertainty estimates to fluxes and stationary distribution quantities to aid interpretation. In applying TPT to infer CSR, backward fluxes (that is, flow from a given isotype to another that is 5′ to itself) are excluded as these transitions are physically prevented via genomic DNA excision during CSR. To evaluate the statistical significance of fluxes, we estimated a $P$ value for each entry in the gross flux matrix $F$ by one-way comparison of whether the observed flux is greater than those obtained by randomly reshuffling the input transition matrix to TPT. We reshuffled columns of the transition matrix for $t$ times (by default $t = 100$; users can change this number as appropriate) and fit $t$ TPT models. The randomized fluxes were modeled as a Gaussian distribution and a one-way $z$-test was performed to derive $P$ values for each element of the gross flux matrix. The resultant $P$ values, after multiple-testing correction implemented using the false discovery rate (FDR) method with the p.adjust function in R, give confidence on how likely the observed fluxes are due to the structure of the transition matrix rather than merely the distribution of cell types/isotypes in the system. For the stationary distribution, we performed bootstrap sampling of cells for each cluster label and summed over their individual stationary distributions, to obtain cluster-wise bootstrapped samples of stationary distributions.

**Visualizing cell state transition inference results.** sciCSR offers the following visualization tools to present the TPT results: first, sciCSR provides function to depict the stationary distribution of states as a bar plot, with the 95% bootstrapped confidence intervals as error-bars. Second, for fluxes, users can project the inferred transitions onto either a grid of arrows or velocity streams, akin to RNA velocity plots generated using velocyto or scVelo. sciCSR uses the scVelo plotting facility in Python in the backend and outputs the figure in R console. This visualization is applicable for TPT inference where the cluster labels are used to group the cells. For analyses where cells groupings are not distinct on the dimensionality reduced space, users can also visualize fluxes as a bubble plot where each bubble corresponds to transitions between any pair of states; the bubble color corresponds to the amount of flux, and bubble size indicates statistical significance (given by −log(adjusted $P$ value)). This visualization is applicable to CSR-specific analyses grouping cells by BCR isotypes.

**Comparing transitions inferred using different biological information.** Since sciCSR allows users to supply either CSR or SHM pseudotime orderings to infer cell transitions, and the underlying CellRank algorithm supports transition inference using RNA velocity information[17], we reason that the CellRank transition matrices derived using these difference sources of information can be directly compared. To do so, we set up separate Markov chains using the transition matrices, sample random walks from these Markov chains, and compared the frequencies of visiting each cell state in the sampled walks to ascertain whether these different transition matrices capture similar information regarding cell state transitions. For each transition matrix, a 'markovchain' object (using the R package markovchain[72], v0.9.0) was created and a total of 1,000 walks (users can change this default parameter) were sampled using the 'markovchainSequence' function. The resultant

frequency matrices which record the number of times each state is visited in the random walks are compared using either Kullback–Leibler divergence or Jensen–Shannon divergence, both calculated with default parameters using the R philentropy[73] package (v0.5.0).

**Implementation.** The sciCSR pipeline has been implemented as a R package. scRNA-seq data are handled using data objects defined in the Seurat (v4.2.2) R package[46]; sciCSR directly extracts and modifies data stored in the Seurat data objects, including inserting the isotype matrix as a separate 'assay' in the Seurat object, and adding and modifying columns of the metadata. The inferences of state transitions using CellRank[17] (v1.5.1) and TPT are implemented in Python (v3.9.12), which sciCSR directly deploys under the R reticulate environment; sciCSR calls the R package SeuratDisk (v0.0.0.9020)[74] to convert between R Seurat object and Python AnnData files as input to CellRank. TPT is fitted to the CellRank transition matrix using the python package deeptime[71] (v0.4.2). A function within the R package ('prepare_sciCSR') enables users to set up a working conda environment containing these Python packages; the RNA velocity Python package scVelo[10] (v0.2.4) is also included in the environment to allow users to run CellRank using RNA velocity information for direct comparisons against the transitions inferred using CSR/SHM information as implemented in sciCSR.

## Datasets

**Data preprocessing.** Unless otherwise stated, the datasets below were obtained and processed as follows: FASTQ files for the datasets were downloaded either by using the wget command-line utility (for ArrayExpress) or the NCBI Sequence Reads Archive (SRA) Toolkit (v2.11.1) (for Gene Expression Omnibus (GEO)). Raw FASTQ files were aligned to either the GRCh38 (for human data) or mm10 (for mouse) reference genomes and cell-by-gene count matrices were generated using 10x Genomics cellranger (v7.0.0). For datasets with scRNA-seq and scBCR-seq data available in parallel, the 'cellranger multi' option was used to simultaneously call cells with both transcriptomic and VDJ data; otherwise the 'cellranger count' option was used. Reference genome annotation (version 3.0.0 for both human and mouse) was downloaded from the 10x Genomics cellranger website. The 'filtered_feature_bc_matrix' folder of cellranger output was read into R using Seurat::Read10X function as a cell-by-gene count matrix. Before generating a Seurat object that holds this count matrix, counts for genes encoding Ig V, D and J segments were summed for each cell as count for a meta-gene 'Ig-vdj' to eliminate any effects caused by individual-/clonotype-specific VDJ gene usage to downstream clustering and differential expression analyses. The count matrix was then subsetted such that only features detected in at least three cells and cells with transcripts from at least 200 distinct features were retained. Dimensionality reduction and clustering analysis were performed stepwise as follows:

(1) removal of cells with the percentage of mitochondrial reads larger than 10%;
(2) log-normalization using Seurat::NormalizeData with default parameters;
(3) identification of variably expressed genes using Seurat::FindVariableMarkers and removal of BCR-/TCR-specific genes from this variably expressed gene list as per Stewart et al.[4];
(4) scale and center the normalized count data using Seurat::ScaleData with default parameters;
(5) principal component analysis using only the pruned variably expressed gene list;
(6) UMAP using Seurat::RunUMAP, calling the Python umap-learn package and using correlation as the metric. We retained only the top $p$ principal components, each of which explained at least 1.5% of the total variance;

(7) construct kNN network using Seurat::FindNeighbors with default parameters and Louvain clustering on the kNN network. The resolution parameter is specific to each dataset and is indicated below separately.

**Aicda gene knockout mice study from Gómez-Escolar et al..** This dataset contains scRNA-seq data from reporter mice where historical AID (*Aicda*) expression could be traced by sorting for expression of the fluorescent tdTomato (Tom) protein versus mice that are AID deficient. Cells from the spleen were collected and prepared for scRNA-seq and scBCR-seq after immunization with injection of the ovalbumin (OVA) protein. FASTQ files were downloaded from SRA accession SRP348368 and processed as detailed above, for sciCSR analysis of productive/sterile IgH transcripts. The transcript count matrix containing data for the other genes, as well as the cell metadata, were directly downloaded from the associated GEO entry GSE189775 and imported using the Seurat package in R. sciCSR was applied to analyze transitions both between cell clusters and between BCR isotypes, separately for the AID-deficient and WT cells, using the signature-based CSR potential defined above. For inferring CSR, the following isotypes were chosen as the source/target states: WT—IgM (source), IgA (target); AID-deficient—IgM (source), IgG2b (target) (since for the AID-deficient cell subset no cells harbor isotypes further beyond IgG2b in the IgH locus).

**Il23 gene knockout mice study from Hong et al..** This dataset contains scRNA-seq and scBCR-seq data from splenocytes of autoimmune BXD2 mice with knockout of the p19 component of *Il23* (hereafter *Il23 p19*$^{-/-}$) and WT BXD2 mice. FASTQ files were downloaded from SRA accession SRP250728 and processed as detailed above. Unspliced and spliced transcripts were quantified using velocyto[9] (v0.17.17), supplying both the mm10 General Transfer Format file from cellranger references (see above) and the mm10 repeat mask General Transfer Format file obtained from the UCSC genome browser (RepeatMasker track) as arguments. RNA velocity estimation using the scVelo 'dynamical' model was performed using the scVelo Python package (v0.2.4) with default parameters. The filtered_contig_annotation files output from 'cellranger vdj' were supplemented with the percentage identity to the germline V gene for each sequence by merging the cellranger filtered_contig_annotation files with the AIRR-formatted output of IMGT/HighV-Quest[75,76] (accessed 5 July 2022) analysis of the same set of contigs. These scBCR-seq annotations were merged with the Seurat data object holding scRNA-seq data as detailed above. The dataset was filtered to retain B cells fulfilling the following criteria: (1) with matching heavy/light chain sequence from scBCR-seq, and (2) non-zero expression of both *Cd19* and *Ms4a1*. A resolution parameter of 0.4 was used in Seurat::FindClusters to determine cell clusters. sciCSR was applied to infer transitions both between cell clusters and between BCR isotypes, using (1) the signature-based CSR potential, (2) SHM level and (3) RNA velocity information. For transitions between cell clusters, the source state was chosen to be the IgM-hi memory cluster (Extended Data Fig. 8) and the target state as the plasmablast cluster (which typically produce a large amount of transcripts mapping to V genes; Extended Data Fig. 8). For inferring CSR, IgM was chosen as the source and IgA as the target state.

**SARS-CoV-2 vaccination longitudinal follow-up from Kim et al..** Raw FASTQ files were downloaded from SRA accession SRP356296 and aligned as described above; these alignments were used as input for sciCSR to enumerate productive/sterile transcripts. The scRNA-seq cell-by-gene count data were downloaded from the Zenodo repository[77] as a Python AnnData object. Only cells from donors 07 and 20 at weeks 4 (day 28), 7 (day 60), 15 (day 110) and 29 (day 201) were retained, exported as .h5ad files and converted to R Seurat objects using the R SeuratDisk package (v0.0.0.9020). Since the count data were already preprocessed and normalized, this Seurat object was directly analyzed.

scBCR-seq data from the same cells were downloaded from the same Zenodo repository. This data table annotated heavy-chain sequences as IgM/G/A but lacked annotations of the subclasses. IgBLAST[78] (v1.19.0) was used (with default parameters) to call subclasses for these sequences, using nucleotide sequences of germline immunoglobulin alleles downloaded from IMGT (accessed 28 July 2022) as reference. The human C-region artificially spliced exons sets were downloaded[79] and used as reference set. The scBCR-seq table annotated with heavy-chain subclass information was merged into the metadata slot of the Seurat data objects. Only the cells labeled 'GC' in the author-provided cell type annotation were considered. Calculation of CSR potential and SHM level was carried out using sciCSR as described above. The transition inference method in sciCSR was used using the CSR potential as pseudotime, to infer CSR using IgM as the source state and IgG4 as the target state; since there were no cells with IgE or IgA2 as their BCR isotypes, IgG4 was the state furthest along the IgH locus for this dataset. The isotype annotation was taken from the merged scBCR-seq metadata.

To compare CSR inference with observed BCR isotype distribution, we first calculated, for each isotype, the total inward flux (that is, amount of flux towards each isotype) inferred using TPT, and reasoned that this total inward TPT flux should predict the isotype distribution observed in a subsequent time point. We therefore compared the TPT inward flux at week 7 to scBCR-seq isotype distribution at week 15, and TPT inward flux at week 15 to observation at week 29. The similarity of the TPT-predicted and observed distributions was evaluated using cosine similarity, calculated using the R philentropy[73] package (v0.5.0).

**Reference atlases.** We integrated previously published datasets to generate the following B cell atlas, profiled using 10x Genomics 5′ technologies, for obtaining reference isotype signatures in estimating CSR potentials for user-supplied datasets. In each case the aligned BAM files were used as input to sciCSR for quantifying productive/sterile transcripts and the resultant count matrix was decomposed using NMF to derive a reference signature matrix (see section 'Deriving CSR potential').

**Human B cell atlas.** We integrated the following two scRNA-seq datasets of human B cells:

(1) Data from Stewart et al.[4] containing B cells from the peripheral blood fluorescence-activated cell sorting (FACS)-sorted into five phenotypically defined populations (Transitional (CD19+ IgD+CD27−CD10+), Naive (CD19+IgD+CD27−CD10−), IgM memory (CD19+IgD+CD27+), Classical Memory (CD19+IgD−CD27+), Double Negative (CD19+IgD−CD27−)). Data from the donor HB6 were considered in this atlas. Data preprocessing was previously described[4], and the aligned BAM files were directly taken as input to sciCSR to generate productive/sterile transcript counts, and merged with the R Seurat data object available at ArrayExpress accession E-MTAB-9544.

(2) Data from King et al.[5] containing B cells from human tonsil samples. Raw FASTQ files were downloaded from ArrayExpress accession E-MTAB-9005 and processed as described above for sciCSR productive/sterile transcript quantification. B cell scRNA-seq transcriptomic count data were downloaded as a Seurat data object from the same ArrayExpress accession record.

**Mouse B cell atlas.** We processed and integrated the following two datasets to form a mouse B cell atlas:

(1) Data from Mathew et al.[6] for B cells from mediastinal lymph node, lung and spleen of mice at days 7, 14 and 28 days after influenza infection. Raw FASTQ files were downloaded from ArrayExpress accession E-MTAB-9478 (scRNA-seq) and E-MTAB-9491 (scBCR-seq) and preprocessed as described above using the 'cellranger multi' function.

(2) Data from Luo et al.[39] for peritoneal B cells sampled from healthy neonates, young adults, and elderly mice. Raw FASTQ files were downloaded from ArrayExpress accession E-MTAB-10081 and preprocessed using 'cellranger multi'. Only data from samples D, E and F were considered as scBCR-seq, and scRNA-seq data were obtained in parallel only for these samples.

**Data integration.** For the mouse atlas, count matrices from Mathew et al. and Luo et al. were read in and directly combined before normalization, dimensionality reduction and clustering. For the human atlas, the two Seurat data objects holding data from King et al. and Stewart et al. were integrated by following the data integration protocol in the Seurat[46] package. Briefly, anchoring points for data integration were established using the SelectIntegrationFeatures followed by FindIntegrationAnchors function, and the two datasets were integrated using the IntegrateData function in Seurat. All functions were evaluated using default parameters.

**scRNA-seq analysis of in vitro cultures of naive B cells in IgG-polarizing conditions**

**Ethics.** Blood samples were collected from healthy individuals. Ethical approval was obtained from the UCL ethical committee, under REC reference no. 14/SC/1200. Sample storage complied with requirements of the Data Protection Act 1998 and the Human Tissue Act 2004. Two male donors and one female donor between 25 and 45 years of age were selected. Informed consent was obtained from all donors.

**In vitro CSR.** PBMCs from three healthy controls were obtained by venepuncture, separated by Ficoll density gradient centrifugation and cryopreserved in fetal bovine serum (FBS) with 10% dimethyl sulfoxide at −80 °C for a week. PBMCs were thawed, and total B cells were obtained by negative selection using the EasySep Human B Cell Isolation Kit (StemCell Technologies, #17954). Total B cells were then stained for further flow cytometry sorting with anti-CD19 (1:200), anti-CD27 (1:200) and anti-IgD (1:200). Viability staining using LIVE/DEAD Fixable Near IR Viability Kit (Invitrogen, #L34975) was performed together with the surface staining. Naive B cells (CD27−IgD+) were sorted by flow cytometry using a BD FACSAria Fusion and BD FACSDiva software version 9.4, and collected at 4 °C in recovery medium (50% heat-inactivated FBS + 50% RPMI 1640 (Sigma-Aldrich, #R7388). Naive B cells at day 0 were washed in PBS with 5% fetal calf serum and run on the Chromium 10x controller as described in the single-cell methodology below. In parallel, the rest of the sorted naive B cells, 250,000 cells per well, were then cultured in vitro under IgG polarizing class switch consisting of complete RPMI (10% heat-inactivated FBS and 1% penicillin–streptomycin) + 1 µg ml⁻¹ anti-IgM (JacksonImmunoResearch, #309-006-043) + 1 µg ml⁻¹ CD40L (Enzo Life Sciences, #ALX-522-110-C010) and 50 ng ml⁻¹ IFNγ (Bio-Techne, #285-IF-100/CF) in a round bottom 96-well plate at 37 °C and 5% CO₂ for 3 or 6 days. IgG polarizing class-switch medium was refreshed on day 2 by gently centrifuging the cells (300g for 10 min) and fresh medium added with the above-mentioned factors.

**Flow cytometry staining and analysis.** Cultured B cells in IgG polarizing class-switch medium from each donor were collected and stained for subsequent flow cytometry measurement. Anti-CD19 (1:200), anti-CD27 (1:200), anti-CD24 (1:200) and anti-CD38 (1:200) were used to stain the cells extracellularly. Viability staining using LIVE/DEAD Fixable Blue Dead Cell Stain Kit (Invitrogen, #L23105) was performed together with the surface staining. Cells were then fixed and permeabilized using eBioscience Intracellular Fixation & Permeabilization Buffer Set (Invitrogen, #88-8824-00) and stained intracellularly with anti-IgD (1:200), anti-IgM (1:200), anti-IgG1 (1:200), anti-IgG2 (1:200), anti-IgG3 (1:200), anti-IgA (1:200) and anti-IgE (1:200). Samples were acquired in a Cytek Aurora cytometer (five lasers) using SpectroFlo software

version 3.0.3 with automated unmixing. Day 3 flow cytometry was performed in a pooled sample of the three healthy controls as the main part of the recovered cells were dedicated for the single-cell readout. Analyses were performed using FlowJo version 10.8.1. Supplementary Fig. 1 illustrates the gating strategy used in the flow cytometry analysis.

**scRNA-seq.** Cultured B cells in IgG polarizing class-switch medium from each donor were collected on days 3 and 6, then counted, and their viability was measured using acridine orange/propidium iodide (Nexcelom, #CS2-0106-5mL) as viability dye in a Cellometer Auto 2000 cell counter (Nexcelom). Dead cells were removed using the EasySep Dead Cell Removal (Annexin V) Kit (StemCell Technologies, 17899). For each time point, samples from each healthy control were stained separately using TotalSeq hashtags 1, 2 and 3 (BioLegend #394661, #394663 and #394665) (1:100) and anti-CD27 Total Seq (BioLegend, #302853) (1:200) for 30 min in PBS with 5% fetal calf serum supplement. Cells from each were then washed, counted and pooled at equal concentrations. Pooled cells were counted and run on a Chromium 10x controller using 5′ v2 chemistry (10x Genomics) in one lane with an expected recovery rate of 5,000 cells per lane, according to the manufacturer's instructions. Libraries were generated according to 10x Genomics instructions and run on a High Output HiSeq2500 in 30-10-100 format with a target of 50,000 reads per GEX library and 5,000 reads per BCR and Feature Barcode library.

**Data analysis.** Raw FASTQ files were aligned and annotated using cellranger (v7.1.0). The 'cellranger multi' option was used to jointly call gene counts, VDJ and Feature Barcode data with matching cell barcodes. Feature barcodes were demultiplexed using the HTODemux function in Seurat (v4.3.0) using default parameters. We follow the procedure detailed above in the Dataset section to perform quality control, normalization and definition of variably expressed genes. Harmony[80] (v0.1.1) was applied to remove batch effects (between donors). The kNN network was calculated on the output from harmony and cell clusters were defined using Seurat::FindClusters on the kNN graph, with resolution parameter of 0.5. For scBCR-seq, the filtered contig FASTA files were subject to IgBLAST[78] (v1.19.0) analysis. Since cellranger prioritizes full-length VDJ sequences for annotations, we reasoned that for a small number of cells where switched (that is, with isotypes beyond IgM and IgD) productive sequences were present, these switched transcripts would be omitted by chance in the final filtered contig annotation output because of incomplete read coverage. These cells were identified by comparing the C gene annotations in the filtered ('filtered_contig_annotations.csv') and unfiltered ('all_contig_annotations.csv') contig annotation files from the cellranger output, and overriding such annotations if switched transcripts with lower read coverage were found in the unfiltered data. Functions in sciCSR were used to integrate the scBCR-seq repertoire data to the prepared Seurat data object, and to enumerate productive and sterile transcripts from the scRNA-seq BAM files. Cells were grouped by the isotype of the sterile transcripts found with the highest transcript counts for each cell; we performed differential expression analysis based on these cell groupings using Seurat::FindMarkers.

### Statistics and data visualization

All analysis was performed under the R statistical computing environment (v4.2.2). Statistical analyses specific to implementations in the sciCSR package were detailed under the 'Implementation' subsection of the 'The sciCSR algorithm' section. Data visualization was generated using the R ggplot2 package (v3.4.1), and *t*-tests and multiple test corrections were performed using stats functions in R v4.2.2.

### Reporting summary

Further information on research design is available in the Nature Portfolio Reporting Summary linked to this article.

## Data availability

The scRNA-seq and scBCR-seq data of the IFNγ culture experiment are accessible via ArrayExpress (accession E-MTAB-13050). All other datasets used in this work are publicly available: Kim et al.[40] (GEO entry GSE195673), Gómez-Escolar et al.[41] (GSE189775), Hong et al.[42] (GSE145922), Stewart et al.[4] (E-MTAB-9544), King et al.[5] (E-MTAB-9005), Mathew et al.[6] (E-MTAB-9478 and E-MTAB-9491) and Luo et al.[39] (E-MTAB-10081). For GEO entries raw FASTQ files were downloaded from the associated SRA entries. Processed data files generated in this study can be found in the Zenodo repository https://doi.org/10.5281/zenodo.8005705. Reference genome data (hg38, mm10) used in aligning scRNA-seq datasets were obtained from the 10x cellranger website (https://support.10xgenomics.com/single-cell-vdj/software/downloads/latest). Source data are provided with this paper.

## Code availability

Source code for the sciCSR package is available at https://github.com/Fraternalilab/sciCSR. Documentation and vignettes can be found in the github repository. Analysis notebooks and code used in generating the analysis presented in this manuscript can be found at https://github.com/Fraternalilab/sciCSR-analysis.

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

## Acknowledgements

This work was funded by the Biotechnology and Biological Sciences Research Council (BB/T002212/1 with F.F. as principal investigator). The funders had no role in the collection and analysis of the samples, in the interpretation of data, in writing the report, nor in the decision to submit the paper for publication. We also acknowledge J. Kleinjung for his critical reading of the manuscript.

## Author contributions

J.C.F.N. designed the method, implemented sciCSR and performed all the scRNA-seq data analysis, with input from F.F. G.M.G. and A.T.S. carried out the IFNγ culture experiment with the help of P.B. C.M. and D.K.D.-W. supervised the IFNγ culture experiment. D.K.D.-W. contributed initial conceptual input in the detection of sterile and productive transcripts. J.C.F.N. wrote the manuscript and the Methods section with critical input from F.F., except that G.M.G. and A.T.S. wrote the methods for the IFNγ culture experiment. F.F. supervised the overall project. All authors read, commented and approved the final manuscript.

## Competing interests

The authors declare no competing interests.

## Additional information

**Extended data** is available for this paper at https://doi.org/10.1038/s41592-023-02060-1.

**Correspondence and requests for materials** should be addressed to Joseph C. F. Ng or Franca Fraternali.

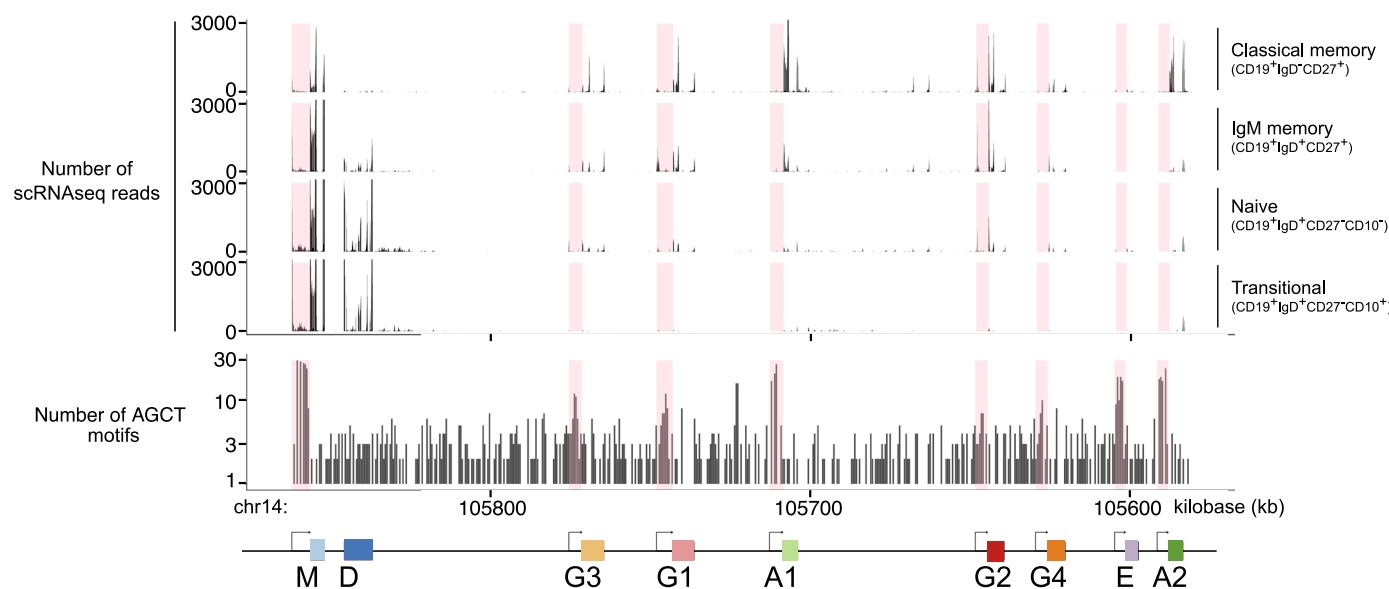

**Extended Data Fig. 1 | Histogram of scRNA-seq read counts from the Stewart et al. dataset across the *IGH* genomic locus.** The distribution of constant region genes was illustrated at the bottom of the histogram (coloured boxes) along with counts of 5′-AGCT-3′ motifs in sliding windows of 500 base-pairs (histogram in the middle). Regions 5′ of the constant region coding segment were highlighted with pink rectangles.

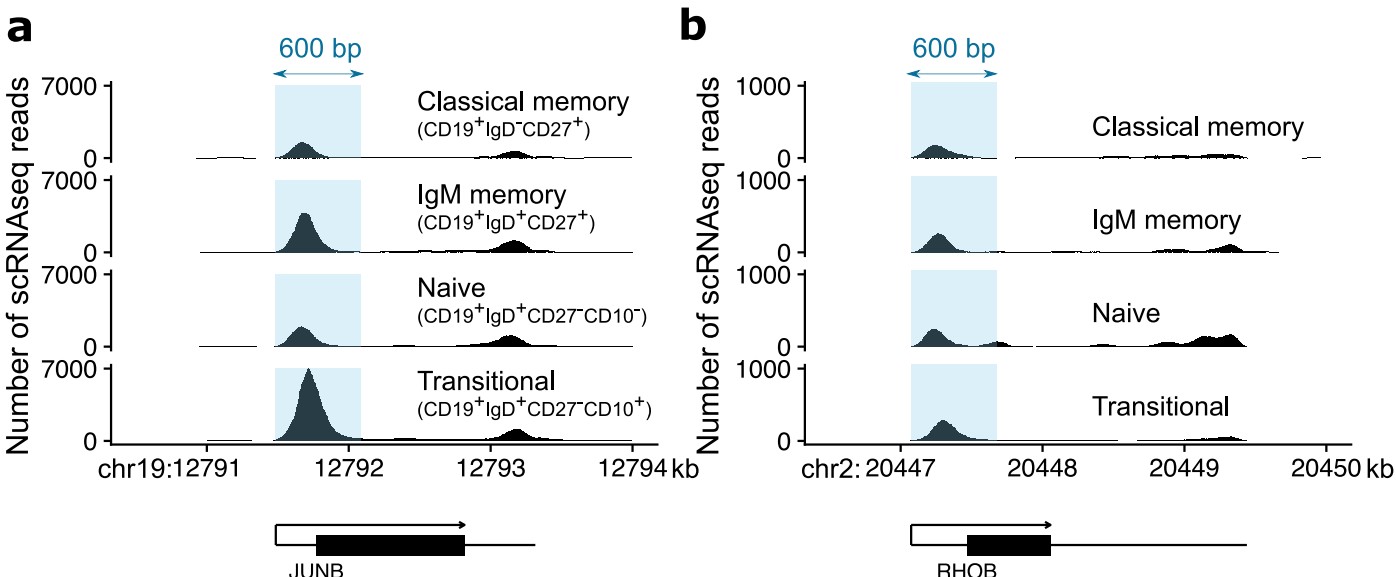

**Extended Data Fig. 2 | Distribution of scRNA-seq reads from Stewart et al. across two single-exon transcripts.** Two genes were shown: *JUNB* (Ensembl transcript ENST00000302754.6, panel **a)** and *RHOB* (ENST00000272233.6, panel **b)**. Regions of 600 base-pairs (bp) at the 5′ end of the transcript were highlighted (blue rectangles) to illustrate the concentration of reads at this region from the scRNA-seq experiments.

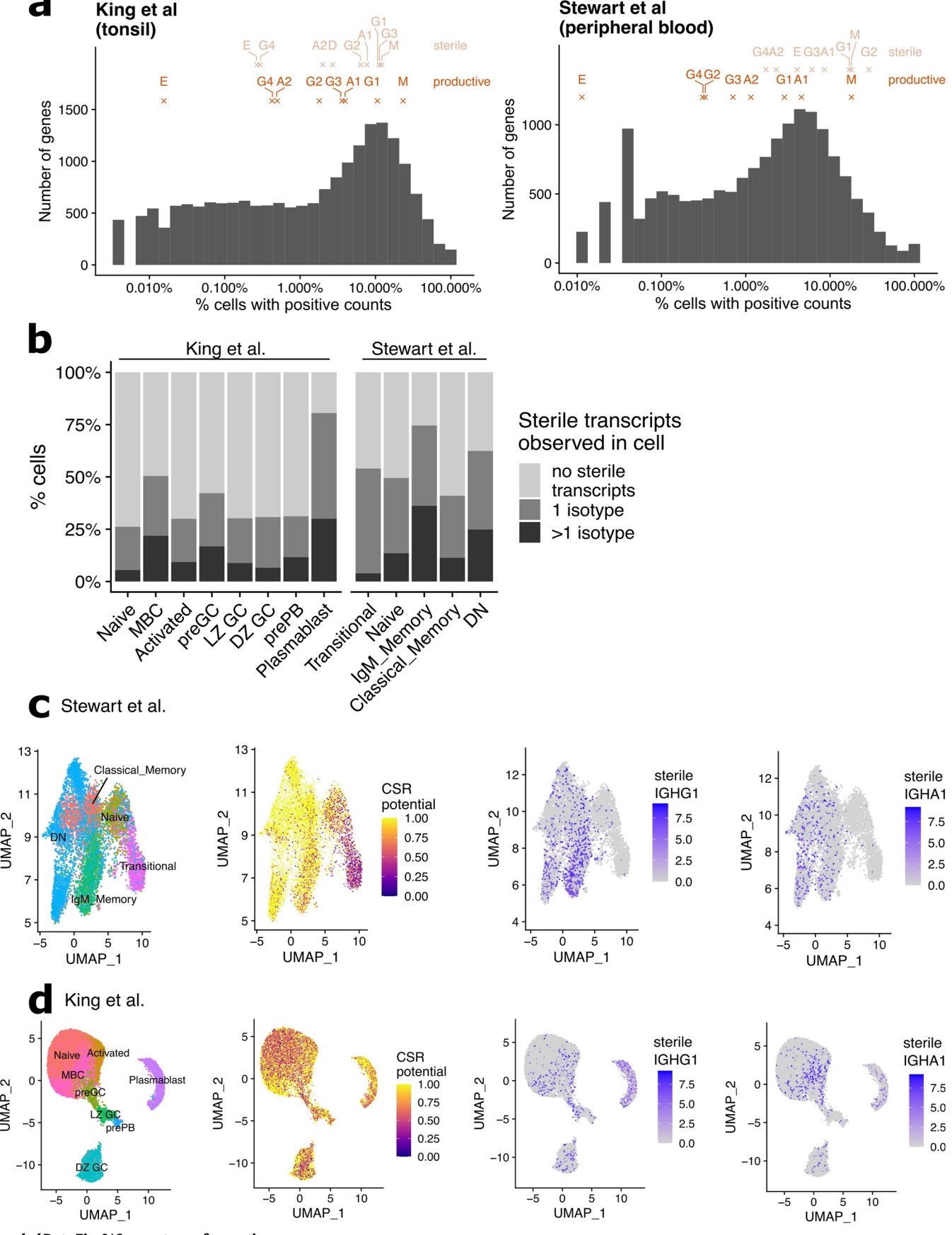

**Extended Data Fig. 3 | See next page for caption.**

**Extended Data Fig. 3 | Distribution of productive/sterile IgH transcripts.**
**(a)** Sparsity of productive/sterile IgH transcripts. Grey bars depict distribution of all transcripts with count data in the King et al. tonsil B cell dataset (left) and the Stewart et al. peripheral B cell dataset (right), showing for each transcript the proportion of cells in the data with positive counts. The values for each productive and sterile IgH transcript are noted at the top of the histogram with crosses. **(b)** The number of isotypes presented in the sterile transcripts observed in each cell in the King et al. and Stewart et al. datasets, expressed per cell type. Each cell was assessed in terms of the number of isotypes represented in the sterile transcripts associated with each cell. **(c,d)** (from left to right) UMAP visualisation with cell type annotation, CSR potential scores, sterile *IGHG1* and sterile *IGHG3* expression from the Stewart et al. (panel c) and King et al. (d) datasets.

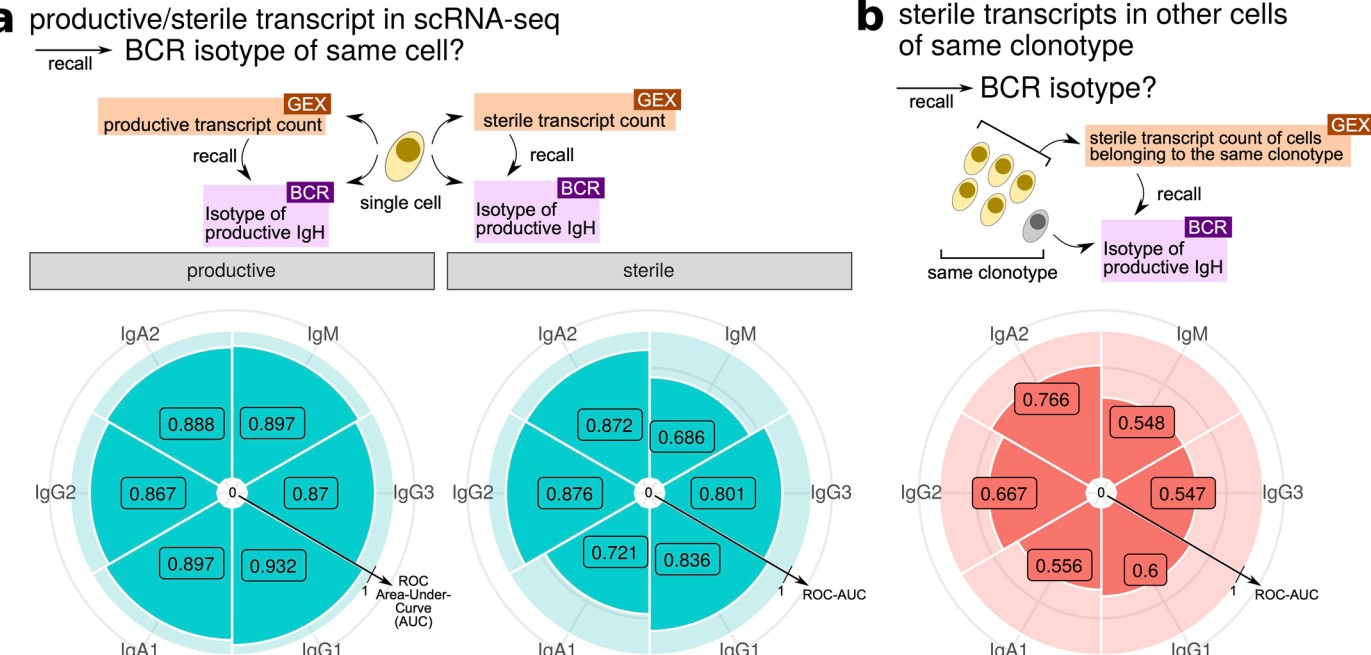

**a** productive/sterile transcript in scRNA-seq
→recall BCR isotype of same cell?

**b** sterile transcripts in other cells of same clonotype
→recall BCR isotype?

**Extended Data Fig. 4 | Recall BCR isotypes using productive/sterile transcript count recovered using sciCSR in the King et al. tonsil B cell dataset. (a)** (left) Productive transcript count was recovered from scRNA-seq data using sciCSR and treated as a predictor of BCR isotype of the same cell according to the scBCR-seq data. Pizza charts depict the area-under-curve of the receiver operating characteristic (AUC-ROC) curve separately for each isotype. (right)

Sterile transcript count was recovered from scRNA-seq data using sciCSR and treated as a predictor of BCR isotype of the same cell. Pizza charts depict AUC-ROC separately for each isotype. **(b)** Sterile transcript count was recovered from scRNA-seq data using sciCSR and treated as a predictor of BCR isotype of cells belonging to the same clonotype. Pizza charts depict AUC-ROC separately for each isotype.

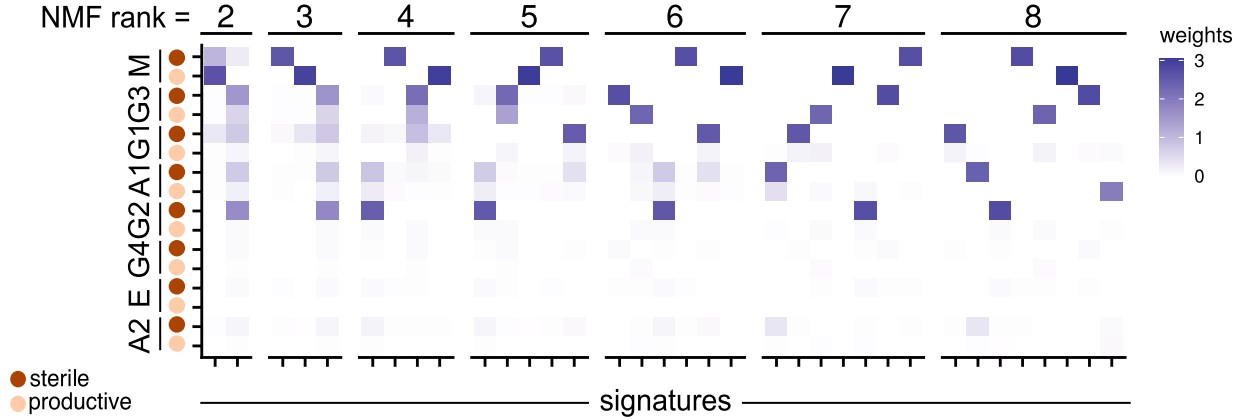

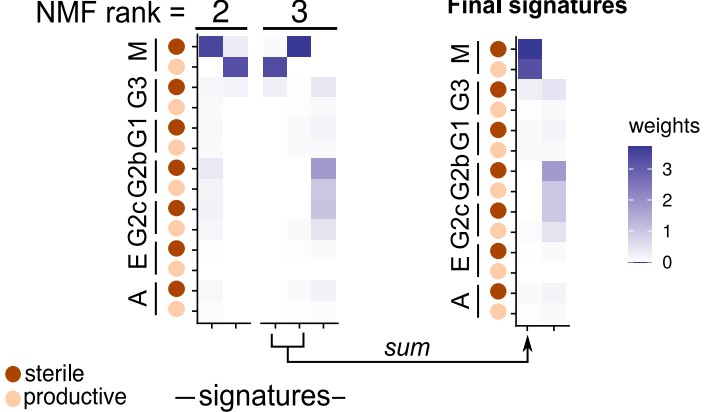

**Extended Data Fig. 5 | Isotype signatures identified using nonnegative matrix factorization (NMF) on human and mouse B cell atlases. (a)** Effect of changing NMF rank (that is the desired number of signatures) on the human B cell atlas. The signature matrix under each NMF setup was shown. The final isotype signature matrix for human is the rank = 2 condition. **(b)** Deriving the mouse signature matrix. (left) the NMF signature matrices using rank = 2 and 3. Neither setup derived a signature matrix similar to the rank = 2 condition in the human atlas (see panel a), although it appears that the IgM-dominant signature was separated into two signatures in the rank = 3 condition (first two columns). (right) The final signature for mouse was derived by summing together the first two signatures in the rank = 3 condition. The final signature matrix was shown here.

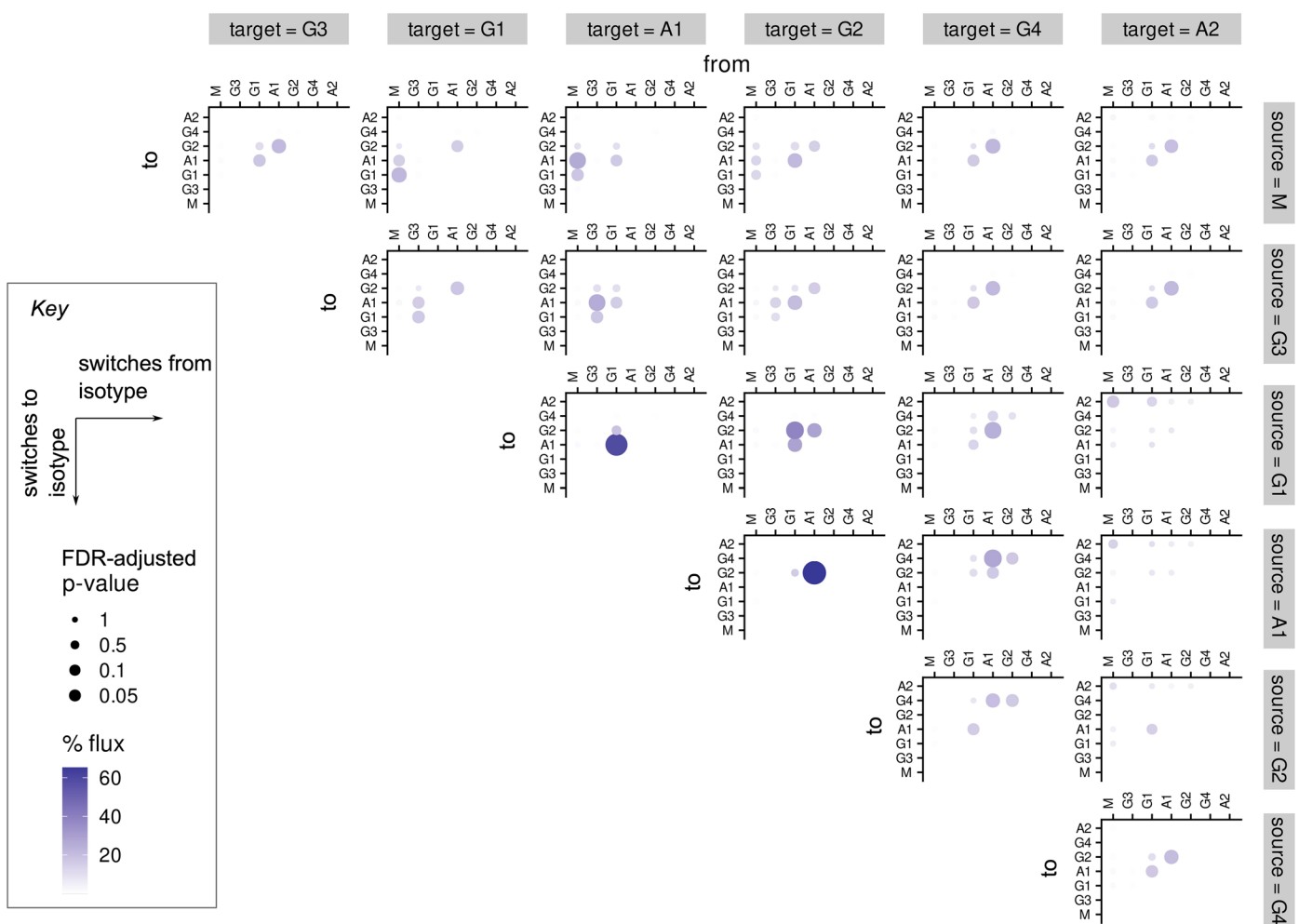

**Extended Data Fig. 6 | Effect of selecting different source and target states on TPT results.** TPT inference was performed using different source and target states, using the week 15 timepoint of donor 07 from the Kim et al. vaccination time-course dataset. All inferences were performed using the NMF-derived CSR potential as pseudotime input to sciCSR which was invoked using identical and default parameters. The inferred fluxes tend to be higher for transitions involving states which are either the chosen source or target states. Statistical significance was obtained by one-sided comparisons to determine whether the flux values are greater than predictions generated from randomised transitions.

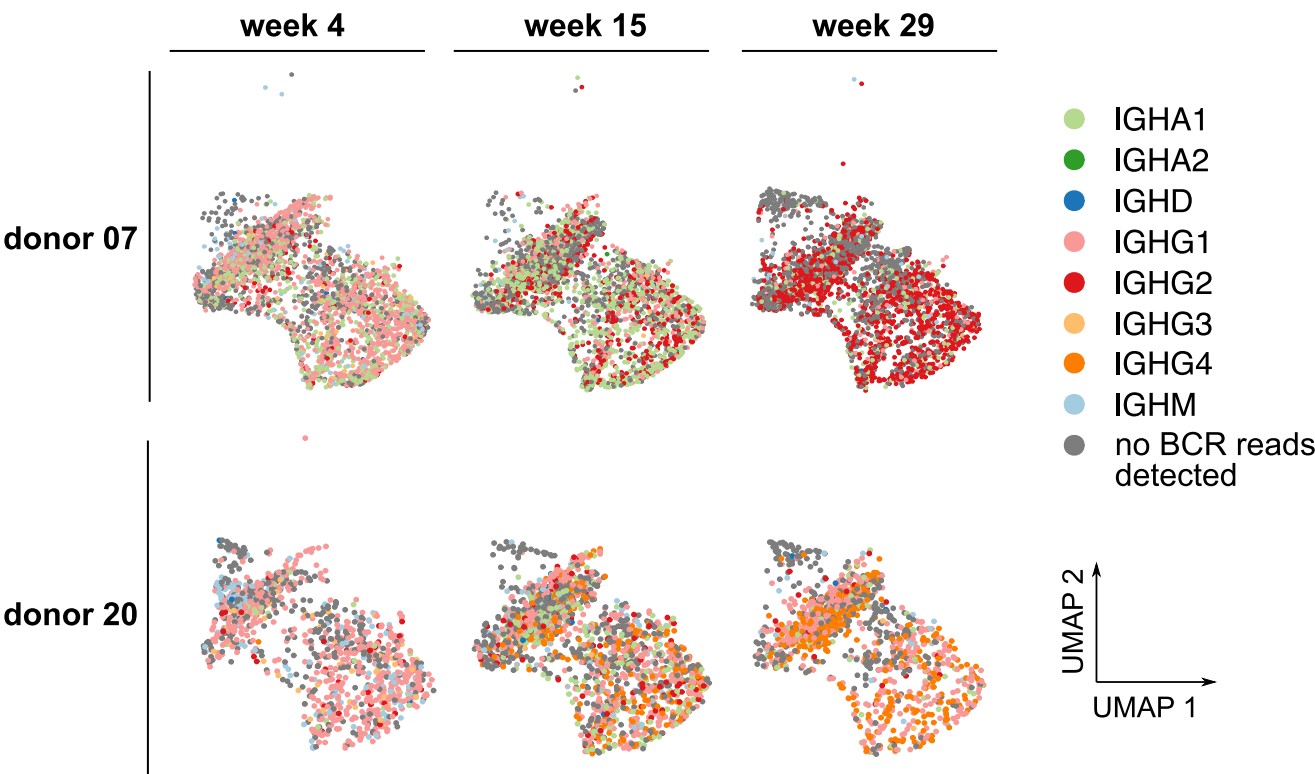

**Extended Data Fig. 7 | UMAP visualisation of GC B cell scRNA-seq data of two donors from Kim et al. dataset.** Visualisations were separated by timepoints (columns) and coloured by BCR isotype indicated in the single-cell-matched scBCR-seq data. All donor/timepoint combinations were projected onto the same UMAP reduction.

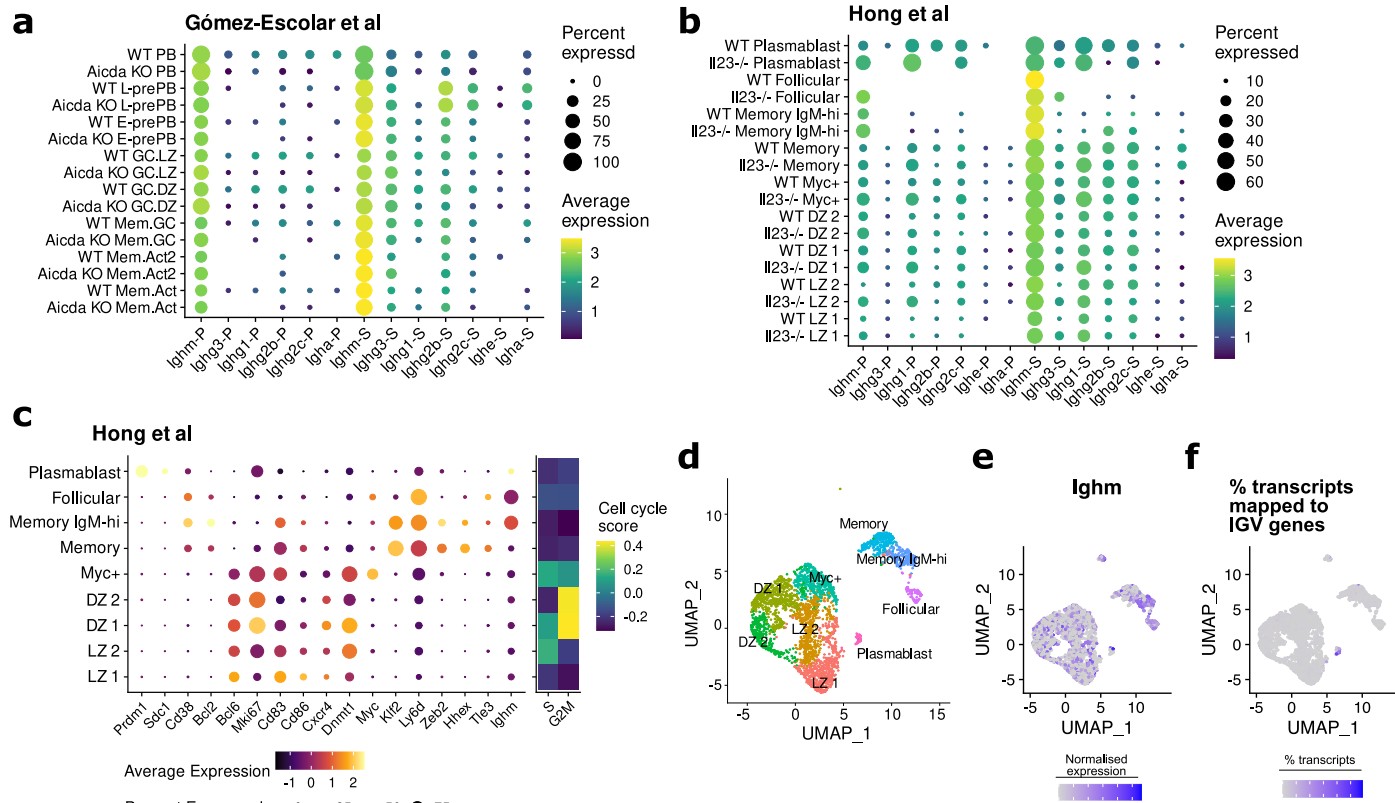

**Extended Data Fig. 8 | Annotations on mouse knockout dataset use cases. (a)** Sterile (horizontal axis, features ending with '-S') and productive (ending with '-P') IgH transcript expression level in the Gómez-Escolar et al. *Aicda* knockout dataset. **(b)** Sterile and productive IgH transcript expression level in the Hong et al. *Il23* knockout dataset. **(c)** (left) Key gene marker expression levels in the Hong et al. dataset. (right) The cell cycle scoring method in Seurat was applied to assign scores for the S and G2M phases for each cell. Heatmap denotes the mean score for each cell cluster. **(d)** UMAP projection of the Hong et al. data grouped into 10 cell clusters (colours). **(e)** Expression of *Ighm* gene in the Hong et al. data. **(f)** Percentage of transcripts which are mapped to *IGV* genes for each cell expressed as a heat scale.

# Reporting Summary

## Statistics

For all statistical analyses, confirm that the following items are present in the figure legend, table legend, main text, or Methods section.

| n/a | Confirmed | |
|---|---|---|
| ☐ | ☒ | The exact sample size (*n*) for each experimental group/condition, given as a discrete number and unit of measurement |
| ☐ | ☒ | A statement on whether measurements were taken from distinct samples or whether the same sample was measured repeatedly |
| ☐ | ☒ | The statistical test(s) used AND whether they are one- or two-sided *Only common tests should be described solely by name; describe more complex techniques in the Methods section.* |
| ☐ | ☒ | A description of all covariates tested |
| ☐ | ☒ | A description of any assumptions or corrections, such as tests of normality and adjustment for multiple comparisons |
| ☐ | ☒ | A full description of the statistical parameters including central tendency (e.g. means) or other basic estimates (e.g. regression coefficient) AND variation (e.g. standard deviation) or associated estimates of uncertainty (e.g. confidence intervals) |
| ☐ | ☒ | For null hypothesis testing, the test statistic (e.g. *F*, *t*, *r*) with confidence intervals, effect sizes, degrees of freedom and *P* value noted *Give P values as exact values whenever suitable.* |
| ☒ | ☐ | For Bayesian analysis, information on the choice of priors and Markov chain Monte Carlo settings |
| ☐ | ☒ | For hierarchical and complex designs, identification of the appropriate level for tests and full reporting of outcomes |
| ☐ | ☒ | Estimates of effect sizes (e.g. Cohen's *d*, Pearson's *r*), indicating how they were calculated |

*Our web collection on statistics for biologists contains articles on many of the points above.*

## Software and code

Policy information about availability of computer code

| Data collection | We used publicly available data deposited in the Gene Expression Omnibus (GEO) and ArrayExpress. FASTQ files were downloaded either by using the wget command-line utility (for ArrayExpress) or the NCBI SRA Toolkit (v2.11.1) (for GEO) |
|---|---|
| Data analysis | Code for data analysis presented in this manuscript has been implemented in the R package sciCSR which is available at: https://github.com/Fraternalilab/sciCSR. Documentation and vignettes can be found in the GitHub repository. Analysis notebooks and code used in generating the analysis presented in this manuscript can be found at https://github.com/Fraternalilab/sciCSR-analysis. Analysis presented in this manuscript was performed using R version v4.2.2. For the simulated dataset (Fig. 2), they were generated and analysed using the R package polyester (v1.29.1.1),  as well as command-line programs HISAT2 (v2.2.1) and STAR (v2.5.1.b). |

For manuscripts utilizing custom algorithms or software that are central to the research but not yet described in published literature, software must be made available to editors and reviewers. We strongly encourage code deposition in a community repository (e.g. GitHub). See the Nature Portfolio guidelines for submitting code & software for further information.

## Data

Policy information about availability of data

All manuscripts must include a data availability statement. This statement should provide the following information, where applicable:
- Accession codes, unique identifiers, or web links for publicly available datasets
- A description of any restrictions on data availability
- For clinical datasets or third party data, please ensure that the statement adheres to our policy

The scRNA-seq and scBCR-seq data of the IFNγ culture experiment are accessible via ArrayExpress (accession E-MTAB-13050). All other datasets used in this work are publicly available: Kim et al.78 (Gene Expression Omnibus [GEO] entry GSE195673), Gómez-Escolar et al.79 (GSE189775), Hong et al.80 (GSE145922), Stewart et al.10 (E-MTAB-9544), King et al.11 (E-MTAB-9005), Mathew et al.13 (E-MTAB-9478 and E-MTAB-9491) and Luo et al.14 (E-MTAB-10081). For GEO entries raw FASTQ files were downloaded from the associated Sequence Read Archive (SRA) entries. Processed data files generated in this study can be found in the Zenodo repository https://dx.doi.org/10.5281/zenodo.8005705.  Reference genome data (hg38, mm10) used in aligning scRNA-seq datasets were obtained from the 10x cellranger website (https://support.10xgenomics.com/single-cell-vdj/software/downloads/latest).

## Human research participants

Policy information about studies involving human research participants and Sex and Gender in Research.

| | |
|---|---|
| Reporting on sex and gender | For the in vitro experiments, they were performed using PBMCs from 3 healthy controls. Although sex is not know to have an effect on the readouts shown in this publication, we aimed to balance this factor by choosing 2 males and one female. |
| Population characteristics | Participants were reported as healthy. They specifically reported free of immune-related diseases. Age was restricted between 25 and 45 years of age as this factor is known to affect immune-related readouts. Ethnicity of the participants/healthy controls was reported as caucasian. |
| Recruitment | Healthy adults were recruited as part of an effort of the research team to build a repository of healthy controls. Samples from this repository include research team co-workers and public health care workers. No self-selection bias that could impact the results has been identified. |
| Ethics oversight | Ethical approval was obtained from the College London Hospital (UCLH) Health Service ethical committee, under REC reference no. 14/SC/1200.  Informed consent was obtained from all donors. |

Note that full information on the approval of the study protocol must also be provided in the manuscript.

# Field-specific reporting

Please select the one below that is the best fit for your research. If you are not sure, read the appropriate sections before making your selection.

☒ Life sciences    ☐ Behavioural & social sciences    ☐ Ecological, evolutionary & environmental sciences

For a reference copy of the document with all sections, see nature.com/documents/nr-reporting-summary-flat.pdf

# Life sciences study design

All studies must disclose on these points even when the disclosure is negative.

| | |
|---|---|
| Sample size | For the in vitro experiments, sample size was chosen based on preliminary experiments where results of IgG induction were consistent across all donors and meaningful differences between groups were achieved. scRNA-seq data were collected from three donors to ensure data cover sufficient cells and also that the data analysed cover both male and female. Analysis was performed on biological samples for which scRNA-seq and scBCR-seq (i.e. VDJ B-cell receptor sequencing) data were publicly available and complete. |
| Data exclusions | All biological samples from the cited reports for which complete scRNA-seq and scBCR-seq data are publicly available have been included in the analysis. Similarly, all data from in vitro experiments has been included. |
| Replication | Analysis can be replicated by using the sciCSR R package made available as part of this work. The algorithm does not involve random initialisation and therefore results are consistent across multiple runs. In vitro experiments using the same conditions as the one featured in this publication have been reproduced at least 3 times with similar results. |
| Randomization | Not applicable, since all biological samples have been labelled by their biological identities/genotypes etc. in the relevant entries in data repositories. For the in vitro experiments, randomization is not applicable since this study did not aim to discover the effect of a specific treatment condition, but rather as a time-course study to monitor IgG induction. Cells from all three donors were subject to the same time-course. |
| Blinding | Not applicable, since all biological samples have been labelled by their biological identities/genotypes etc. in the relevant entries in data repositories. Similarly, sequencing data from in vitro experiment was labelled with relevant biological condition. |

# Reporting for specific materials, systems and methods

We require information from authors about some types of materials, experimental systems and methods used in many studies. Here, indicate whether each material, system or method listed is relevant to your study. If you are not sure if a list item applies to your research, read the appropriate section before selecting a response.

## Materials & experimental systems

| n/a | Involved in the study |
|---|---|
| ☐ | ☒ Antibodies |
| ☒ | ☐ Eukaryotic cell lines |
| ☒ | ☐ Palaeontology and archaeology |
| ☒ | ☐ Animals and other organisms |
| ☒ | ☐ Clinical data |
| ☒ | ☐ Dual use research of concern |

## Methods

| n/a | Involved in the study |
|---|---|
| ☒ | ☐ ChIP-seq |
| ☐ | ☒ Flow cytometry |
| ☒ | ☐ MRI-based neuroimaging |

## Antibodies

| | |
|---|---|
| Antibodies used | Anti-human CD19 - BV785 (Biolegend, Cat# 302240, clone HIB19, lot# B339489, dilution 1:200)<br>Anti-human CD27 - BV711 (BD, Cat# 740291, clone M-T271, lot# 2140520, dilution 1:200)<br>Anti- human CD24 - PE-Cy7 (Biolegend, Cat# 311120, clone ML5, lot# B5345471, dilution 1:200)<br>Anti-human CD38 - BV605 (Biolegend, Cat# 356642, clone HB-7, lot# B310479, dilution 1:200)<br>Anti-human IgD - PerCP-Cy5.5 (BD, Cat# 561315, clone IA6-2, lot# 1060319, dilution 1:200)<br>Anti-human IgM - APC/Fire™ 750 (Biolegend, Cat# 314546, clone MHM-88 , lot# B283355, dilution 1:200)<br>Anti-human IgG1 - PE (Cytogonos, Cat# CYT-IGG1PE , clone SAG1, lot# 2207125, dilution 1:200)<br>Anti-human IgG2 - PE (Cytogonos, Cat# CYT-IGG2PE , clone SAG2, lot# 2112026, dilution 1:200)<br>Anti-human IgG2 - FITC (Cytogonos, Cat# CYT-IGG2F , clone SAG2, lot# 2112024/2, dilution 1:200)<br>Anti-human IgG3 - FITC (Cytogonos, Cat# CYT-IGG3F , clone SAG3, lot# 2207060, dilution 1:200)<br>Anti-human IgA - PE-Vio 615 (Miltenyi Biotec, Cat# 130-116-882 , clone REA1014, lot# 5220507430, dilution 1:200)<br>Anti-human IgE - BUV615 (BD, Cat# 751346, clone G7-28, lot# 3046667, dilution 1:200) |
| Validation | All antibodies has been validated by other publications and/or the manufacturer in the format used in the present study. In addition, all antibodies were validated in total PBMCs as positive control where clear positive and negative populations were observed. |

## Flow Cytometry

### Plots

Confirm that:

☒ The axis labels state the marker and fluorochrome used (e.g. CD4-FITC).

☒ The axis scales are clearly visible. Include numbers along axes only for bottom left plot of group (a 'group' is an analysis of identical markers).

☒ All plots are contour plots with outliers or pseudocolor plots.

☒ A numerical value for number of cells or percentage (with statistics) is provided.

### Methodology

| | |
|---|---|
| Sample preparation | Cultured B cells in IgG polarising class-switch medium from each donor were collected and stained for subsequent flow cytometry measurement. Anti-CD19, anti-CD27, anti-CD24 and anti-CD38 were used to stain the cells extracellularly. Viability staining using LIVE/DEAD™ Fixable Blue Dead Cell Stain Kit (Invitrogen, #L23105) was performed together with the surface staining. Cells were then fixed and permeabilised using the eBioscience™ Intracellular Fixation & Permeabilization Buffer Set (Invitrogen, #88-8824-00) and stained intracellularly with anti-IgD, anti-IgM, anti-IgG1, anti-IgG2, anti-IgG3, anti-IgA and anti-IgE. |
| Instrument | Sorter: BD FACSAria™ Fusion<br>Analyser: CytekTM Aurora cytometer (5 lasers) |
| Software | Acquisition software for sorter: BD FACSDiva version 9.4<br>Acquisition software for analyser: SpectroFlo version 3.0.3 with automated unmixing.<br>Analysis software: FlowJo version 10.8.1 |
| Cell population abundance | Reported ratios in the figures are percentages of positive population among total B cells. |
| Gating strategy | Lymphocytes were selected based on SSC-H (0.1x10^6 to 3x10^6) and FCS-H (0.5x10^6 to 4x10^6). Doublets were excluded in two consecutive gates. First by using FCS-H (0.5x10^6 to 4x10^6) vs FCS-A (0.4x10^6 to 4x10^6) and selecting the singlets |

found in the diagonal. Then a second time using SSC-H (0.1x10^6 to 4x10^6) and SSC-W (<0.2 x10^6). Once doublets were excluded, living B cells were selected as CD19 positive (>10^4) cells and negative for the viability dye (<10^4). From living B cells, double-negative cells for IgM (<2x10^4) and IgD (<10^3) were selected as class-switched B cells. Within class-switched B cells, IgG1, IgG2 or IgG3 positive cells were selected based on a combination of antibodies (anti-IgG1 PE, anti-IgG2 PE, anti-IgG2 FITC and anti-IgG3 FITC). For this gating FITC positivity was considered at >10^3, and PE positivity was considered at >10^4. Withing class-switched B cells without expression of IgG1/2/3, IgA (>10^3) or IgE (>10^3) positive B cell were gated. Gating strategy for immunophenotyping of cultured B cells are presented in Supplementary Figure 1.

☒ Tick this box to confirm that a figure exemplifying the gating strategy is provided in the Supplementary Information.

