## [Peer Review File · Nature Methods]

Peer Review Information

Manuscript Title: sciCSR infers B cell state transition and predicts class-switch recombination dynamics using single-cell transcriptomic data

Corresponding author name(s): Franca Fraternali, Joseph Ng

Editorial Notes: n/a

Reviewer Comments & Decisions:

Decision Letter, initial version:

Dear Franca,

I hope all is well.

Your Article, "sciCSR infers B cell state transition and predicts class-switch recombination dynamics using single-cell transcriptomic data", has now been seen by 3 reviewers. As you will see from their comments below, although the reviewers find your work of considerable potential interest, they have raised a number of concerns. We are interested in the possibility of publishing your paper in Nature Methods, but would like to consider your response to these concerns before we reach a final decision on publication.

We therefore invite you to revise your manuscript to address these concerns. In addition, I would recommend ensuring that paper can be easily understood and adopted by researchers without a strong computational background.

* include a point-by-point response to the reviewers and to any editorial suggestions

* please underline/highlight any additions to the text or areas with other significant changes to facilitate review of the revised manuscript

* address the points listed described below to conform to our open science requirements

* ensure it complies with our general format requirements as set out in our guide to authors at www.nature.com/naturemethods

* resubmit all the necessary files electronically by using the link below to access your home page

[Redacted] This URL links to your confidential home page and associated information about manuscripts you may have submitted, or that you are reviewing for us. If you wish to forward this email to co-authors, please delete the link to your homepage.

We hope to receive your revised paper within eight weeks. If you cannot send it within this time, please let us know. In this event, we will still be happy to reconsider your paper at a later date so long as nothing similar has been accepted for publication at Nature Methods or published elsewhere.

OPEN SCIENCE REQUIREMENTS

REPORTING SUMMARY AND EDITORIAL POLICY CHECKLISTS

Please note that these forms are dynamic ‘smart pdfs’ and must therefore be downloaded and completed in Adobe Reader. We will then flatten them for ease of use by the reviewers. If you would like to reference the guidance text as you complete the template, please access these flattened versions at <http://www.nature.com/authors/policies/availability.html>.

DATA AVAILABILITY

All novel DNA and RNA sequencing data, protein sequences, genetic polymorphisms, linked genotype and phenotype data, gene expression data, macromolecular structures, and proteomics data must be deposited in a publicly accessible database, and accession codes and associated hyperlinks must be provided in the “Data Availability” section.

Please include a “Data availability” subsection in the Online Methods. This section should inform readers about the availability of the data used to support the conclusions of your study, including accession codes to public repositories, references to source data that may be published alongside the paper, unique identifiers such as URLs to data repository entries, or data set DOIs, and any other statement about data availability. At a minimum, you should include the following statement: “The data that support the findings of this study are available from the corresponding author upon request”, describing which data is available upon request and mentioning any restrictions on availability. If DOIs are provided, please include these in the Reference list (authors, title, publisher (repository name),

identifier, year). For more guidance on how to write this section please see:

<http://www.nature.com/authors/policies/data/data-availability-statements-data-citations.pdf>

CODE AVAILABILITY

Please include a “Code Availability” subsection in the Online Methods which details how your custom code is made available. Only in rare cases (where code is not central to the main conclusions of the paper) is the statement “available upon request” allowed (and reasons should be specified).

MATERIALS AVAILABILITY

SUPPLEMENTARY PROTOCOL

To help facilitate reproducibility and uptake of your method, we ask you to prepare a step-by-step Supplementary Protocol for the method described in this paper. We [encourage authors to share their step-by-step experimental protocols](https://www.nature.com/nature-research/editorial-policies/reporting-standards#protocols) on a protocol sharing platform of their choice and report the protocol DOI in the reference list. Nature Portfolio's Protocol Exchange is a free-to-use and open resource for protocols; protocols deposited in Protocol Exchange are citable and can be linked from the published article. More details can found at www.nature.com/protocolexchange/about.

ORCID

Sincerely,
Madhura

Madhura Mukhopadhyay, PhD
Senior Editor
Nature Methods

Reviewers' Comments:

Reviewer #1:
Remarks to the Author:
Summary

During B cell maturation class-switch recombination occurs. This allows the B cell to switch the effector function of the antibody they produce. Understanding how B cell isotype switching occurs is important to understand how B cells mature across time. Here, the authors introduce sciCSR, a tool that can infer B cell state transitions and predict class-switching. This method focuses on B cells and immune repertoire data, unlike conventional RNA velocity methods, by identifying productive and sterile transcripts and using a Markov state model to infer CSR dynamics.

The authors claim that this approach leads to assessing isotype distribution with high accuracy, accurately capturing class-switch patterns in mouse models, and leads to a tool better than other RNA velocity methods for analyzing immune dynamics. To show this, the authors benchmarked their tool using both simulations and empirical datasets.

My strongest endorsement of this paper is that, before I was even asked to write this review I had already read the preprint and was using the software on my own data. I have some concerns detailed below, but if addressed I think this paper will be an important and novel contribution to both immunology and single cell analysis.

Validity and originality/significance/conclusion

The major strengths of this approach is the differentiation between sterile and productive transcripts, and its use of this information to infer B cell dynamics. Shown through their benchmarking, the method appears to capture known biological trends and independent information from traditional RNA velocity methods when applied to B cells. The methods are both novel and address a clear need in the field.

Comments

My greatest concern was with the experiments shown in Fig 5. I like the idea of using these knock-out mice to try to recapitulate their unique biology with sciCSR. However, the knockout experiments are not well motivated in the text and the results are not adequately explained. What is expected from AICDA, HSP13, and IL23 knockouts and how are the results validated by them?

- 5A and 5D are low in information content, while 5C and 5E are so dense they're hard to interpret. For instance, I think it would be better to show the schematic of 5D using the actual data, so we can see how the trajectories are different.

- 5B Why is no CSR predicted for AICDA KO mice? I understand that AICDA KO should have not have class switch recombination, but I would expect those mice to still have sterile isotype transcription. Why does this not lead to predicted switching? Additionally, what does HSP13 do in class switch recombination and why do the observations fit those expectations? This should be explained in the main text. It would also be useful to show the sterile vs productive dot plot for all of these mice, perhaps as a supplemental figure.

- 5C I am generally confused as to what is shown and why it fits with the expectations of the experimental system.

- 5E it is not clear to me what is supposed to be concluded from this figure. What switching patterns are expected from this mouse system? Why would we expect the CSR and SHM trajectories to be more similar to RNA velocity in the IL23p19 mouse?

Further to my point about Fig 5B, it wasn't clear what isotype information is used to predict potential isotype switching to other types. More specifically, is switching to an isotype predicted by sterile or productive transcription of that isotype?

When describing the results of Fig 5, the authors claim that CSR and SHM trajectories give very similar predictions and that this information is different from what is provided by RNA velocity. It's noted that RNA velocity predictions are noisier and potentially less reliable, but this is not demonstrated, and it's possible the CSR/SHM trajectories are in fact less accurate. While addressing which prediction is better in general may be beyond the scope of the paper, it's important to show how these predicted trajectories differ and what aspects of B cell biology are being captured by RNA velocity vs CSR/SHM trajectories.

For Figure 4, it is not clear from the figure how the isotype predictions are different from the prior state of the system. While the isotype frequency at each week is shown in Fig 4b as a heatmap, it is hard to compare to the histograms in the rest of the figure. Instead, the frequency at w7 and w15 should be shown in the same vertical histogram as the others in Fig 4c and 4d to allow for better comparisons. Further, how well is the isotype frequency at the later timepoint (e.g. w15) predicted from the simple frequency at the earlier timepoint (e.g. w7)? This could be answered by showing the cosine similarity between the frequency at w7 vs frequency at w15 in addition to the cosine similarity between the prediction at w7 and the frequency at w15.

In my experimentation with the software, I noticed that in my own data the uncategorized (-C) reads made up a substantial portion of the overall constant region read count. It appears as if these reads are ignored, which makes me nervous given they are so numerous. Can the authors demonstrate that there are no systematic biases introduced by ignoring uncategorized reads, or show that such biases would not affect their analyses? For instance, it is possible that productive reads are more likely to be uncategorized than sterile reads, leading to systematic underestimation of productive reads.

Additionally, the methods do not sufficiently explain TPT. The description is very process-oriented, describing data structures and software functions, but does not adequately describe conceptually what is being done. For instance, where do the values in the F matrix come from and what do they represent? How is TPT different from regular continuous time Markov chains, which are used, for instance, in phylogenetic models? What is the "transition rate" in line 763?

It is difficult to understand how CSR is incorporated in tandem with the SHM information when creating the M_pseudotime matrix. For example, how would a cell of "early" CSR information like high IGM productive transcripts but high SHM rate compare to cells with higher CSR potential but low SHM?

Code review

As mentioned above, I had previously downloaded and used sciCSR on my own data. I applied it to a 10X dataset in a project oriented around isotype switching. We had previously used RT-PCR to quantify sterile isotype transcription of a particular isotype in a population of B cells. Applying sciCSR to this population confirmed those results - much more sterile than productive transcripts. Further, cells with productive transcripts of that isotype generally mapped back to cells with those constant region BCRs (despite not including BCR data in that run of sciCSR). While I haven't tested the cell trajectory analysis, I'm happy with sciCSR's performance on real data.

I found the functionality and documentation of the software to be good. One thing that appears to be missing from the github is automatic unit tests. While probably not required for publication, I strongly suggest adding those for code maintenance. For the software documentation I investigated the basic tutorial on their site. While the package seems to be a mixture of R and Python, setup was relatively painless. By the standards of most academic software, the documentation and version control is good. My one gripe is that the code in the section on adding "a zero matrix" for missing cells is a bit onerous. Better to just add that code block into a function.

While the empirical data analyzed was previously published, the simulated data was not, and should be made available on a permanent repository, such as Zenodo. Further, while the sciCSR package was available, the scripts used to actually perform the analyses in the paper should be made publicly available as well.

Reviewer #2:

Remarks to the Author:

Ng et al present a new method (sciCSR) for quantifying transcription at the immunoglobulin heavy chain locus in single-cell RNA-seq datasets and using this cell intrinsic information to predict B cell class-switch recombination (CSR) trajectories. This is a well written manuscript that offers a unique and valuable contribution to study of B cell biology with single-cell technologies. Its major advances are providing a method/software package to 1) quantify different types of transcription from IgH isotypes and 2) integrating B cell-specific information (sterile transcripts, isotype expression) and constraints (directionality of the CSR hierarchy) to inform cell trajectory predictions. This has only been possible with bespoke custom scripting in the past and not accessible to the majority of immunologists working with single-cell datasets. By applying sciCSR to observational datasets from human B cell development, a longitudinal time course of human vaccination, and mouse knockout gene models the authors provide evidence that sciCSR can model and predict the CSR dynamics in an immune response, which has a lot of potential to improve our understanding across many fields of immunology. The conclusions drawn are

for the most part measured and based on robust methods that are well described in the method section.

My major concerns relate to if/how CSR potentials of single-cells can guide downstream analyses about biological questions, and the accuracy of trajectory predictions compared to RNA velocity in Figure 5. It is unclear to me from the figures presented how accurate the cellular trajectories are compared to existing methods.

Major comments

1. Fig 2b – I found this figure difficult to interpret quantitatively – if possible please present as for analysis between IgG subtypes in Fig 3b.
2. The NMF model uses a training set from peripheral blood of naïve and memory B cells. Does this capture potentially different CSR potential/isotype features of plasmablasts or cells?
3. Line 202-203 says that the NMF is applied to reference atlases of human/mouse B cell maturation, but I don't see any figures/analysis referred to here? Does this relate to the analysis in Figure 4 and 5? Inclusion of this analysis in a similar vein to Fig 3b/c but for CSR potential between naïve -> GC -> plasma (or similar) would strengthen this section.
4. It is a little unclear to me how/if the CSR potential/signature from sciCSR can be used for single cells or to compare the biology/gene expression of different B cell groups ie a lot of the analysis is presented as pseudobulk quantitation for B cells with different isotype. If possible, some visualisation of the scores for single cells as a feature plot or similar. If the data is too sparse for this, it should be discussed.
5. An exploration of how cells with different CSR potential/signatures could be compared would be a valuable addition to this manuscript e.g. can sciCSR allow for comparison of IgG2-“destined” with IgA1-“destined” B cells to predict potential regulators or cellular mediators of this class switch decision? Is this analysis possible? If not, please discuss why.
6. Figures 4 and 5 – the analyses of existing datasets in human and mouse would benefit from including some visualisation of the single-cell maps considered. For example, I find Sup Fig S4 to be very effective visualisation of those datasets that would fit well in the main figure and is then supported by the predictive power of sciCSR. Can the authors include this information to support that that the datasets are of high quality?
7. Fig 5B – I am unclear how these predicted isotype frequencies compare with the actual isotype frequencies. Can this be presented in the figure? Was the stationary distribution determined from naïve B cells, or all IgM cells?
8. Fig 5d-e – More information is needed to appraise how these CSR/SHM trajectories compare with RNA velocity or other pseudotemporal ordering methods. For example, is the RNA velocity trajectory calculated in Fig 5d meaningful in anyway or is it purely noise? This could be assessed by looking at known B cell maturation markers. There is a lot of discussion throughout the manuscript about the limitations of alternative RNA-based trajectory inference methods, and while there are significant limitations in the application of these methods, it is also not clear if the transitions/trajectories from sciCSR are any more biologically meaningful at a cellular level – Fig 5e just shows that they are different (if I have read it correctly). Visualisation of pseudotemporal trajectory vs CSR flux on the UMAP plots

could be one option. This subject should be explored and presented more comprehensively, potentially with the different reference datasets already included.

Minor points

1. Line 55 = “Pseudotemporal ordering” rather than “Pseudotime ordering”
2. Not much discussion or reference to unguided pseudotemporal ordering methods (eg Monocle can derive pathways from a starting population, without setting the end point. Whether these are accurate or not is another matter.
3. Did the authors consider using the WRCY motif compared to AGCT – the redundancy in AICDA targeting is well documented and may show the Switch regions even more convincingly.
4. Line 99 – reference to transcriptomic footprints from CSR made me think of class switch circles. A possible extension of sciCSR for this or another methods paper could be to try and detect/quantify transcripts from these DNA circle products which are proposed to exist (Kinoshita et al PNAS 2001; doi:10.1073/pnas.221454398). If detectable this could provide information about the class switch history of a single cell which would compliment the predictive power of sciCSR.
5. Line 181 – this refers to cells circulating in blood, that have undergone B cell maturation but the text is a little vague and implies they are undergoing this process. Please rephrase to more accurately reflect this.
6. Fig 2a – annotation of sterile transcripts coordinates/intervals on genome track would perhaps help
7. Is the unique mappability of sterile Ig transcripts as shown in Figure 2 greater than that of the productive transcripts? The accuracy of IgH isotype quantitation (eg IgG1 vs IgG3 expression) in scRNA is something that would be valuable to address.
8. Fig 1a – I would consider moving the SHM part of this schematic to supplementary figure to focus on CSR, potentially combining this figure with that in Fig 3.

Reviewer #3:

Remarks to the Author:

In their manuscript 'sciCSR infers B cell state transition and predicts class-switch recombination dynamics 2 using single-cell transcriptomic data' Ng et al develop a novel computational approach to the analysis of class switching outcomes during the B cell response from single-cell RNA sequencing data. In developing this approach, the authors overcome a significant and very interesting analytical challenge, that is accurately discriminating between (and subsequently quantifying) germline transcripts and genuine antibody transcripts. This has been a major confounder of single cell genomics studies of the human and mouse B cell response to date, and is a problem for which a standardised analytical approach has yet to emerge. Likely this is because such a task is highly laborious and, as is evident here, requires the manual annotation of intervening exon sequences using genomic coordinates derived from the foundational papers that first identified sterile/germline transcript sequences using molecular

cloning techniques. By leveraging manual annotations in combination with UMIs in scRNA-seq data, the authors could accurately map individual transcripts that span both the IgH constant coding region and either the recombined VDJ or intervening exons, depending on the transcript from which the mapped read originated. This then enabled the enumeration of both genuine antibody mRNA sequences as well as germline transcripts in single B cells. The authors should be congratulated for their efforts in deriving an elegant solution to this problem. They have written a manuscript and piece of software that will be of much interest and value to the field.

In addition to the above, a key highlight of this paper is found in figure 4. Here, the authors utilise their algorithm, sciCSR, to accurately predict the class switching trajectory of B cells across three time points during a vaccine response in two human donors. The accuracy of prediction demonstrated here is quite remarkable, and validates not only their algorithm, but also their hypothesis driven approach to utilising well-characterised features of B cell responses to augment and improve cell state transition inference. Something that has been lacking from many single cell studies.

Overall, this paper is well written, with innovative analytical approaches and detailed explanations of the methods used. The authors sample from appropriate and diverse publicly available data sets that strengthen their claims of robustness and reproducibility. I would endorse this paper for publication pending some responses to minor comments below.

Major comments:

None

Minor comments:

- Could the authors please clarify how their calculation of "% flux" maps onto a biological meaning? In figure 4, % flux appears to be used as a direct prediction of the percentage of B cells in the subsequent time point that have undergone switching to a specified isotype. Does that mean % flux is equivalent to a quantitative prediction of the future frequency of CSR within a population of B cells?

- How does % flux differ from simply using the quantified % positive cells for a given sterile transcript to make a prediction of future isotype frequency? For instance, in figure 4, donor 07, in the prediction from week 7 to week 15, do you observe a large percentage of IgG1+ B cells that also express Ia1 sterile transcripts and does this percentage correlate with future IgA1 switching in week 15? What additional information is gained from using flux rather than the simple enumeration of the frequency of sterile transcript expressing cells?

To this point, could the authors comment on a potential discrepancy between this paper and the one published recently by the Hodgkin group, which has been discussed in this manuscript. The predictive

model in the Hodgkin paper used both Aicda and sterile transcripts to make isotype predictions, where it seems sciCSR can make similarly highly accurate quantitative predictions using sterile transcripts alone. Why might this be the case? Are the cells analysed in figure 4 all GC B cells and therefore highly enriched for Aicda expression, making its incorporation into a prediction redundant? Or is there another explanation?

- Perhaps some of the detail behind figure 5 has been lost on me, but the analysis of the Gomez-Escobar et al and Zhai et al data does not seem entirely necessary in the absence of a direct comparison between the predicted isotype distributions and the distributions measured in the corresponding data sets. If I have misunderstood the meaning of figure 5b, I would greatly appreciate some clarification from the authors. Alternatively, could the authors include a direct comparison of sciCSR predicted distributions of isotypes versus what was measured in the publications?

Further to this, I would ask the authors to consider moving figure 5 into supplementary entirely. Although the comparative analysis of RNA velocity versus CSR/SHM for B cell state transition inference is interesting, I think it is significantly less impactful than the strong punchline to the paper that is provided by figure 4. This is, of course, merely a suggestion for the the authors to consider, but the conclusion from figure 4 is the natural high point in the narrative of the paper and I believe it is the ideal final figure on which to conclude.

Typos: line 473 usea -> uses

Author Rebuttal to Initial comments

Response to Reviewers

Ng et al., “sciCSR infers B cell state transition and predicts class-switch recombination dynamics using single-cell transcriptomic data”

We would like to thank the three reviewers for their thorough review of the manuscript and software. We are very pleased that all reviewers recognise the novelty of sciCSR as well as its potential to address a clear gap in existing scRNA-seq analysis tools as applied in B-cell transcriptomics datasets.

Below we include a point-by-point response to comments made by each reviewer (printed in red). Changes in the manuscript are highlighted in blue. Wherever possible we have referenced the relevant line numbers where changes have been made to the main manuscript; line numbers were taken from the revised version *with* track changes. We note that in the revised manuscript we have added a new figure (Figure 3 in the revision) corresponding to the new IFN γ scRNA-seq data; therefore wherever possible we will clarify references to figure as “old/original Figure 4”, “new Figure 6” etc. to avoid confusion.

Reviewer #1

During B cell maturation class-switch recombination occurs. This allows the B cell to switch the effector function of the antibody they produce. Understanding how B cell isotype switching occurs is important to understand how B cells mature across time. Here, the authors introduce sciCSR, a tool that can infer B cell state transitions and predict class-switching. This method focuses on B cells and immune repertoire data, unlike conventional RNA velocity methods, by identifying productive and sterile transcripts and using a Markov state model to infer CSR dynamics.

The authors claim that this approach leads to assessing isotype distribution with high accuracy, accurately capturing class-switch patterns in mouse models, and leads to a tool better than other RNA velocity methods for analyzing immune dynamics. To show this, the authors benchmarked their tool using both simulations and empirical datasets.

My strongest endorsement of this paper is that, before I was even asked to write this review I had already read the preprint and was using the software on my own data. I have some concerns detailed below, but if addressed I think this paper will be an important and novel contribution to both immunology and single cell analysis.

Validity and originality/significance/conclusion

The major strengths of this approach is the differentiation between sterile and productive transcripts, and its use of this information to infer B cell dynamics. Shown through their benchmarking, the method appears to capture known biological trends and independent information from traditional RNA velocity methods when applied to B cells. The methods are both novel and address a clear need in the field.

We thank Reviewer 1 for their thorough review of both the code and the manuscript. We are encouraged by their positive endorsement of this work and the constructive comments they made for improving the tool and the paper – we have implemented most of the suggestions. We are particularly pleased to hear the reviewer’s first-hand experience in using sciCSR to analyse their own dataset.

Comments

My greatest concern was with the experiments shown in Fig 5. I like the idea of using these knock-out mice to try to recapitulate their unique biology with sciCSR. However, the knockout experiments are not well motivated in the text and the results are not adequately explained. What is expected from AICDA, HSP13, and IL23 knockouts and how are the results validated by them?

We apologise and agree that the analysis shown in the original Figure 5 could be better motivated and explained. In this new version we have revised this figure (now Figure 6) and the surrounding discussion (lines 395-401; line numbers refer to version with tracked changes) to give more context to what the expected CSR outcomes of these systems are, and how sciCSR recapitulate these findings. All three knockouts have been described in their original publications to have CSR effects – the idea was to apply sciCSR to validate that the method could reproduce these results:

- *Aicda* (Gómez-Escolar et al. EMBO Rep 2022, <https://doi.org/10.15252/embr.202255000>): Knockout reduced CSR to any isotype. The original paper discussed the consequence on *Aicda* deficiency on transcriptomic profiles and transitions between B cell maturation stages. We verify that in the knockout mice there is a reduction of isotype-switched B cells in the scRNA-seq data (new Figure 6b in the main text). This dataset was originally used with the intention as a negative control (given that *Aicda* is a known critical factor in initiating CSR).
- *Il23* (Hong et al. J Immunol. 2020, <https://doi.org/10.4049/jimmunol.2000280>): Knockout specifically reduced CSR towards IgG2b. This was verified by the authors via measuring serum immunoglobulin levels using ELISA (Hong et al. Figure 4b) and immunofluorescence (Figure 4c in Hong et al.). We have now shown the sciCSR predicted isotype distribution side-by-side with the measured isotype distribution based on scBCR-seq from the Hong et al. dataset (new Figure 6c in the main text).
- *Hspa13* (Zhai et al. Mol Immunol 2022, <https://doi.org/10.1016/j.molimm.2021.11.014>): Knockout reduced CSR and SHM to any isotype. This is shown in the original publication (Figure 4a) using scBCR-seq data, and can be corroborated with the expression of *Aicda* in the wild-type versus *Hspa13* knockout cells (Figure 4c, Zhai et al.). Unfortunately scBCR-seq data were not available for all samples assayed in scRNA-seq, and we believe that for one of the samples where scBCR-seq data were collected, the correct FASTQ files were not uploaded to the ArrayExpress repository by mistake (the available scBCR-seq FASTQ file corresponds to scRNA-seq reads, not scBCR-seq). We did not find the raw data corresponds to the isotype distribution depicted in Zhai et al Figure 4a, and therefore in this revision we removed analysis of the Zhai et al. data as we could not obtain and display the isotype distribution using a measurement orthogonal to scRNA-seq + sciCSR.

For the Gómez-Escolar et al. (*Aicda*) and Hong et al. (*Il23*) datasets, we now display the isotype distribution obtained from scBCR-seq data, side-by-side with the stationary distributions obtained from the Markov state models fitted using sciCSR (now main text Figure 6b,c). These data show that the sciCSR model recapitulates the observed isotype distribution of the cells.

- 5A and 5D are low in information content, while 5C and 5E are so dense they're hard to interpret. For

instance, I think it would be better to show the schematic of 5D using the actual data, so we can see how the trajectories are different.

We have reorganized this figure and added new figure panels to present a more thorough analysis of these data. Specifically, we present the simulated trajectories based on the Markov state models fitted using RNA velocity/CSR/SHM information, which show that in the Hong et al. data RNA velocity biases sampling to specific cell clusters whilst CSR and SHM allows a more even sampling of states in the models (now Figure 6e,f); these results are presented both by projecting the Markov-chain-sampled trajectories onto the UMAP plot (now Figure 6f), as well as depiction of stationary distributions from the Markov models in comparison with the observed distribution of cell clusters (now Figure 6e). We have kept the schematic in Figure 6a just to provide a clear illustration of the experimental setup in these two datasets deployed as case studies.

- 5B Why is no CSR predicted for AICDA KO mice? I understand that AICDA KO should have not have class switch recombination, but I would expect those mice to still have sterile isotype transcription. Why does this not lead to predicted switching? Additionally, what does HSP13 do in class switch recombination and why do the observations fit those expectations? This should be explained in the main text. It would also be useful to show the sterile vs productive dot plot for all of these mice, perhaps as a supplemental figure.

The reviewer is correct that there is still sterile transcription in the *Aicda* (and *Hspa13*) knockouts - sterile and productive transcription levels of these cells were shown in the Supplementary Figure S9. The original Figure 5b shows the isotype distribution predicted by sciCSR (i.e. we are showing predicted *productive* immunoglobulins). In the revision we show these predicted productive distributions side by side with the isotype distribution obtained from scBCR-seq to show that these predictions reproduce the observed isotypes of the cells (new Figure 6b,c). We hope our response to the first question in this reply, as well as the reorganised discussion surrounding this analysis (lines 395-401) help in clarifying how sciCSR predictions conform to the expected outcomes of the experiments.

Further to my point about Fig 5B, it wasn't clear what isotype information is used to predict potential isotype switching to other types. More specifically, is switching to an isotype predicted by sterile or productive transcription of that isotype?

We thank the reviewer for this question which allows us the opportunity to clarify how we define the "CSR prediction" problem: we use the productive transcript expression to group cells, and both the productive and sterile transcription to predict isotype switching (here framed as transitions between states under the Markov chain framework) - below we will explain conceptually how the productive and sterile transcripts are used in approaching this problem, with reference to the figures shown in the manuscript:

Operationally, in sciCSR CSR is considered as the transitions between states (isotypes), defined as group of B cells of each given *productive* isotype. The figures depicting the prediction results (new Figures 5, 6) show the summary of the dynamics of this process of state transition, between groups of B cells of different *productive* isotypes. That is, in the bar plots we show the predicted proportions of cells expressing the productive transcripts of each isotype, and in the dot-plots depicting fluxes (e.g. new Figure 5c,d) the dots represent the amount of transitions between B cells expressing productive transcripts of isotypes X and Y.

The productive transcripts are used as a grouping variable of the individual observations (cells); as such productive transcripts alone do not give information about the likelihood to switch between our defined states. Sterile transcripts provide information to instruct the model on the likelihood of transitions between the different groups of cells defined by their isotypes. We use both sterile and productive transcripts to derive a pseudotime ordering ("CSR potential" as defined in the manuscript) that encodes instruction as to how the cells should be ordered based on isotype expression. This ordering is used to predict switches from and towards every isotype. This procedure of ordering cells based on the productive *and* sterile transcript expression is depicted in the new Figure 4.

- 5C I am generally confused as to what is shown and why it fits with the expectations of the experimental system.

The IL23 knockout in Hong et al. reduces switching to IgG2b: we now show the observed isotype distribution based on the scBCR-seq data included in Hong et al., alongside with the stationary distribution obtained from fitting Markov state model on the data using sciCSR (new Figure 6c). These data show that the sciCSR model recapitulates the drop in IgG2b/c in the knockouts, which is identical to what is shown in the observed scBCR-seq isotype distributions.

- 5E it is not clear to me what is supposed to be concluded from this figure. What switching patterns are expected from this mouse system? Why would we expect the CSR and SHM trajectories to be more similar to RNA velocity in the IL23p19 mouse?

In this revision we present a more detailed analysis of the trajectories sampled based on CSR/SHM/RNA velocity information. We observe that the Markov models fitted using RNA velocity only managed to sample one germinal centre dark zone B cell cluster as well as the memory clusters, whilst missing the majority of other states involving different GC B cell subsets. We now show these sampled trajectories both as heatmaps depicting the sampling distribution (now Figure 6e) and as contour plots showing the frequency of visiting each state in the sampled transitions (now Figure 6f).

Since the models were fitted separately on wild-type and knockout cells, the comparisons of trajectories are intended to be considered internally, i.e. within the wild-type condition to compare velocity/CSR/SHM and similarly for the knockouts. Inspecting the sampling distributions (new Figure 6f), the sampling bias in the RNA velocity models appear to be more severe in the WT data in comparison with the knockouts, and consequently the WT RNA velocity trajectory is more different to CSR/SHM-based trajectories compared to the knockouts.

When describing the results of Fig 5, the authors claim that CSR and SHM trajectories give very similar predictions and that this information is different from what is provided by RNA velocity. It's noted that RNA velocity predictions are noisier and potentially less reliable, but this is not demonstrated, and it's possible the CSR/SHM trajectories are in fact less accurate. While addressing which prediction is better in general may be beyond the scope of the paper, it's important to show how these predicted trajectories differ and what aspects of B cell biology are being captured by RNA velocity vs CSR/SHM trajectories.

We hope our response to the comment immediately before this has addressed this point.

For Figure 4, it is not clear from the figure how the isotype predictions are different from the prior state of the system. While the isotype frequency at each week is shown in Fig 4b as a heatmap, it is hard to compare to the histograms in the rest of the figure. Instead, the frequency at w7 and w15 should be shown in the same vertical histogram as the others in Fig 4c and 4d to allow for better comparisons. Further, how well is the isotype frequency at the later timepoint (e.g. w15) predicted from the simple frequency at the earlier timepoint (e.g. w7)? This could be answered by showing the cosine similarity between the frequency at w7 vs frequency at w15 in addition to the cosine similarity between the prediction at w7 and the frequency at w15.

For Figure 4 (now Figure 5), the observed isotype frequencies at w7 and w15 in the heatmap (original Figure 5b) are identical to the bar plots in panels c and d – the underlying data are the same, just presented with two different data visualisation formats. We have re-plotted the heatmap as bar plot (now Figure 5b); we have also included the raw numbers behind these figures as Source Data File 4 for reference.

We thank the reviewer for the suggestion of examining similarity between the frequency at w7 versus frequency at w15 to compare against the predictions made using sciCSR. These are now included in the new Figure 5b, and show that in all but one case sciCSR predictions are substantially more similar to the future isotype distribution than merely using the current (productive) isotype distribution for prediction; the sciCSR predictions therefore do include additional insights which cannot be gauged simply by analysing scBCR-seq data across timepoints.

In my experimentation with the software, I noticed that in my own data the uncategorized (-C) reads made up a substantial portion of the overall constant region read count. It appears as if these reads are ignored, which makes me nervous given they are so numerous. Can the authors demonstrate that there are no systematic biases introduced by ignoring uncategorized reads, or show that such biases would not affect their analyses? For instance, it is possible that productive reads are more likely to be uncategorized than sterile reads, leading to systematic underestimation of productive reads.

We thank the reviewer for this question. The uncategorised (-C) reads do make up a substantial portion of reads in the constant region. The under-/over-sampling of reads could either be due to (i) the sequence similarity between isotypes which leads to uncertainty in aligning these reads to the reference genome, or (ii) the enrichment of transcripts at either the 5' or the 3' end during library preparation, most notably using the popular 10X Genomics single-cell technologies. Our analyses on the simulated data suggest that (i) is unlikely, because typical (sc)RNA-seq alignment tools can accurately reproduce the ground-truth isotypes even when presented with a mixture of IgG/IgA subtypes (main text Figure 2c). For (ii), in theory the further a region of interest is from the 5'/3' end, the less likely it is for a read to be sampled in the resulting sequencing library. In this sense, the exonic region encoding the constant regions (C) is further away from the 5' end in the sterile case rather than productive: these C-region exons are around 5 kb from the transcription start site for sterile transcript, and VDJ-recombined sequences are typically 300-400 bp long. Therefore, the reviewer is correct that the -C read counts are more likely to originate from productive transcripts. This is an unavoidable caveat due to the nature of the scRNA-seq library preparation protocols which involve tagging and enrichment of short reads.

We can directly quantify this in the simulated data where we know the origin (sterile or productive) of the simulated reads. The simulated data do confirm this idea – because the C exonic region is closer to the 5'

end for the productive transcripts, it is far more likely for these –C reads to arise from productive transcripts (Table i):

Table i. Reads sampled from exonic region of the C genes in the simulated data (cf main text Figure 2b,c)

Ground-truth	% of read sampled from C exonic region
100% IGHG1-productive	85.01%
100% IGHA1-productive	71.61%
100% IGHG1-sterile	32.04%
100% IGHA1-sterile	27.31%

We have recognised this under-sampling issue of the productive transcripts, therefore wherever possible we use the isotype annotation from scBCR-seq data to annotate the productive isotype of the cells, taking advantage of the targeted enrichment of productive transcripts in scBCR-seq library preparation. In our analysis we focus on the sterile transcript counts when interpreting dot-plots such as that shown in Figure 2d, and use the scBCR-seq-based isotype annotations to describe the productive transcript landscape of the cells. We have added a caution with respect to interpreting productive transcription levels in the main text (lines 204-208, and 494-498).

Besides grouping cells by productive transcript expression, the only other application of the productive transcript counts is in deriving the pseudotemporal ordering to infer transitions: here, we have shown with real-life data that the CSR potential, derived using the productive and sterile transcripts enumerated directly from scRNA-seq, is capable of sorting different B cell subsets in their expected order (Figure 4b,c). We have also generated similar visualisations for B cells sampled from tonsils, showing that the ordering derived from the productive/sterile transcript counts from scRNA-seq can also correctly order cells from naïve → germinal centre → plasmablasts (also included in the new Figure 4b). These suggest that these CSR potential scores still function expectedly despite the aforementioned sampling issues. We do note, however, that this may not apply when confronted with datasets generated using other single-cell protocols, or sequenced to much higher depth than that used in generating the reference atlases. Unfortunately without a benchmarking dataset collection of scRNA-seq B cells generated using different protocols, we were unable to explore this further; we have noted this in the discussion section (lines 504-506) in this revision to emphasise that further systematic study of sterile transcription, including its detection based on various scRNA-seq library preparation and sequencing methods, would contribute to a more accurate characterisation of the productive/sterile transcription landscape of B cells.

Additionally, the methods do not sufficiently explain TPT. The description is very process-oriented, describing data structures and software functions, but does not adequately describe conceptually what is being done. For instance, where do the values in the F matrix come from and what do they represent? How is TPT different from regular continuous time Markov chains, which are used, for instance, in phylogenetic models? What is the “transition rate” in line 763?

We apologise and agree that in the original version the explanation of TPT was too operative and didn't offer a conceptual overview of its principles. Here we offer a brief conceptual overview of TPT; a brief

summary of these concepts have now been added to the manuscript (lines 307-324 in the Result section; lines 772-780 & 798-802 in the Methods) and as Supplementary Note 2. We would first like to point out that the Markov models fitted in sciCSR are essentially continuous-time Markov chains: we are modelling the state transitions (class-switching) along a pseudotemporal axis, which is defined using CSR/SHM signals from the data. In this sense, we can perform typical Markovian analyses on the Markov models generated in sciCSR, such as simulating new sequences of states (switching events) which are consistent with the seen data generated under the same experimental condition (we used this property to compare the transition information embedded in CSR/SHM/RNA velocity, by comparing the frequency different cell states are visited in these simulated sequences, see main text new Figure 6d-f).

Classically, in applying Markov chains to model biological sequences and phylogenies, we are interested in a maximum likelihood solution to describe the observed sequences and/or tree topologies. Whilst we can formulate the problem of modelling CSR as nominating a maximum-likelihood solution, i.e. the most likely series of switching events that is consistent with the given data, here we are interested in the dynamics of these switching events, not only the maximum-likelihood solution: given the data, we aim to extract all the possible transitions and describe their dynamics, assigning likelihood of sampling these transitions in the Markov chains. Note this is also in contrast to models of cellular dynamics in developmental biology systems, where there is normally one clear pathway towards a certain cell fate from the progenitor; here we are interested more in the range of possible pathways the system can take in CSR.

Transition Path Theory (TPT) provides a solution which allows us to analyse this ensemble of transition paths: we start at a pre-defined source state and enumerate, given the transition likelihoods between the states, possible paths which can traverse the states and reach the (again predefined) target. This allows us to calculate the frequency of transition paths that would flow between any two states: this represents the “flux” values in the F matrix. Because TPT considers all transitions, fluxes refer to *any* switching events involving the two states: for example, to switch from isotype X to Y, one can switch directly without an intermediate isotype, or switch via an intermediate Z. Moreover, the transition does not always occur successfully – it is possible to have attempted transitions which ultimately fall back to the starting state because, for example, the sterile transcription level is insufficient or AID is absent. Fluxes incorporate all successful transitions that switch from state X to Y; these ‘fluxes’ therefore represent a holistic, information-rich summary of the dynamics of the system, normally omitted in maximum-likelihood analyses of outputs from Markov models.

Additionally, with this ensemble of possible paths of traversal, one can also estimate the amount of time (in arbitrary unit) needed to reach a target state after leaving the source state; this is what ‘transition rate’ means in this context; we have rephrased this sentence in the revision (lines 816-818). This number on its own would not have much meaning since it is in an arbitrary unit and one would need to contextualise this, e.g. by comparing against controls, to conclude anything regarding the efficiency of a specific transition of interest.

It is difficult to understand how CSR is incorporated in tandem with the SHM information when creating the M_pseudotime matrix. For example, how would a cell of “early” CSR information like high IGM productive transcripts but high SHM rate compare to cells with higher CSR potential but low SHM?

In the original version of the manuscript, we did not use CSR in tandem with SHM to generate the pseudotime ordering of cells. This would be a useful extension of the method to examine the interplay

between these two processes in estimating the dynamics of B cells, however, it is not immediately obvious how these two sources of information can be integrated in one transition matrix. The CellRank method, which provides the code-base for deriving transition matrices, does so by biasing the transition matrix defined using transcriptomic similarity (let this be M_{RNA} for notation purposes) with a separate transition matrix defined using directional information such as RNA velocity (let this be $M_{velocity}$). These two sources of transition inferences are integrated as such:

$$M = 0.8 \times M_{velocity} + 0.2 \times M_{RNA}$$

The authors of CellRank claimed that the choice of constants here are optimal in the datasets they have used (Lange et al. Nature Methods 2022 doi: 10.1038/s41592-021-01346-6). Combining CSR and SHM information would involve adding a third term to the equation here and calibration of the weights for each individual M matrix. We feel that doing so would require benchmarking using datasets where a range of possible combinations of CSR/SHM levels (such as what the reviewer proposes) have been sampled and the ground-truth transitions being well-established. We decided to leave this to a future extension of the software, upon availability of such datasets. We have clarified in the text that either CSR or SHM can be used as information to estimate the transition matrix (lines 333-335).

Code review

As mentioned above, I had previously downloaded and used sciCSR on my own data. I applied it to a 10X dataset in a project oriented around isotype switching. We had previously used RT-PCR to quantify sterile isotype transcription of a particular isotype in a population of B cells. Applying sciCSR to this population confirmed those results - much more sterile than productive transcripts. Further, cells with productive transcripts of that isotype generally mapped back to cells with those constant region BCRs (despite not including BCR data in that run of sciCSR). While I haven't tested the cell trajectory analysis, I'm happy with sciCSR's performance on real data.

We are very pleased to hear first-hand experience of using sciCSR to analyse their own datasets.

I found the functionality and documentation of the software to be good. One thing that appears to be missing from the github is automatic unit tests. While probably not required for publication, I strongly suggest adding those for code maintenance. For the software documentation I investigated the basic tutorial on their site. While the package seems to be a mixture of R and Python, setup was relatively painless. By the standards of most academic software, the documentation and version control is good. My one gripe is that the code in the section on adding "a zero matrix" for missing cells is a bit onerous. Better to just add that code block into a function.

While the empirical data analyzed was previously published, the simulated data was not, and should be made available on a permanent repository, such as Zenodo. Further, while the sciCSR package was available, the scripts used to actually perform the analyses in the paper should be made publicly available as well.

We are pleased to hear the reviewer's positive comments about the level of documentation we made available for the sciCSR package as well as their comments about its ease to set up given it deploys a mixture of python and R code. We have spent a substantial amount of time on optimising these details so it is a pleasure to receive recognition for this. Regarding the reviewers' suggestion on points for

improvement, we take them fully on board and agree these changes improve the transparency, reproducibility and sustainability from the perspective of code maintenance:

- Unit tests have now been implemented and included in the R package source code.
- The code to generate a zero matrix for missing cells have been wrapped as a function (“mergeIghCountsToSeurat”).
- Simulated data (Figure 2) has now been made available on Zenodo (<https://dx.doi.org/10.5281/zenodo.8005705>).
- The analysis scripts used are now made available on GitHub (<https://github.com/Fraternalilab/sciCSR-analysis>). This includes the code used to perform the simulation in Figure 2.

Reviewer #2

Ng et al present a new method (sciCSR) for quantifying transcription at the immunoglobulin heavy chain locus in single-cell RNA-seq datasets and using this cell intrinsic information to predict B cell class-switch recombination (CSR) trajectories. This is a well written manuscript that offers a unique and valuable contribution to study of B cell biology with single-cell technologies. Its major advances are providing a method/software package to 1) quantify different types of transcription from IgH isotypes and 2) integrating B cell-specific information (sterile transcripts, isotype expression) and constraints (directionality of the CSR hierarchy) to inform cell trajectory predictions. This has only been possible with bespoke custom scripting in the past and not accessible to the majority of immunologists working with single-cell datasets. By applying sciCSR to observational datasets from human B cell development, a longitudinal time course of human vaccination, and mouse knockout gene models the authors provide evidence that sciCSR can model and predict the CSR dynamics in an immune response, which has a lot of potential to improve our understanding across many fields of immunology. The conclusions drawn are for the most part measured and based on robust methods that are well described in the method section. My major concerns relate to if/how CSR potentials of single-cells can guide downstream analyses about biological questions, and the accuracy of trajectory predictions compared to RNA velocity in Figure 5. It is unclear to me from the figures presented how accurate the cellular trajectories are compared to existing methods.

We thank the reviewer for their appreciation of the potential significance of this work.

Major comments

1. Fig 2b – I found this figure difficult to interpret quantitatively – if possible please present as for analysis between IgG subtypes in Fig 3b.

We assume the reviewer refers to the old Figure 2b and wishes the data to be presented in the same form as the old Figure 2c (in the old Figure 3b we do not present analysis between IgG subtypes)? Since the cases considered in the old Figure 2b refer to transcripts of one specific isotype we do not expect alignment to other isotypes and the figure is confirming this – we observed < 1% of reads aligned to the isotypes other than the one prescribed in the simulation ground-truth. In this revision, conforming to Nature Methods' requirement, we have made available the numbers behind all panels in this Figure 2 as Source Data File 1, so we hope this will help interpreting this and other figure panels.

2. The NMF model uses a training set from peripheral blood of naïve and memory B cells. Does this capture potentially different CSR potential/isotype features of plasmablasts or cells?

The datasets used for NMF include not only B cells from peripheral blood, but also from lymphoid organs (data from human tonsils, King et al. Sci Immunol 2021 doi: 10.1126/sciimmunol.abe6291; murine lymph node, lung and spleen, Mathew et al. Cell Rep 2021 doi: 10.1016/j.celrep.2022.111764). Plasmablasts/plasma cells are included in these datasets. Visualising the CSR potential scores calculated for cells collected from the King et al. tonsil dataset confirm that this pseudotime ordering is capable of highlighting naïve → GC → plasmablast transitions. These data have been added in the new Figure 4b.

3. Line 202-203 says that the NMF is applied to reference atlases of human/mouse B cell maturation, but I don't see any figures/analysis referred to here? Does this relate to the analysis in Figure 4 and 5? Inclusion of this analysis in a similar vein to Fig 3b/c but for CSR potential between naïve → GC → plasma (or similar) would strengthen this section.

Continuing from our response to the last comment, these figures have now been added to the new Figure 4b to highlight that CSR potential correctly orders B cells found in either the periphery or secondary lymphoid organs (tonsils in this case) by the maturation stages of the cells. The NMF signatures are depicted in detail in Supplementary Figure S6.

4. It is a little unclear to me how/if the CSR potential/signature from sciCSR can be used for single cells or to compare the biology/gene expression of different B cell groups ie a lot of the analysis is presented as pseudobulk quantitation for B cells with different isotype. If possible, some visualisation of the scores for single cells as a feature plot or similar. If the data is too sparse for this, it should be discussed.

The CSR potential scores can be visualised for single cells (see Supplementary Figure S3). Similarly, expression levels of sterile transcripts of specific isotypes can also be visualised in a similar manner. We do acknowledge sparsity in the read counts especially for productive transcripts, since their definition requires at least 2 reads sampled from one single molecule (1 on the VDJ region and 1 on the exons of the C gene). This is discussed in the text (lines 204-208; line number refers to version with tracked changes) and illustrated in Supplementary Figure S3. For this reason, wherever possible we annotate the isotype for each single cell based on scBCR-seq data.

5. An exploration of how cells with different CSR potential/signatures could be compared would be a valuable addition to this manuscript e.g. can sciCSR allow for comparison of IgG2-“destined” with IgA1-

“destined” B cells to predict potential regulators or cellular mediators of this class switch decision? Is this analysis possible? If not, please discuss why.

Yes, sciCSR will be the starting point of this analysis. Our tool allows these “CSR-destined” cells to be defined, by means of examining the expression level of sterile transcripts. Users can integrate the productive/sterile quantification offered by sciCSR with other methods to investigate different aspects of molecular profiles associated with these decisions. In this revision we present new data (new Figure 3) where we analyse naïve B cells exposed to anti-IgM, CD40L and IFN γ , which subsequently generate populations “destined” to switch to different IgG subtypes, evidenced by the sterile transcription patterns they display. Using sciCSR we enumerated sterile transcription which confirm the treatment biases towards producing sterile IgG, specifically with strong signal towards IgG3 (new Figure 3d). Users can group the cells by the sterile transcripts they express (by default sciCSR labels cells by the isotype with the highest sterile transcription level), and perform differential gene expression analysis. The grouping function would allow users to compare the molecular profiles of cells exhibiting different class-switching fates, either via differential expression or more sophisticated methods to decipher the underlying gene regulatory network, activated pathways etc. For example, we found, as one would expect, cells positive for IgG sterile transcripts in the cell culture system can be characterized by elevated IFN γ signalling and response (new Figure 3g,h,i).

6. Figures 4 and 5 – the analyses of existing datasets in human and mouse would benefit from including some visualisation of the single-cell maps considered. For example, I find Sup Fig S4 to be very effective visualisation of those datasets that would fit well in the main figure and is then supported by the predictive power of sciCSR. Can the authors include this information to support that that the datasets are of high quality?

We think that current Figure 5b (distribution of isotypes based on scBCR-seq data presented as bar plots) depicts essentially the same information as in Supplementary Figure S4. Since we have added new analysis in the new Figure 5, we opted not to move this Supplementary figure to the main text in order to prevent this figure from being even busier. For the other use cases (now Figures 3, 6), we present single-cell UMAP visualisations (new Figure 3b-c; Figure 6d) to show the subpopulations as well as selected gene markers.

7. Fig 5B – I am unclear how these predicted isotype frequencies compare with the actual isotype frequencies. Can this be presented in the figure? Was the stationary distribution determined from naïve B cells, or all IgM cells?

The actual isotype distribution is now shown side-by-side against the stationary distributions obtained from Markov models fitted using sciCSR (new Figure 6b,c). The stationary distributions were determined from all cells in their respective conditions (wild-type/knockout).

8. Fig 5d-e – More information is needed to appraise how these CSR/SHM trajectories compare with RNA velocity or other pseudotemporal ordering methods. For example, is the RNA velocity trajectory calculated in Fig 5d meaningful in anyway or is it purely noise? This could be assessed by looking at known B cell maturation markers. There is a lot of discussion throughout the manuscript about the limitations of alternative RNA-based trajectory inference methods, and while there are significant limitations in the application of these methods, it is also not clear if the transitions/trajectories from sciCSR are any more

biologically meaningful at a cellular level – Fig 5e just shows that they are different (if I have read it correctly). Visualisation of pseudotemporal trajectory vs CSR flux on the UMAP plots could be one option. This subject should be explored and presented more comprehensively, potentially with the different reference datasets already included.

The reviewer is correct in that in the original version we merely showed that the transitions inferred based on RNA velocity is different from that inferred using CSR/SHM, but we did not describe these differences. In this revision we present a more detailed analysis of these sampled trajectories. We observe that the Markov models fitted using RNA velocity only managed to sample one germinal center dark zone B cell cluster as well as the memory clusters, whilst missing the majority of other states involving different GC B cell subsets. We now show these sampled trajectories both as heatmaps depicting the sampling distribution (now Figure 6h) and as contour plots showing the frequency of visiting each state in the sampled transitions (now Figure 6i).

Minor points

1. Line 55 = “Pseudotemporal ordering” rather than “Pseudotime ordering”

Noted and corrected (line 64).

2. Not much discussion or reference to unguided pseudotemporal ordering methods (eg Monocle can derive pathways from a starting population, without setting the end point. Whether these are accurate or not is another matter).

We have now referenced these methods in the introduction (lines 66-68) – in fact one of the main selling points of CellRank (which sciCSR imports for its machinery for constructing transition matrices) is that the models are capable of automatically detecting the start and end states of the system (this has been mentioned in the main text, lines 295-298). We feel that in the case of CSR, we are investigating states (BCR isotypes) which (a) are straightforward to define (as opposed to cell type annotation in typical scRNA-seq analysis which can be a challenge, depending on the availability of known gene markers), and (b) have well-defined directionality in terms of transitions between states (switches are only possible for isotypes 3' to the current isotype in the genome). We therefore directly group cells by their isotypes and study their transition dynamics.

3. Did the authors consider using the WRCY motif compared to AGCT – the redundancy in AICDA targeting is well documented and may show the Switch regions even more convincingly.

We assume the reviewer is referring to Figure 2a. The distribution of AGCT and WRCY motifs are almost identical (Figure i, see below). Just to clarify, in defining the positions of the sterile transcription initiation sites, we did not directly use the counts of these motifs as evidence – our definitions were based on studies where germline transcripts were identified via restriction cloning and Sanger sequencing; this has been described in the Methods section (lines 614-618). In main text Figure 2a we show AGCT just to highlight the location of the switch regions, following descriptions of many original papers describing switch regions and relevant reviews (Mills, Brooker, and Camerini-Otero. Nucleic Acids Res. 18(24): 7305–

7316. (1990)); Zarrin et al. *Nat. Immunol.* 5, 1275–1281 (2004); Xu et al. *Nat. Rev. Immunol.* 12, 517–531 (2012).

Figure i. Distribution of AGCT (top) and WRCY nucleotide strings across the human IGH locus.

4. Line 99 – reference to transcriptomic footprints from CSR made me think of class switch circles. A possible extension of sciCSR for this or another methods paper could be to try and detect/quantify transcripts from these DNA circle products which are proposed to exist (Kinoshita et al *PNAS* 2001; doi:10.1073/pnas.221454398). If detectable this could provide information about the class switch history of a single cell which would compliment the predictive power of sciCSR.

We thank the reviewer for this insightful comment. In theory transcripts originating from these DNA “switch circles” should exist in the sequenced pool of cDNA molecules – during the initial conception of this work when we analysed the data presented in Stewart et al. (*Front Immunol* 2021; shown in the current Figure 2d, 4b of this paper) we did observe anecdotal examples of reads sharing a unique molecule identifier (UMI) but mapped to the constant region exons of multiple isotypes; presumably these represent read-throughs of the switch circles. We did record these instances in one of the immediate steps of counting the productive/sterile transcripts (function “getIGHmapping”: this function returns a list of a number of outputs, the second of which corresponds to a table where UMIs listing multiple isotypes are tabulated). However, for detecting these extremely rare events it would be more advisable to enriching these switch circles via PCR during the preparation of sequencing libraries. In deriving the count matrix for productive and sterile transcripts we only considered molecules mapped uniquely to only one isotype.

5. Line 181 – this refers to cells circulating in blood, that have undergone B cell maturation but the text is a little vague and implies they are undergoing this process. Please rephrase to more accurately reflect this.

Thank you for pointing this out – we have rephrased this sentence to reflect this (lines 201-204 in this revised version).

6. Fig 2a – annotation of sterile transcripts coordinates/intervals on genome track would perhaps help

The sites for initiating sterile transcriptions have been annotated (as arrows) in Figure 2a. The sterile transcripts share the exons encoding the constant region immunoglobulin protein domains (coloured

boxes in the genome track in Figure 2a) with the productive transcripts. We have made this clear in the legend for Figure 2a.

7. Is the unique mappability of sterile Ig transcripts as shown in Figure 2 greater than that of the productive transcripts? The accuracy of IgH isotype quantitation (eg IgG1 vs IgG3 expression) in scRNA is something that would be valuable to address.

Similar to main text Figure 2c we simulated mixtures of IgG and IgA productive transcripts and aligned them to identify isotypes (Figure ii, see below) – the aligners were still able to reconstitute the ground-truth proportion of isotypes in the simulated mixture except a slight underestimation of IgG1. We however do not observe significant misassignments within IgG/IgA subtypes. RNA-seq aligners typically discard reads which are either unaligned or mapped to more than one loci; judging from these results we are confident that the reported productive transcripts are uniquely assigned to their respective isotypes.

Figure ii. Accuracy of alignment softwares to identify productive IgH isotypes. (a) Comparison of alignment results from the HISAT2 and STAR aligners against ground truth isotype distribution of simulated reads from a mixture of IgG (left) and IgA (right) subtypes. The ground-truth proportions are noted by crosses and reads aligned are divided by the regions they are aligned to (VDJ or C exonic regions). (b) Comparison of ground-truth (x-axis) versus aligned (y-axis) isotypes of reads aligned using the two aligners in the two test cases. Numbers noted correspond to the percentage of reads belonging to these isotypes.

8. Fig 1a – I would consider moving the SHM part of this schematic to supplementary figure to focus on CSR, potentially combining this figure with that in Fig 3.

Figure 1a includes illustration of both the CSR and SHM processes with the intention to explain the mechanisms of both to readers who do not have a B cell immunology background. The idea is to illustrate the point that both processes leave footprints in the scRNA-seq/scBCR-seq data for these processes to be studied, and highlight that sciCSR first attempts to implement steps that extract these relevant signals from the data. In this revision we kept the illustration for both CSR and SHM for this reason.

Reviewer #3

In their manuscript 'sciCSR infers B cell state transition and predicts class-switch recombination dynamics 2 using single-cell transcriptomic data' Ng et al develop a novel computational approach to the analysis of class switching outcomes during the B cell response from single-cell RNA sequencing data. In developing this approach, the authors overcome a significant and very interesting analytical challenge, that is accurately discriminating between (and subsequently quantifying) germline transcripts and genuine antibody transcripts. This has been a major confounder of single cell genomics studies of the human and mouse B cell response to date, and is a problem for which a standardised analytical approach has yet to emerge. Likely this is because such a task is highly laborious and, as is evident here, requires the manual annotation of intervening exon sequences using genomic coordinates derived from the foundational papers that first identified sterile/germline transcript sequences using molecular cloning techniques. By leveraging manual annotations in combination with UMIs in scRNA-seq data, the authors could accurately map individual transcripts that span both the IgH constant coding region and either the recombined VDJ or intervening exons, depending on the transcript from which the mapped read originated. This then enabled the enumeration of both genuine antibody mRNA sequences as well as germline transcripts in single B cells. The authors should be congratulated for their efforts in deriving an elegant solution to this problem. They have written a manuscript and piece of software that will be of much interest and value to the field.

In addition to the above, a key highlight of this paper is found in figure 4. Here, the authors utilise their algorithm, sciCSR, to accurately predict the class switching trajectory of B cells across three time points during a vaccine response in two human donors. The accuracy of prediction demonstrated here is quite remarkable, and validates not only their algorithm, but also their hypothesis driven approach to utilising well-characterised features of B cell responses to augment and improve cell state transition inference. Something that has been lacking from many single cell studies.

Overall, this paper is well written, with innovative analytical approaches and detailed explanations of the methods used. The authors sample from appropriate and diverse publicly available data sets that strengthen their claims of robustness and reproducibility. I would endorse this paper for publication pending some responses to minor comments below.

Major comments:

None

We thank Reviewer 3 for their generous comments and compliments on the novelty of our method. We feel the comments raised below have given us the opportunity to explain better the conceptual advances and uniqueness of the method which we have now clarified in the manuscript.

Minor comments:

- Could the authors please clarify how their calculation of "% flux" maps onto a biological meaning? In figure 4, % flux appears to be used as a direct prediction of the percentage of B cells in the subsequent time point that have undergone switching to a specified isotype. Does that mean % flux is equivalent to a quantitative prediction of the future frequency of CSR within a population of B cells?

The fluxes describe the amount of *direct* transitions between each pair of isotypes based on cells sampled at the given timepoint, regardless of whether switches involve a specific isotype as an intermediate (i.e. the cells would ultimately switch further to other isotypes). Therefore fluxes are indeed quantitative predictions of the isotype distribution at a future timepoint, although their exact accuracies would depend on how much CSR involving each isotype as an intermediate step and how transient this intermediate state is.

- How does % flux differ from simply using the quantified % positive cells for a given sterile transcript to make a prediction of future isotype frequency? For instance, in figure 4, donor 07, in the prediction from week 7 to week 15, do you observe a large percentage of IgG1+ B cells that also express Ia1 sterile transcripts and does this percentage correlate with future IgA1 switching in week 15? What additional information is gained from using flux rather than the simple enumeration of the frequency of sterile transcript expressing cells?

In theory the presence of sterile transcript should indicate future switching events. We can directly quantify this using the data presented in the new Figure 5 as the reviewer suggested (Figure iii below):

Figure iii. Predicting isotype distribution based only on sterile transcription using the Kim et al. data (cf main text Figure 5). (a) Amount of cells of every productive isotype which also express sterile transcripts. Bubble size corresponds to proportion of cells expressing the given productive transcript (x-axis); bubble colour corresponds to proportion of cells expressing the given sterile transcript (y-axis). (b) comparison of distribution

of cells expressing each sterile transcript (green) versus observed isotype distribution (cyan). (c) Cosine similarities between the observed isotype distribution and predicted isotype distribution based on sterile transcript (first column) and inferred flux (second column).

Figure iii above clearly shows that the percentage of cells expressing a given sterile transcript is not as good a predictor of the future isotype distribution compared to the fluxes inferred from the Markov models. An issue with simply using the frequency of sterile transcript expressing cells for prediction is that a substantial number of cells express sterile transcripts of multiple isotypes, and hence its presence/absence alone does not necessarily indicate its class-switch fate. Our approach instead uses the productive/sterile transcription patterns to order cells by their likely positions in the CSR trajectory, and takes advantage of the scRNA-seq dataset in constraining the transitions so that we prioritise transitions between cells with greater transcriptomic similarity. These imply that the model has access to the class-switching potentials of each cell, *relative to other cells in the data*. In applications such as those shown in the current Figure 5, we show that this knowledge of the relative CSR statuses of cells is sufficient to reproduce a measurement made at the population level, namely the isotype distribution of cells at a future timepoint. On the other hand, whether the presence/absence of sterile transcripts can successfully predict future isotype distributions would depend on a number of factors, such as the diversity of sterile transcription within a given pool of cells subject to immune challenge and/or other stimuli, or other factors downstream of sterile transcription which restrict CSR frequency.

To this point, could the authors comment on a potential discrepancy between this paper and the one published recently by the Hodgkin group, which has been discussed in this manuscript. The predictive model in the Hodgkin paper used both *Aicda* and sterile transcripts to make isotype predictions, where it seems sciCSR can make similarly highly accurate quantitative predictions using sterile transcripts alone. Why might this be the case? Are the cells analysed in figure 4 all GC B cells and therefore highly enriched for *Aicda* expression, making its incorporation into a prediction redundant? Or is there another explanation?

In the Hodgkin work (Horton et al. *Immunity* 2022, doi: 10.1016/j.immuni.2022.08.004) the authors monitored CSR at the level of individual lineages of B cells and, as the reviewer summarised, modelled CSR fate using both *Aicda* expression and sterile transcription as predictive features. The difference between the Hodgkin model and sciCSR is, the Hodgkin model predicts the class-switch fate of an individual B cell lineage, whilst sciCSR attempts to describe CSR dynamics of *groups* of (polyclonal) B cells.

By considering the class-switch behaviour of a heterogeneous group of B cell lineages as a whole, sciCSR attempts to offer a description of the CSR dynamics across these lineages. The single-cell resolution of the underlying data allows us to capture the diversity within the population and account for this in the models, but we are not making CSR predictions at the level of individual B cell lineages as in the Hodgkin model. Here we do agree that more information, specifically *Aicda* expression, would be required to make predictions at the level of single B cells, as demonstrated in Horton et al.

In showcasing the application of sciCSR, we compared our predictions, made at the level of the sampled populations of B cells, against the isotype distribution at a subsequent timepoint, also an observation made at the population level. This is in contrast to the Horton et al. paper where predictions of switching fate were made for a descendent cell *of the given lineage*, based on measurements from its precursor(s).

We agree that predictions made at the level of individual cells would have been much more instructive of the general rule of class-switch decisions; whilst this should be the aspiration, there are two complicating factors which make attempting this on currently available scRNA-seq datasets a challenging task:

1. In scRNA-seq datasets, AID transcription expression is typically very low, and is observed almost exclusively in GC B cells. Whilst this is expected, it renders a CSR predictive model based on this feature contingent on the read depth of the scRNA-seq libraries and the cell type makeup of the sample.
2. Many scRNA-seq library preparation protocols, including droplet-based and cell-sorting-based methods, typically sample a very limited number of B cells which preclude the definition of B cell lineages where CSR can be followed through time. Therefore, while we typically could sample thousands of single B cells in a scRNA-seq (and scBCR-seq in parallel) experiment, we would be analysing a mosaic of many B cell lineages, each the result of a very sparse sampling regime of B cells at variable stages of their CSR trajectories.

In developing sciCSR we feel that it is important to create a general-purpose tool which allows us to extract B-cell-specific information to describe the class-switch dynamics of the system – one which is broadly applicable to different scRNA-seq datasets and allow us to mine the ever-growing spectrum of B cell scRNA-seq data made available by the community. The approach of modelling groups of cells instead of individual cellular dynamics is a necessary compromise on this. A future direction on approaching the individual class-switch decision of B cells with sciCSR would probably sampling much more B cell lineages than we currently do in typical scRNA-seq experiments, in conjunction with lineage tracing approaches either via scBCR-seq or the experimental system employed in Horton et al. We have added discussion on this point of modelling the group CSR dynamics, as well as the distinction of sciCSR versus the method applied in Horton et al. in the discussion (lines 520-524; line numbers refer to version with tracked changes). The above points have been summarised and included in the new Supplementary Note 2 as part of the motivation of the sciCSR method.

- Perhaps some of the detail behind figure 5 has been lost on me, but the analysis of the Gomez-Escobar et al and Zhai et al data does not seem entirely necessary in the absence of a direct comparison between the predicted isotype distributions and the distributions measured in the corresponding data sets. If I have misunderstood the meaning of figure 5b, I would greatly appreciate some clarification from the authors. Alternatively, could the authors include a direct comparison of sciCSR predicted distributions of isotypes versus what was measured in the publications?

Further to this, I would ask the authors to consider moving figure 5 into supplementary entirely. Although the comparative analysis of RNA velocity versus CSR/SHM for B cell state transition inference is interesting, I think it is significantly less impactful than the strong punchline to the paper that is provided by figure 4. This is, of course, merely a suggestion for the the authors to consider, but the conclusion from

figure 4 is the natural high point in the narrative of the paper and I believe it is the ideal final figure on which to conclude.

The reviewer is correct that in the original version, for the current Figure 6 we did not provide a direct comparison of the sciCSR predictions against measurements of isotype distribution from the same pool of cells. We have since addressed these comments by showing direct comparisons of sciCSR predictions against the observed distribution. We also present a more thorough analyses of the inferences based on RNA velocity/CSR/SHM (now Figure 6e,f). We think these are very interesting results which show the limitation of RNA velocity in sampling certain B cell states given the data, and support the idea behind using CSR/SHM information to study transitions between B cell isotypes and states, which is one of the core functions of sciCSR. We think the revised analyses shown in this figure merits its inclusion and fits the flow of the manuscript, in showing that the predictions enabled by sciCSR offer a more thorough view of cellular dynamics in this case of analysing B cells, as a conclusion relevant also to the general audience including those coming from the perspective of single-cell data analytics at large, not specific to B cell immunology.

Typos: line 473 usea -> uses

Noted – this typo in the figure legend of the current Figure 4c has now been corrected.

Decision Letter, first revision:

Dear Franca,

Thank you for submitting your revised manuscript "sciCSR infers B cell state transition and predicts class-switch recombination dynamics using single-cell transcriptomic data" (NMETH-A51799A). It has now been seen by the original referees and their comments are below. The reviewers find that the paper has improved in revision, and therefore we'll be happy in principle to publish it in Nature Methods, pending minor revisions to satisfy the referees' final requests and to comply with our editorial and formatting guidelines.

TRANSPARENT PEER REVIEW

Nature Methods offers a transparent peer review option for new original research manuscripts submitted from 17th February 2021. We encourage increased transparency in peer review by publishing the reviewer comments, author rebuttal letters and editorial decision letters if the authors agree. Such peer review material is made available as a supplementary peer review file. Please state in the cover letter 'I wish to participate in transparent peer review' if you want to opt in, or 'I do not wish to participate in transparent peer review' if you don't. Failure to state your preference will result in delays in accepting your manuscript for publication.

ORCID

Sincerely,
Madhura

Madhura Mukhopadhyay, PhD
Senior Editor
Nature Methods

Reviewer #1 (Remarks to the Author):

The authors have done a great job of responding to the critiques. I'm satisfied with the author's changes besides the following minor points:

- When dealing with uncategorized (-C) reads, the authors show that these likely originate from productive transcripts, and that the transcript levels are therefore likely biased away from productive transcripts by a significant fraction. I appreciate the author's thorough response to this question and understand that it likely doesn't affect the analyses presented. However, I don't think their response in the main text (lines 204-207) adequately addresses this issue as it is quite unspecific. If the method is biased against detecting productive transcripts, this is a significant caveat to interpreting the output and should be plainly stated in the text, preferably accompanied with something like Table i. For example, it may be tempting for users to compare the level of productive vs sterile transcripts returned by the method. However, it doesn't seem like this comparison should not be done as the method is biased against detecting productive transcripts.
- Could the authors provide more detail and justification for the settings used to run the RNA velocity analysis in Fig 6e-6f?

Kenneth Hoehn, Noah Yann Lee, Cole Jensen

Reviewer #2 (Remarks to the Author):

I'm pleased to endorse publication - they have addressed all of my comments and concerns.

Reviewer #3 (Remarks to the Author):

The authors have addressed all of the concerns I raised during the initial round of review. I believe the inclusion of the new analysis has improved the manuscript and is suitable for publication, which will be of significant interest to the field.

Author Rebuttal, first revision:

Response to Reviewers

Ng et al., “sciCSR infers B cell state transition and predicts class-switch recombination dynamics using single-cell transcriptomic data”

We would like to thank the three reviewers for once again agreeing to review this paper, and we are happy that all reviewers recognise the improvements of the manuscript. We are glad that Reviewers 2 and 3 have endorsed publication of this work.

Below we respond to comments made by Reviewer 1 (printed in red). Changes in the manuscript are highlighted in blue. Wherever possible we have referenced the relevant line numbers (on the version with tracked changes, showing ‘All Markup’ in Microsoft Word) where changes have been made to the main manuscript.

Reviewer #1

(Kenneth Hoehn, Noah Yann Lee, Cole Jensen)

The authors have done a great job of responding to the critiques. I’m satisfied with the author’s changes besides the following minor points:

- When dealing with uncategorized (-C) reads, the authors show that these likely originate from productive transcripts, and that the transcript levels are therefore likely biased away from productive transcripts by a significant fraction. I appreciate the author’s thorough response to this question and understand that it likely doesn’t affect the analyses presented. However, I don’t think their response in the main text (lines 204-207) adequately addresses this issue as it is quite unspecific. If the method is biased against detecting productive transcripts, this is a significant caveat to interpreting the output and should be plainly stated in the text, preferably accompanied with something like Table i. For example, it may be tempting for users to compare the level of productive vs sterile transcripts returned by the method. However, it doesn’t seem like this comparison should not be done as the method is biased against detecting productive transcripts.

We thank Dr Hoehn et al. for their thorough review of this work. We agree that this should be stated explicitly in the manuscript. In this revision we have included the data shown in Table i in the previous reviewer response as a new figure panel (Figure 2e) accompanied by text (starting from line 190 in the new version) which states that sciCSR undersamples productive transcripts in simulated data and therefore direct comparison of productive versus sterile transcription level of a given isotype would not be recommended.

- Could the authors provide more detail and justification for the settings used to run the RNA velocity analysis in Fig 6e-6f?

Details of the settings used to run the RNA velocity analysis are already included in the Methods (lines 839-844). We followed default settings recommended by the developers of the RNA velocity packages: briefly, unspliced and spliced transcripts were enumerated using `velocyto` (v.0.17.17), masking expressed repetitive elements (annotation acquired from the RepeatMasker track of UCSC genome browser for the mm10 mouse reference genome). RNA velocity analysis was performed using `scVelo` using default parameters recommended in the `scVelo` vignette. We have added a sentence in the main text (line 363) to refer the readers to consult the relevant section in the Methods.

Reviewer #2

I'm pleased to endorse publication - they have addressed all of my comments and concerns.

Reviewer #3

The authors have addressed all of the concerns I raised during the initial round of review. I believe the inclusion of the new analysis has improved the manuscript and is suitable for publication, which will be of significant interest to the field.

We thank Reviewers 2 and 3 for revisiting this manuscript and approving our previous revision.

Final Decision Letter:

Dear Franca,

I am pleased to inform you that your Article, "sciCSR infers B cell state transition and predicts class-switch recombination dynamics using single-cell transcriptomic data", has now been accepted for publication in Nature Methods. Your paper is tentatively scheduled for publication in our December print issue, and will be published online prior to that. The received and accepted dates will be Feb 23,

2023 and Oct 02, 2023. This note is intended to let you know what to expect from us over the next month or so, and to let you know where to address any further questions.

Over the next few weeks, your paper will be copyedited to ensure that it conforms to Nature Methods style. Once your paper is typeset, you will receive an email with a link to choose the appropriate publishing options for your paper and our Author Services team will be in touch regarding any additional information that may be required.

You will receive a link to your electronic proof via email with a request to make any corrections within 48 hours. If, when you receive your proof, you cannot meet this deadline, please inform us at rjsproduction@springernature.com immediately.

Please note that *Nature Methods* is a Transformative Journal (TJ). Authors may publish their research with us through the traditional subscription access route or make their paper immediately open access through payment of an article-processing charge (APC). Authors will not be required to make a final decision about access to their article until it has been accepted. [Find out more about Transformative Journals](https://www.springernature.com/gp/open-research/transformative-journals)

Your paper will now be copyedited to ensure that it conforms to Nature Methods style. Once proofs are generated, they will be sent to you electronically and you will be asked to send a corrected version within 24 hours. It is extremely important that you let us know now whether you will be difficult to contact over the next month. If this is the case, we ask that you send us the contact information (email, phone and fax) of someone who will be able to check the proofs and deal with any last-minute problems.

If, when you receive your proof, you cannot meet the deadline, please inform us at rjsproduction@springernature.com immediately.

Once your manuscript is typeset and you have completed the appropriate grant of rights, you will receive a link to your electronic proof via email with a request to make any corrections within 48 hours. If, when you receive your proof, you cannot meet this deadline, please inform us at rjsproduction@springernature.com immediately.

Once your paper has been scheduled for online publication, the Nature press office will be in touch to confirm the details.

Once your paper has been scheduled for online publication, the Nature press office will be in touch to confirm the details.

Content is published online weekly on Mondays and Thursdays, and the embargo is set at 16:00 London time (GMT)/11:00 am US Eastern time (EST) on the day of publication. If you need to know the exact publication date or when the news embargo will be lifted, please contact our press office after you have submitted your proof corrections. Now is the time to inform your Public Relations or Press Office about your paper, as they might be interested in promoting its publication. This will allow them time to prepare an accurate and satisfactory press release. Include your manuscript tracking number NMETH-A51799B and the name of the journal, which they will need when they contact our office.

About one week before your paper is published online, we shall be distributing a press release to news organizations worldwide, which may include details of your work. We are happy for your institution or funding agency to prepare its own press release, but it must mention the embargo date and Nature Methods. Our Press Office will contact you closer to the time of publication, but if you or your Press Office have any inquiries in the meantime, please contact press@nature.com.

Nature Portfolio journals [encourage authors to share their step-by-step experimental protocols](https://www.nature.com/nature-research/editorial-policies/reporting-standards#protocols) on a protocol sharing platform of their choice. Nature Portfolio 's Protocol Exchange is a free-to-use and open resource for protocols; protocols deposited in Protocol Exchange are citable and can be linked from the published article. More details can found at www.nature.com/protocolexchange/about.

Best regards,
Madhura

Madhura Mukhopadhyay, PhD
Senior Editor
Nature Methods